# Catch-22[*]: Pareto Frontier for Detectability and Robustness in LLM Watermarking

## Abstract

Large Language Models (LLMs) generate text through probabilistic token sampling, a mechanism increasingly leveraged for inference-time watermarking to verify AI-generated content. As watermarking schemes proliferate, assessing their robustness-detectability trade-off becomes essential to determine whether watermarks can survive output editing while remaining invisible to adversaries. Current evaluation relies on empirical tests lacking provable guarantees. In this work, we present the first information-theoretic framework that rigorously characterizes this fundamental trade-off. We first establish a hierarchy of sampling-time watermark detectability, ranging from undetectable (distribution-preserving) to highly detectable (biased sampling) schemes. Second, we demonstrate an inverse relationship: watermarks robust to text modifications are inherently more detectable by adversaries, creating an irreducible trilemma: no scheme simultaneously achieves high robustness, low detectability, and reliable verification. Motivated by these theoretical constraints, we propose a hybrid watermarking system that adaptively switches sampling strategies based on LLM output edit levels, achieving Pareto-optimal trade-offs. We show that distribution-preserving schemes provide perfect undetectability; however, they are only robust to near-zero adversarial edits. On the other hand, bias-free and biased sampling offer high robustness guarantees at 15-20% output editing, but with detectable output statistics. At high output editing rates, no watermarking provides robustness guarantees. Lastly, we empirically validate our theoretical trade-off claims with Llama 2 7B and Mistral 7B models under paraphrasing attacks, thereby confirming that Pareto-optimality is only achieved by a hybrid watermarking scheme. Overall, our framework provides watermark evaluation beyond empirical testing via principled design, revealing information-theoretic limits for sampling-based watermarking and how computational hardness shapes which regimes are algorithmically achievable.

## 1 Introduction

Large Language Models (LLMs) have fundamentally transformed natural language generation, producing text increasingly indistinguishable from human authorship Radford et al. (2019). As these models become ubiquitous in text generation Chung et al. (2024) and summarization Liu & Lapata (2019), they enable malicious applications, including the dissemination of misinformation at scale, contamination of training datasets, and erosion of trust in legitimate AI-generated content. The challenge of distinguishing AI-generated from human-written text has thus become critical Stokel-Walker (2022), with inference-time watermarking emerging as the dominant approach for attribution. However, current watermarking schemes face a fundamental trade-off: robust watermarks that survive text editing introduce detectable statistical artifacts ( Gloaguen et al. (2025); Liu et al. (2025)), while provably undetectable watermarks Christ et al. (2024) fail catastrophically under LLM editing as token entropy used to embed the watermark drops Moitra & Golowich (2024).

The rapidly growing class of inference-time LLM watermarking schemes (Fig. 1) employs cryptographic primitives at different stages of token generation to embed verifiable signals in LLM outputs. Biased sampling methods (Kirchenbauer et al. (2023); Zhao et al. (2023)) use hash functions

---

[*]The name alludes to Joseph Heller's *Catch-22*, a paradoxical dilemma in which one decision cannot be made without negating another. In the context of LLMs, watermarks face an analogous bind: improving robustness often makes them more detectable, while reducing detectability weakens their robustness.

Figure 1: Watermarking schemes in modern LLMs exhibit a trade-off between detectability via statistical tests and robustness against LLM output editing.

to designate "green" tokens whose logits are systematically increased, creating detectable statistical signals. Bias-free approaches (Hu et al. (2024); Wu et al. (2024)) employ key-dependent reweighting that preserves expected token distributions while encoding information in variance patterns. Provably undetectable schemes (Christ et al. (2024)) replace sampling randomness with pseudorandom functions (PRFs), achieving perfect undetectability by maintaining exact output distributions. While probability-modifying schemes (biased and bias-free) create redundant statistical signals enabling detection after substantial editing, these deviations are increasingly exposed by black-box statistical tests (Gloaguen et al. (2025)) and targeted prompt analysis (Liu et al. (2025)). Conversely, provable distribution-preserving schemes (Christ et al. (2024)) achieve perfect undetectability but rely on PRF sequences that break under output perturbation, leading to poor robustness. Although recent work claims provable robustness for undetectable watermarks under bounded edit distance (Moitra & Golowich (2024)), their guarantees rely on a language-model vocabulary whose size, while polynomial in a security parameter for any fixed robustness setting, grows exponentially in the inverse of an entropy-rate controlling substring robustness. This dichotomy raises a fundamental question: *What is the inherent trade-off between watermark robustness and detectability?*

In this work, we provide a definitive answer through a unified theoretical framework that establishes the fundamental impossibility of simultaneously achieving high robustness, low detectability, and reliable verification. Our analysis reveals that the empirically observed trade-offs (Kirchenbauer et al. (2024); Zhao et al. (2023)) reflect deep information-theoretic constraints rather than limitations of current techniques. Our framework proceeds in two steps: (i) we quantify detectability via total variation distance between watermarked and unwatermarked distributions, establishing a hierarchy of detectbility (Theorem 1), and then (ii) we characterize the information capacity of watermarked LLM outputs under different-editing levels perceived as noise, revealing how capacity determines robustness guarantees (Theorem 2). This framework allows us to ask the question: *What is an optimal watermarking scheme?*

We answer this through the construction of a hybrid watermarking scheme, which selects between probability-modifying and distribution-preserving methods based on noise levels. This hybrid scheme optimizes the watermark parameters to achieve a Pareto-optimal detectability-robustness trade-off (Theorem 3). Experiments with paraphrasing attacks on watermarked outputs from Llama and Mistral models confirm our hybrid scheme achieves a superior trade-off across all noise regimes.

To summarize, our principal contributions are as follows:

1. **Universal detectability bounds:** We establish design-time information-theoretic limits on watermark detectability independent of specific statistical tests or targeted prompt attacks. Detectability remains constant for Greedy sampling, whereas it increases by $O(|\delta|\sqrt{T})$ for biased sampling with bias $\delta$ and length $T$, $O(\sqrt{T})$ for bias-free sampling, while dropping to zero for distribution-preserving schemes (Theorem 1).

2. **Detectability-robustness characterization using information capacity:** We prove that information capacity is inversely related to the detectability. The channel capacity together with the watermark encoding scheme determines robustness guarantees (Theorem 2).

3. **Optimal hybrid watermark construction:** We propose a hybrid watermarking scheme that switches between probability-modifying and distribution-preserving methods based on the noise levels, achieving Pareto-optimal detectability-robustness trade-offs (Theorem 3).

4. **Experimental validation:** We demonstrate the validity of our theoretical predictions through paraphrasing attacks across open-source Llama and Mistral models, confirming that our hybrid scheme achieves Pareto-optimal robustness even with a 15-20% editing rate, while simultaneously maintaining a total variation distance of $< 0.1$ compared to unwatermarked outputs.

The remainder of this paper is organized as: Section 2 reviews existing watermarking approaches and their limitations. Section 3 develops our information-theoretic framework, followed by Section 4, which derives the optimal hybrid watermark construction. Section 5 validates our theoretical predictions through comprehensive experiments. Finally, Section 6 concludes the paper.

## 2 RELATED WORK ON LLM WATERMARKING AND RESEARCH GAP

Inference-time watermarking for LLMs has evolved rapidly, with schemes progressively trading robustness for undetectability. We categorize existing approaches by their sampling strategies and identify critical gaps that motivate our theoretical framework. Due to space limitations, a comprehensive technical analysis of existing watermarking schemes, along with their corresponding detection schemes, is provided in Appendix A.

**Watermarking via Sampling Modifications.** Existing watermarking schemes modify the token generation process through three distinct approaches:

1. **Biased sampling** (Kirchenbauer et al. (2023); Zhao et al. (2023)) designates certain tokens as "green" at each step and applies an exponential tilt to the sampling probability. While achieving strong empirical robustness (Kirchenbauer et al. (2024)), these schemes are easily detected through statistical tests (Sadasivan et al. (2023); Gloaguen et al. (2025); Liu et al. (2025)).
2. **Bias-free sampling** (Hu et al. (2024); Wu et al. (2024); Kuditipudi et al. (2024)) employs reweighting functions $R_E$ that preserve expected distributions: $\mathbb{E}_E[R_E(p_t)] = p_t$. Despite maintaining first-order unbiasedness, recent work (Gloaguen et al. (2025)) proves all such schemes remain detectable through variance analysis.
3. **Distribution-preserving sampling**[1] (Christ et al. (2024); Zamir (2024)) provably maintains exact token probabilities ($q_t \equiv p_t$) while replacing true randomness with PRFs: $U_t = \mathrm{PRF}(k, \mathrm{context}_t)$. Though achieving provable undetectability, these schemes fail catastrophically under perturbation to LLM outputs.

It is worth noting that, although Moitra & Golowich (2024) proposed a provably undetectable and substring-robust watermarking scheme, their theoretical result requires a language-model alphabet whose size is polynomial in the security parameter but whose polynomial degree scales with $\Theta\left(\frac{1}{\alpha} \log \frac{1}{\alpha}\right)$ in an entropy-rate parameter $\alpha$ governing substring robustness. For realistic natural-language entropy levels and constant-fraction edit robustness, this implies vocabulary sizes far exceeding those of practical fixed-vocabulary LLMs, as described in Appendix A.5.

This landscape reveals a critical gap: **no existing framework quantifies the fundamental limits of the robustness-detectability trade-off**. Prior works lack: (i) information-theoretic bounds on achievable detectability for given robustness requirements, (ii) analysis revealing why undetectable schemes fail under noise, and (iii) principled construction of schemes that optimally navigate this trade-off. Our work addresses these gaps via an information-theoretic framework, as described next.

## 3 INFORMATION-THEORETIC FRAMEWORK FOR ROBUSTNESS VS. DETECTABILITY TRADE-OFF ANALYSIS

The detectability and robustness of watermarked text fundamentally depend on how tokens are sampled during generation. When a language model generates text, it proceeds token by token, computing probability distributions over its vocabulary at each step. The actual text produced depends not just on these probabilities but on the sampling rule that converts probabilities into token choices.

---

[1]Note that we term use *distribution-preserving* for provably undetectable watermarks such as in Christ et al. (2024) unlike statistically indistinguishable watermarks using the same term (Wu et al. (2024)).

Randomness enters this process at each generation step $t$ [2]. The model provides a conditional distribution $p_t(\cdot) = p_\theta(\cdot \mid x, \mathbf{y}_{<t})$ over its vocabulary $\Sigma$, where $x$ denotes the initial prompt and $\mathbf{y}_{<t} = (y_1, \ldots, y_{t-1})$ represents the sequence of tokens already generated. To select a token, we need a source of randomness, typically a uniform random variable $U_t \sim \mathrm{Uniform}[0,1]$. A sampling rule $s$ is a function that takes both $p_t$ and this random variable $U_t$ (possibly along with secret keys) to produce the next token $y_t$. A watermarked sampling rule modifies either the probabilities (creating $q_t \neq p_t$) or the random variable itself (using a keyed pseudorandom function (PRF)), or both.

**Definition 1** (Detectability). *Let $s$ denote a baseline sampling rule that induces a distribution $P^s$ over the space of complete texts $\Omega$, and let $\tilde{s}$ be a keyed watermarked sampling rule that induces $Q^{\tilde{s}}$.*

*(i) **Information-theoretic detectability:** of $\tilde{s}$ relative to $s$ is*

$$\mathrm{Detect}_{\mathrm{IT}}(\tilde{s}) := \mathrm{TV}\big(P^s, Q^{\tilde{s}}\big) = \sup_{A \subseteq \Omega} \big| P^s(A) - Q^{\tilde{s}}(A) \big| \leq \sqrt{\frac{1}{2}\,\mathrm{KL}\big(Q^{\tilde{s}} \| P^s\big)}. \qquad (1)$$

*(ii) **Computational detectability:** Let $\lambda \in \mathbb{N}$ denote a security parameter and $\{\tilde{s}_\lambda\}_{\lambda \in \mathbb{N}}$ a family of watermarked sampling rules with associated baseline and watermarked distributions $P_\lambda$ and $Q_\lambda$ over $\Omega$. For any probabilistic polynomial-time (PPT) detector $D_\lambda : \Omega \to \{0, 1\}$, define distinguishing advantage $\mathrm{Adv}_{D_\lambda}(\lambda) := \big| \mathrm{Pr}_{y \sim Q_\lambda}\big[D_\lambda(y) = 1\big] - \mathrm{Pr}_{y \sim P_\lambda}\big[D_\lambda(y) = 1\big] \big|$. The computational detectability of the family $\{\tilde{s}_\lambda\}$ is given by $\mathrm{Detect}_{\mathrm{comp}}(\tilde{s}_\lambda) := \sup_{D_\lambda \in \mathsf{PPT}} \mathrm{Adv}_{D_\lambda}(\lambda)$*

We analyze four sampling approaches that span the complete spectrum of detectability, based on the watermarking schemes described in Section 2. In addition to the three watermarking approaches, we include greedy sampling as a baseline, which eliminates all randomness by always selecting the most probable token: $v_t^\star = \arg\max_v p_t(v)$. Together, these four approaches enable us to characterize how the $\mathrm{Detect}_{\mathrm{IT}}$ depends on the degree of randomness modification, ranging from complete elimination (greedy) to biased probability adjustments (biased and bias-free sampling) to exact distribution preservation with controlled randomness (distribution-preserving sampling). The corresponding computational detectability $\mathrm{Detect}_{\mathrm{comp}}$, obtained by restricting to polynomial-time detectors, is always upper-bounded by these information-theoretic quantities.

### 3.1 Detectability Characterization

**Theorem 1** (Information-theoretic Detectability). *Fix a prompt $x$ and length $T$. Let $P^s$ denote the baseline distribution induced by standard stochastic sampling from the model, and let $Q$ denote the distribution induced by a given sampling rule. The* single-shot information-theoretic detectability *of the rule, as measured by the total variation distance* $\mathrm{TV}(P^s, Q)$*, satisfies:*

| Sampling Method | Total Variation Distance | Scaling in $T$ |
|---|---|---|
| Greedy | $\mathrm{TV}(P^s, Q^{\mathrm{greedy}}) = 1 - P^s(\mathbf{y}^\star)$ | $O(1)$ |
| Biased ($\delta$-tilt) | $\mathrm{TV}(P^s, Q^{\mathrm{bias}_\delta}) \leq \|\delta\| \sqrt{\dfrac{1}{4} \displaystyle\sum_{t=1}^{T} g_t(1 - g_t)}$ | $O(\|\delta\|\sqrt{T})$ |
| Bias-free (fixed key/code $E$) | $\mathrm{TV}(P^s, Q_E^{\mathrm{bf}}) \leq \sqrt{\dfrac{1}{4} \displaystyle\sum_{t=1}^{T} \sum_v \dfrac{\mathrm{Var}_E[R_E(p_t)(v)]}{p_t(v)}}$ | $O(\sqrt{T})$ |
| Distribution-preserving (per draw) | $\mathrm{TV}(P^s, Q^{\mathrm{prf}}) = 0$ | $0$ |

We denote the distributions as follows: (a) $Q^{\mathrm{greedy}}$, is the unit mass on the deterministic greedy sequence $\mathbf{y}^\star$; (b) $Q^{\mathrm{bias}_\delta}$, is tilted distribution with bias $\delta$ over a keyed green set $G_t$, where $g_t = p_t(G_t) = \sum_{v \in G_t} p_t(v)$; (c) $Q_E^{\mathrm{bf}}$, obtained from an unbiased reweighting operator $R_E$ with $\mathbb{E}_E[R_E(p_t)] = p_t$; and (d) $Q^{\mathrm{prf}}$, which preserves $q_t \equiv p_t$. As clarified in the remark (Appendix C.4), quantities such as $g_t$ and the variance terms, though defined via the random history and key, only appear under expectations in the proof, so the resulting bounds on $\mathrm{TV}(P^s, Q)$ are deterministic for each fixed prompt $x$ and length $T$. The computational detectability always satisfies $\mathrm{Detect}_{\mathrm{comp}}(\tilde{s}_\lambda) \leq \mathrm{TV}(P_\lambda, Q_\lambda)$ (Lemma 1); for the non-cryptographic families in Theorem 1 we later show equality, while cryptographic PRF/PRG-based schemes can exhibit a strict gap. The proof is provided in Appendix C.

---

[2] All the math notations used in this work are described in Appendix B.

> **Interpretation of Theorem 1**
>
> - **Accumulation over length:** $\mathrm{TV}(P^s, Q)$ scales with sequence length $T$ and watermark parameters.
> - **Detectability hierarchy:** Greedy sampling has $O(1)$ detectability, biased sampling grows as $O(|\delta|\sqrt{T})$, bias-free sampling as $O(\sqrt{T})$, and distribution-preserving schemes satisfy $\mathrm{TV} = 0$.
> - **Computational implications:** For PPT keyless detectors, $\mathrm{Detect}_{\mathrm{comp}} \leq \mathrm{Detect}_{\mathrm{IT}}$ and can be strictly smaller for cryptographic watermarks (Appendix C.5); for the non-cryptographic greedy, biased, and bias-free families, the NP test is efficient, so $\mathrm{Detect}_{\mathrm{comp}} = \mathrm{Detect}_{\mathrm{IT}}$.

The analysis, therefore, reveals a clear hierarchy: information-theoretic detectability diminishes as sampling rules employ more sophisticated mechanisms, with PRF-based distribution-preserving schemes achieving $\mathrm{TV} = 0$. Importantly, Theorem 1 is purely information-theoretic and therefore does *not* refute the "provably undetectable" key-based schemes of Christ et al. (2024); Zamir (2024). These constructions explicitly target the distribution-preserving corner of our framework, where $q_t = p_t$ at every step and hence $\mathrm{TV} = 0$ for every key $k$. Their undetectability relies on the computational hardness of distinguishing PRF outputs from true randomness, which is complementary to the scope of Theorem 1. Our key message is therefore targeted: for any *probability-modifying* watermark with a fixed key $k$, one has $Q_k^s \neq P^s$ and thus $\mathrm{TV}(P^s, Q_k^s) > 0$, so an information-theoretic distinguisher always exists, even though exploiting this gap with an efficient keyless detector can itself be computationally hard (Appendix C.5). This is consistent with recent statistical detection schemes, which have already succeeded against biased and bias-free watermarks Gloaguen et al. (2025), and more such detectors are emerging. Theorem 1 provides an information-theoretic explanation for why these practical detectors become increasingly powerful as the sequence length $T$ grows, thereby underpinning the robustness to detectability trade-off explored next.

### 3.2 ROBUSTNESS ANALYSIS UNDER TEXT PERTURBATIONS

The fundamental tension in watermarking lies in balancing *stealth*, i.e., keeping the generated distribution statistically close to the baseline so unauthorized parties cannot reliably distinguish it [3], with *robustness*, i.e., enabling an authorized key holder to detect the watermark after edits or paraphrasing. We quantify stealth via per-sample KL-divergence drift, and robustness via the detection power of a Neyman–Pearson (NP) test at miss probability $\beta$ (power $1 - \beta$).

**Definition 2** (Robustness). *Fix a text length $T$, an edit tolerance $\varepsilon \in [0,1]$, and a false-alarm level $\alpha \in (0,1)$. Let $\mathrm{ED}(\cdot, \cdot)$ denote the edit distance on length-$T$ token sequences. Consider a family of watermarking schemes $\{\tilde{s}_\lambda\}_{\lambda \in \mathbb{N}}$ with corresponding baseline and watermarked sequence distributions $P_\lambda$ and $Q_\lambda$, and let a detector be any map $D : \Omega \to \{0, 1\}$.*

*(i) **Information-theoretic** $(\varepsilon, \alpha, \beta)$-**robustness.** The information-theoretic detection power after edits, denoted $\mathrm{Power}_{\mathrm{IT},\lambda}(\varepsilon, \alpha)$, is the supremum of $\Pr[D(\tilde{y}) = 1 \mid y \sim Q_\lambda, \mathrm{ED}(y, \tilde{y}) \leq \varepsilon T]$ over all measurable detectors $D : \Omega \to \{0, 1\}$ satisfying the false-alarm constraint $\Pr_{y \sim P_\lambda}[D(y) = 1] \leq \alpha$. The family $\{\tilde{s}_\lambda\}$ is said to be $(\varepsilon, \alpha, \beta)$-information-theoretically robust at length $T$ if $\mathrm{Power}_{\mathrm{IT},\lambda}(\varepsilon, \alpha) \geq 1 - \beta$ for all $\lambda$.*

*(ii) **Computational** $(\varepsilon, \alpha, \beta)$-**robustness.** The computational detection power after edits, denoted $\mathrm{Power}_{\mathrm{comp},\lambda}(\varepsilon, \alpha)$, is defined analogously but with the supremum restricted to probabilistic polynomial-time detectors $D \in \mathsf{PPT}$. The family $\{\tilde{s}_\lambda\}$ is said to be $(\varepsilon, \alpha, \beta)$-computationally robust at length $T$ if $\mathrm{Power}_{\mathrm{comp},\lambda}(\varepsilon, \alpha) \geq 1 - \beta$ for all $\lambda$.*

#### 3.2.1 EDIT CHANNEL MODEL

We model edit-based attacks via an abstract substitution channel that changes an $\varepsilon$-fraction of tokens in the watermarked sequence. Concretely, at each position $t$, the original token is left unchanged with probability $1 - \varepsilon$ and, with probability $\varepsilon$, replaced by a token drawn from a fixed noise distribution over the vocabulary $\Sigma$. This i.i.d. channel is not meant to capture the full syntactic or semantic structure of paraphrasing; rather, it is a tractable first-order model whose single parameter $\varepsilon$ is chosen to match the *empirical token edit rate* of the attack as shown via empirical analysis in Appendix G.

To characterize robustness under this channel, we define the *per-token information* at zero edits, $D_0$, for the two watermarking schemes as follows:

---

[3]We use *stealth* to mean low *detectability* (i.e., small total variation between the baseline $P^s$ and $Q$ under a fixed prompt and length $T$) for untrusted parties who do not possess the knowledge of secret key.

- **Biased sampling.** This mechanism applies an exponential tilt toward a key-dependent green set $G \subseteq \Sigma$ with baseline mass $\gamma = \sum_{v \in G} p_t(v)$, producing the modified distribution $q_{t,\delta}(v) \propto p_t(v) e^{\delta \mathbf{1}[v \in G]}$. In the small-signal regime, the per-token information is $D_0^{(\text{biased})} \approx \frac{\delta^2 \gamma(1-\gamma)}{2 \ln 2}$.
- **Bias-free sampling.** This mechanism reweights by $R_E(v)$ with $\mathbb{E}_E[R_E(v)] = 1$, giving $q_{t,E}(v) = p_t(v) R_E(v)$. Defining $\sigma^2(v) = \text{Var}_E[R_E(v)]$ and the average variance $\hat{\sigma}^2 = \sum_v p_t(v) \sigma^2(v)$, the per-token information in the small-signal regime is $D_0^{(\text{bias-free})} \approx \frac{\hat{\sigma}^2}{2 \ln 2}$.

These $D_0$ values represent the noise-free *per-token information budgets* available to an optimal Neyman–Pearson detector in Definition 2, and thus upper-bound the usable signal for any detector, including computationally bounded ones. The *total information budget* across $T$ tokens is $\text{TI}(T) := T \cdot D_0$, the natural analogue of blocklength times per-use information in digital communication. Under edits at rate $\varepsilon$, the difference between the watermarked and baseline token distributions at each position is linearly attenuated by $1 - \varepsilon$. Because KL divergence is locally quadratic in perturbations, this linear attenuation produces a quadratic contraction in the effective per-token information: $D_\varepsilon \approx (1-\varepsilon)^2 D_0$. The resulting information-theoretic channel capacity is therefore:

$$C(\varepsilon) := \sum_{t=1}^{T} D(q_{t,\varepsilon} \| p_{t,\varepsilon}) \approx T(1-\varepsilon)^2 D_0. \tag{2}$$

**Theorem 2** (Watermark Robustness–Detectability). *Fix $T$ and the substitution channel described above. In the small-signal regime, the noise-free per-token information is $D_0^{(\text{biased})} \approx \delta^2 \gamma(1 - \gamma)/(2 \ln 2)$ and $D_0^{(\text{bias-free})} \approx \hat{\sigma}^2/(2 \ln 2)$. Under edits at rate $\varepsilon$, the detector's usable information is $C(\varepsilon) \approx T(1-\varepsilon)^2 D_0$. A sufficient condition for power $1 - \beta$ in the Neyman–Pearson test (at false-alarm level $\alpha$) is $T(1-\varepsilon)^2 D_0 \geq \log_2(1/\beta)$, which yields the maximal tolerable edit rate*

$$\varepsilon_\beta(T, D_0) = 1 - \sqrt{\log_2(1/\beta)/(TD_0)}. \tag{3}$$

This theorem characterizes *information-theoretic* robustness: it describes when there exists a level-$\alpha$ NP detector achieving miss probability of at most $\beta$ after edits, as in Definition 2. For the non-cryptographic biased and bias-free families, this condition also characterizes computational $(\varepsilon, \alpha, \beta)$-robustness, since the Neyman–Pearson detector is efficiently implementable (Appendix D.10). More generally, $C(\varepsilon)$ remains an upper bound on what any detector can achieve, and computational $(\varepsilon, \alpha, \beta)$-robustness for keyless PPT adversaries can be strictly smaller for cryptographic watermarking schemes Christ et al. (2024). The proof (per-token KL expansions, $(1 - \varepsilon)^2$ contraction, chain rule in $T$) is provided in Appendix D. Furthermore, the $(1 - \varepsilon)^2$ bound is specific to the robustness of LLM watermarks, distinct from error correction codes (ECC) on strings which do not have the constraint of string selection from a fixed vocabulary (Appendix D.12).

> **Interpretation of Theorem 2**
>
> - **No single "critical noise" point.** There is no universal edit level at which all methods fail. Each watermark has a turning point (knee) determined by the number of tokens examined and the amount of watermark signal placed per token. More tokens or a stronger signal increase the knee value.
> - **Total information budget.** The watermark provides a fixed information budget $TD_0$ spread across the output. After editing, only a fraction $C(\varepsilon) \approx T(1 - \varepsilon)^2 D_0$ remains, and beyond the knee, no detector, even an information-theoretic one, can compensate once the budget falls below this.
> - **Stealth versus robustness.** If the watermark must remain hard to detect, especially to outsiders who can collect many tokens, the per-token information $D_0$ must be small. Stronger stealth, therefore, lowers the knee and reduces the tolerable editing level.
> - **Computational implications.** The capacity $C(\varepsilon)$ upper-bounds the usable watermark signal for *any* detector, including PPT ones. For non-cryptographic watermarks, the Neyman–Pearson detector is efficient and attains this bound; for cryptographic, keyless PPT adversaries may fall strictly below this envelope.

### 3.3 Implications for watermark design

Theorem 2 provides a simple rule of thumb for design at the information-theoretic level: robustness improves by increasing redundancy via the number of tokens $T$ and the per-token information budget $D_0$, and it degrades quadratically with the edit rate through the factor $(1 - \varepsilon)^2$. For a given power target $\beta$ (in the NP test), the operating boundary is a *knee* in $\varepsilon$ determined solely by $(T, D_0, \beta)$. Any computationally bounded detector, including practical key-holding verifiers, must operate within

this information-theoretic region, i.e., cryptographic design can only shrink the achievable subset (e.g., by weakening keyless adversaries), not expand it. Watermarking methods that allocate the same total information $TD_0$ will therefore share the same boundary (with respect to the NP test), even if they realize the information in different statistical features across LLM outputs. Distribution-preserving (undetectable) schemes sit at the opposite end of this spectrum: because their verification relies on stringent entropy conditions rather than accumulated statistical drift, they are brittle to edits and offer only vanishing tolerance under adversarial perturbations (shown in Appendix D). The following corollary consolidates the baseline information-theoretic impossibility region implied by Theorem 2 and its strengthening when an explicit stealth constraint is imposed.

**Corollary 1** (Impossibility Result). *Fix length $T$, watermark strength $D_0$ (bits/token), and target power $1 - \beta$. Define the knee*

$$\varepsilon_\beta(T, D_0) = 1 - \sqrt{\frac{\log_2(1/\beta)}{T \cdot D_0}}. \tag{4}$$

*For any $\varepsilon > \varepsilon_\beta(T, D_0)$, we have $T(1 - \varepsilon)^2 D_0 < \log_2 \frac{1}{\beta}$, so beyond this boundary, reliable detection at the specified power is unattainable for any probability-modifying watermark with the given $(T, D_0)$, even for an information-theoretic detector. In particular, seeking high robustness (e.g., $\varepsilon \gtrsim 0.3$) together with strong stealth (small $\tau$ for nontrivial $M$) is incompatible at fixed $T$.*

The proof of the above corollary is provided in Appendix D.9. At a high level, the corollary formalizes the design dilemma: one cannot simultaneously achieve large edit tolerance, stringent stealth, and guaranteed verification. Practical watermarking must therefore select an operating point along this trade-off, or adopt hybrid schemes that adapt the information budget to the anticipated edit regime while acknowledging the fundamental *information-theoretic* boundary imposed by $\varepsilon_\beta(T, D_0)$. In the next section, we propose the latter as a Pareto-optimal watermarking scheme in terms of detectability and robustness.

## 4 CONSTRUCTING OPTIMAL WATERMARKS UNDER OUTPUT EDITING

Building upon the robustness-detectability trade-off in Theorem 2, in this section, we develop a principled construction that finds the optimal watermark parameters based on the edit rate of the output channel. The key idea is that no single family is uniformly optimal across noise regimes. Instead, the operating point should be chosen as a function of the edit rate $\hat{\varepsilon}$, the text length $T$, and the per-token information budget available to the detector. We refer to the three watermark families: distribution-preserving (DP), bias-free (BF), and biased (B), as described in Section 2, as well as to their detectability behavior (Section 3.1) and small-signal information expansions (Section 3.2).

### 4.1 A COMPOSITE LOSS FUNCTION

The design objective is to maintain a clear link between stealth and robustness while enabling a clean optimization program. Let $D_0(\theta)$ denote the per-token information (in bits) induced by watermark parameters $\theta$ at zero edits. Under the substitution channel, Theorem 2 states that the usable sequence-level signal at edit rate $\varepsilon$ equals

$$C(\varepsilon; \theta) = T(1 - \varepsilon)^2 D_0(\theta), \qquad D_0(\theta) \geq D_{\text{req}}(\varepsilon, T, \beta) := \frac{\log_2(1/\beta)}{T(1 - \varepsilon)^2}, \tag{5}$$

which yields a sufficient condition for achieving miss probability at most $\beta$ with a level-$\alpha$ Neyman–Pearson detector. On the stealth side, Theorem 1 formalizes detectability in terms of total variation for a single shot. We denote the resulting monotone penalty by $\text{TV}_{\text{pen}}(D_0; M)$ for an outsider that can pool $M$ tokens, and we summarize the corresponding stealth cap as $D_{\text{stealth}}(M, \tau)$ for a target TV budget $\tau$. These ingredients motivate an information-aware loss that enforces robustness while discouraging unnecessary statistical drift:

$$\mathcal{L}(\theta; \hat{\varepsilon}, M, \tau) = \lambda_r \left[\log_2(1/\beta) - T(1 - \hat{\varepsilon})^2 D_0(\theta)\right]_+ + \lambda_q \text{TV}_{\text{pen}}(D_0(\theta); M) + \lambda_a \text{Amp}(\theta). \tag{6}$$

The hinge in the first term compels the design to supply just enough information to meet the detection requirement in equation 5, and no more. The second term translates the single-shot detectability perspective of Theorem 1 into a conservative, sequence-level penalty that grows monotonically with $D_0$. The final term regularizes signal amplitude at the parameter level (e.g., $\sqrt{\hat{\sigma}^2}$ for BF and $|\delta|$ for B), thereby favoring parameterizations that realize the same information with smaller perturbations.

## 4.2 OPTIMAL WATERMARKING THROUGH LOSS MINIMIZATION

Minimizing equation 6 reveals a simple and interpretable structure. Because both the detectability penalty and the amplitude penalty increase with $D_0$, whereas the robustness hinge vanishes once the inequality in equation 5 is met, any minimizer must operate at the smallest feasible per-token information. This observation leads to the target level

$$D^\star := \min\{ D_{\text{stealth}}(M, \tau), D_{\text{BF}}^{\max} + D_{\text{B}}^{\max} \} \quad \text{subject to} \quad D^\star \geq D_{\text{req}}(\varepsilon, T, \beta), \quad (7)$$

where $D_{\text{BF}}^{\max}$ and $D_{\text{B}}^{\max}$ denote the small-signal budgets specified previously. If the inequality in equation 7 cannot be satisfied, then the requested power $1 - \beta$ is unattainable at the given edit rate under the available stealth and budget constraints.

For a feasible $D^\star$, the remaining decision concerns how to realize this information across the two types of probability-modifying watermarks. Since the TV penalty depends only on $D^\star$ (and not on how it is decomposed), the optimal split minimizes the amplitude term. The family-specific mappings between information and parameters yield a closed-form allocation that prioritizes the bias-free family up to its budget and uses the biased family only for any residual information.

**Theorem 3** (Optimal hybrid watermarking). *Fix $T$, $\varepsilon$, a detector level $\alpha$, and a power target $1 - \beta$. Consider a DP watermark with $K$ marked positions and correction radius $t$, and the statistical families BF and B with budgets $D_{\text{BF}}^{\max}$ and $D_{\text{B}}^{\max}$ defined earlier. For the loss in equation 6, an optimal strategy $\mathcal{W}^\star(\varepsilon)$ is:*

1. ***DP region (perfect stealth).*** *If the verifier succeeds with probability at least $1 - \beta$ under edits (equivalently, if $X \sim \text{Binomial}(K, 1 - \varepsilon)$ obeys $\Pr[X < K - t] \leq \beta$ as stated once in the prior section), then DP achieves the target power with $\text{TV} = 0$ and thus minimizes equation 6.*

2. ***Statistical region (information targeting).*** *Otherwise, choose the target information $D^\star$ via equation 7. If $D^\star < D_{\text{req}}(\varepsilon, T, \beta)$, then no watermark can meet the power target at this edit rate.*

3. ***Allocation and parameters.*** *Among all decompositions $D^\star = D_0^{\text{BF}} + D_0^{\text{B}}$ that respect the budgets, the amplitude-minimizing split and corresponding parameter read-off are*

$$D_0^{\text{BF}\star} = \min\{D^\star, D_{\text{BF}}^{\max}\}, \qquad \hat{\sigma}^{2\star} = \text{BF map applied to } D_0^{\text{BF}\star}, \quad (8)$$

$$D_0^{\text{B}\star} = D^\star - D_0^{\text{BF}\star}, \qquad \delta^\star = \text{B map applied to } D_0^{\text{B}\star}, \qquad \gamma^\star = \tfrac{1}{2}, \quad (9)$$

*where the "BF/B map applied to $D_0$" refers to $D_0^{(biased)}$ or $D_0^{(bias\text{-}free)}$ (Section 3.2). In particular, if $D_{\text{req}}(\varepsilon, T, \beta) \leq D_{\text{BF}}^{\max}$, then the optimizer selects a pure BF design; otherwise, BF is saturated and the remainder is realized with B.*

The proof of the above theorem is provided in Appendix E.

> **Interpretation of Theorem 3**
>
> 1. **No universal optimum.** There is no single watermarking scheme that is best in all situations. When the distribution-preserving verifier succeeds, it should be used because it achieves perfect stealth. Otherwise, a statistical scheme should be selected and tuned to the smallest signal level that still guarantees the target detection power.
> 2. **Preference for bias-free information.** Among statistical options, the bias-free family is favored first because it achieves the same detection capability with a smaller parameter change. Only when this budget is exhausted should the biased family be used to supply any remaining information.
> 3. **Limits of feasibility.** If the information required for the desired reliability exceeds what is permitted by stealth constraints and family budgets, then reliable detection cannot be achieved. This identifies a true impossibility region rather than a shortcoming of the detector.

In summary, the composite loss equation 6 combines the detectability perspective of Theorem 1 with the robustness requirement of Theorem 2 into a single optimization framework. The hinge enforces the minimal information level needed for the desired power, the TV penalty internalizes conservative single-shot detectability into sequence-level design, and the amplitude regularizer privileges parameter-efficient realizations of a fixed information budget. Next, we compare the detectability vs. robustness of our hybrid watermark with other schemes through paraphrasing attacks on LLMs, compare it with the existing watermarking schemes.

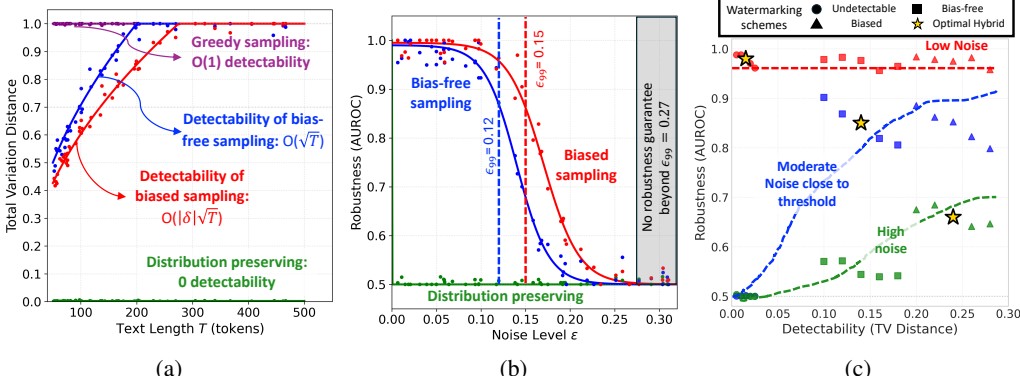

(a)                 (b)                 (c)

Figure 2: Empirical validation showing: (a) dependence of total variation (TV) on sampling rule and sequence length, (b) detection AUROC versus edit noise in generated text, and (c) trade-off between attack resistance and detectability across low, moderate, and high noise regimes. The hybrid scheme aligns with the Pareto optimal boundary in every regime.

## 5 EXPERIMENTAL EVALUATION

This section empirically validates our information-theoretic framework using three families of watermarking schemes, evaluating both detectability and robustness against paraphrasing attacks. All the relevant code for replicating the experiments is available at `https://anonymous.4open.science/r/Catch-22-Pareto-Frontier-Watermark-in-LLMs-040B`. The repository will be made publicly available, ensuring replicability and full functionality, along with detailed user manuals, once the paper is accepted.

---

**Experimental Setup**

***Dataset and Models.*** For our non-watermarked baseline, we generate text using 500 prompts randomly sampled from the LFQA dataset, which contains long-form questions from Reddit spanning six domains (July to December 2021) Krishna et al. (2023). We conduct our analysis using open-source Llama-2 7B Touvron et al. (2023) and Mistral 7B Jiang et al. (2023) models on a single NVIDIA H100 GPU, generating outputs ranging from 100 to 1000 tokens.

***Watermarking Schemes.*** We evaluate three categories of watermarking: biased sampling (KGW in Kirchenbauer et al. (2023) and Unigram in Zhao et al. (2023)), bias-free sampling (DiPMark in Wu et al. (2024) and HCW in Hu et al. (2024)), and distribution-preserving sampling (CGZ scheme in Christ et al. (2024)). Additionally, we test our optimal hybrid sampling scheme derived from Theorem 3, which dynamically adapts to observed edit noise levels.

***Paraphrasing Attacks.*** We employ two attack methods: the DIPPER paraphraser in Krishna et al. (2023) with variable token edit rates, and the OPT-2.7B model Zhang et al. (2022) prompted with "Rewrite the following paragraph:", which produces an average edit rate of 15%. In addition, we evaluate robustness under synonym-substitution, adversarial watermark-removal, and back-translation (English-French-English) paraphrasing attacks using GPT 3.5 API prompting Liu et al. (2025).

---

**Appendix G, Table 1 and Table 2 provide a comprehensive comparison of robustness versus detectability**, demonstrating that our hybrid scheme achieves Pareto optimality across different noise regimes. While this tabular analysis offers a model-agnostic view of the detectability-robustness space, we now present a detailed analysis focusing on Llama 7B outputs subjected to DIPPER paraphrasing at varying edit levels, which is empirically validated to follow the i.i.d. edit channel assumption (Appendix G.1).

### 5.1 TRADE-OFFS BETWEEN ATTACK RESISTANCE AND DETECTABILITY

**Figure 2(a)** demonstrates how total variation (TV) distance scales with output token length, confirming the predictions of Theorem 1. Greedy decoding exhibits $O(1)$ TV scaling with sequence length $T$, empirically approaching the upper bound of 1, reflecting the length-independent distributional shift induced by deterministic selection. Biased sampling shows TV growing as $|\delta|\sqrt{T}$, where $\delta$ represents bias magnitude. Bias-free sampling displays similar $\sqrt{T}$ growth but with a different constant factor determined by variance modulation rather than mean shifts. Distribution-preserving

sampling maintains near-zero TV across all sequence lengths, remaining effectively undetectable. These results validate and formalize previous empirical observations in Kirchenbauer et al. (2024) regarding the improved detectability that comes with increased token length.

**Figure 2(b)** illustrates detection performance (AUROC) under varying paraphrasing intensities. The curves exhibit a characteristic knee point corresponding to the threshold where the Neyman-Pearson test maintains 99% detection power. The critical noise thresholds $\varepsilon_{99} \approx 0.15$ for biased sampling and 0.12 for bias-free sampling aligns with $T(1 - \varepsilon)^2 D_0 \geq \log_2(1/\beta)$ in Theorem 2, with $\beta = 0.01$ and initial information budget $TD_0 = 10$ bits. Below this threshold, both schemes maintain high AUROC, though their degradation patterns differ: bias-free (variance-based) encoding exhibits a sharp decline beyond the knee, while biased (mean-shift) encoding degrades more gradually. Distribution-preserving sampling proves fragile to edits, as its decoding depends on intact high-entropy substrings, rendering it undetectable even under minimal paraphrasing.

**Figure 2(c)** synthesizes the complete landscape by plotting AUROC against TV distance across three noise regimes, each sampled at five edit rates: low noise (red, $\varepsilon_{99} < 0.005$), moderate noise (blue, $\varepsilon_{99} \approx 0.15$), and high noise (green, $\varepsilon_{99} > 0.15$). No single scheme achieves both undetectability and attack resistance across all regimes. However, the optimal hybrid from Theorem 3 consistently traces the Pareto frontier, crucially outperforming the best existing scheme within each regime. This adaptive approach emerges as the most reliable and stealthy watermarking solution across all noise conditions. By adjusting watermark parameters based on observed edit rates, the hybrid maintains superiority in the AUROC-TV plane, with operating points aligning precisely with theoretical predictions and surpassing any fixed scheme across the entire spectrum of edit intensities. Our framework, therefore, serves as a practical guide for constructing watermarks on the Pareto-optimal frontier of the AUROC-TV plane, as discussed next.

## 6 DISCUSSION AND CONCLUSION

This work establishes an information-theoretic framework that characterizes the fundamental trade-off between detectability and edit tolerance in language model watermarks. Biased and bias-free sampling schemes accumulate detectable statistical signals across tokens, enabling reliable recovery under text edits while remaining statistically distinguishable. Conversely, distribution-preserving techniques achieve provable undetectability but fail under minimal editing because they rely on intact, high-entropy patterns. We frame watermark detection as one-bit extraction over a noisy channel, proving that redundancy enhances robustness at the cost of statistical visibility. This fundamental trade-off is inherent and cannot be circumvented: any scheme seeking both properties must necessarily compromise on at least one. Building on these insights, we develop a hybrid watermarking scheme that operates at the Pareto-optimal boundary and consistently outperforms existing approaches across all noise regimes. This information-theoretic perspective moves beyond the adversarial arms race of watermarking attacks by providing principled guidance for practical system design. Rather than pursuing simultaneous robustness and undetectability, designers can strategically select schemes based on application requirements: deploying undetectable watermarks in privacy-sensitive contexts and robust watermarks in open-access applications, recognizing that no single approach can satisfy both objectives. Furthermore, our results highlight the distinct roles of information theory and cryptography in LLM watermark analysis: we provide information-theoretic results on detectability and robustness, establish fundamental bounds on the total statistical signal any watermark can encode, while computational hardness determines which practical detectors (e.g., key holders versus outsiders) can feasibly exploit that signal. Importantly, computational assumptions affect which points on this information-theoretic trade-off are algorithmically attainable, but do not alter the frontier itself.

**Extensions and Implications.** While our analysis focuses on inference-time watermarking, it provides insights for training-time watermarks embedded in model parameters (Appendix F). Since model architecture has a minimal impact on inference-time performance, we believe training-time schemes can potentially exhibit different trade-offs that are worthy of future investigation Block et al. (2025). Additionally, undetectable watermarking introduces security concerns: the surplus entropy concealing one-bit signals can encode multi-bit payloads, creating covert channels within LLM outputs as proposed in Gaure et al. (2024); Zamir (2024), and we also analyze it further in Appendix H. All in all, our work identifies Pareto-optimal LLM watermarking solutions and establishes theoretical foundations for practical watermark designs, even when the conflicting goals of high robustness and undetectability cannot be simultaneously achieved.

IMPACT STATEMENT

**Practical Deployment.** Our work reveals that no watermarking scheme can simultaneously achieve high robustness, strong undetectability, and reliable detection. For controlled environments (enterprise, academic), we recommend undetectable watermarks paired with access controls and key rotation. For public deployments, use detectable watermarks with documented failure modes and regular auditing. System operators should monitor real-world editing patterns and adjust watermarking strategies based on our theoretical thresholds: use distribution-preserving methods for minimal editing ($\varepsilon < 0.05$), variance-based encoding near the critical threshold ($\varepsilon \approx 0.15$), and bias-based methods under heavy editing ($\varepsilon > 0.3$).

**Future Work.** While our analysis focuses on inference-time watermarking, several directions merit investigation. First, training-time watermarks (Gu et al. (2024); Block et al. (2025)) that embed signals directly into model weights could enable watermarking for open-source models where users control decoding. Key challenges include resistance to fine-tuning attacks and minimizing distillation-induced quality loss. Second, semantic watermarking operating in embedding space may offer orthogonal robustness properties worth characterizing theoretically, such as in images and multimodal data. Finally, the covert channel vulnerability in watermarks (Appendix H) requires further investigation, including the development of detection methods for unauthorized payload embedding.

**Limitations.** While our framework establishes fundamental bounds for LLM watermarking, it assumes independent token-level editing that sophisticated paraphrasing attacks may violate through correlated changes. However, since the attack on LLMs is an active research area, such paraphrasing attacks are crucial for vulnerability assessment of LLMs, which in turn enhances their security. Additionally, although our hybrid scheme achieves Pareto optimality across noise regimes, it requires accurate estimation of editing levels, which remains challenging in adversarial settings and can be considered as a future direction of research. Nevertheless, our theoretical insights provide essential guidance for practical deployments.

ETHICAL CONSIDERATIONS

We conduct all our experiments on open-source large language models with known vulnerabilities, such as loss of watermarking robustness due to LLM output editing. This research is essential from the perspective of LLM vulnerability assessment, given that these systems are increasingly becoming part of our daily lives. We believe that our theoretical framework and results will assist the research community in designing improved LLM watermarking schemes.

REPRODUCIBILITY

We are firm believers and remain committed to open-source research. The relevant code and its corresponding instructions is available at `https://anonymous.4open.science/r/Catch-22-Pareto-Frontier-Watermark-in-LLMs-040B` for replication of results. This includes models, prompts, watermarking schemes, and paraphrasing attacks to support comparative studies and encourage the community to adopt joint reporting of detectability and robustness of new LLM watermarking schemes.

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

## A  EXTENDED REVIEW OF LLM WATERMARKING LITERATURE

We provide here a comprehensive technical analysis of existing watermarking schemes for large language models, extending the overview presented in Section 2. This review organizes prior work according to its fundamental design principles and analyzes its theoretical guarantees, practical limitations, and empirical vulnerabilities.

### A.1  PROBABILITY-MODIFYING WATERMARKS

Probability-modifying watermarks alter token selection probabilities during generation to embed detectable signals. This broad category encompasses all schemes that deviate from the original model's distribution, whether through direct biasing or more subtle statistical modifications.

**Biased Sampling Schemes** The seminal work of Kirchenbauer et al. (2023) introduced soft watermarking through dynamic vocabulary partitioning. Their scheme computes a cryptographic hash function based on the preceding $k - 1$ tokens to partition the vocabulary at each generation step. Specifically, for position $t$, the vocabulary $\mathcal{V}$ is divided into a green list $G_t$ containing a fraction $\gamma$ of tokens and a red list $R_t = \mathcal{V} \setminus G_t$. The watermark manifests through logit modification:

$$\hat{\ell}_t[v] = \ell_t[v] + \delta \cdot \mathbf{1}[v \in G_t] \tag{10}$$

where $\delta$ controls watermark strength. This induces an exponential tilt in the sampling distribution, increasing the probability of green tokens by approximately a factor $e^{\delta}$. Note that in this work, we use $k - 1 = 1$ preceding tokens when referring to the KGW scheme.

A significant advancement came from Zhao et al. (2023), who demonstrated that fixing the green-red partition across all positions yields superior robustness properties. Their UNIGRAM-WATERMARK scheme establishes tight bounds on output quality degradation through Rényi divergence analysis and proves quantitative robustness guarantees, tolerating $O(n)$ adversarial edits for sequences of length $n$.

***Shortcoming.*** *Although robust against moderate edits, both KGW and UNIGRAM accumulate outsider evidence at rate $O(|\delta|\sqrt{T})$. Even modest biases create detectable frequency shifts that can be flagged by chi-square tests or amplified by adversarial prompting. Thus, robustness is achieved only at the cost of increased detectability.*

**Bias-Free Sampling Schemes** While still modifying probabilities, bias-free approaches attempt to preserve expected token distributions through sophisticated reweighting mechanisms. The framework introduced by Hu et al. (2024) employs context-dependent reweighting functions satisfying:

$$\mathbb{E}_E[R_E(p_t)] = p_t \tag{11}$$

where $E$ is a watermark code derived from context and secret key. This ensures the expected distribution over random keys matches the original model's output, though individual samples are drawn from modified distributions.

Similarly, Wu et al. (2024) achieves expectation preservation through vocabulary permutations, while Kuditipudi et al. (2024) employs inverse transform sampling with controlled randomness. All these schemes modify the sampling distribution $q_t \neq p_t$ at each step but maintain $\mathbb{E}[q_t] = p_t$ through careful construction.

***Shortcoming.*** *Despite unbiasedness in expectation, these methods inevitably introduce higher-order variance signatures that grow as $O(\sqrt{T})$. Such distortions are detectable by second-moment tests Gloaguen et al. (2025). To sustain resilience under edits, the injected watermark signal must be amplified, which further undermines stealth. Hence, they cannot simultaneously ensure strong robustness and low detectability.*

### A.2 Distribution-Preserving Watermarks

The most recent class of watermarking schemes achieves provable undetectability by maintaining exact output distributions while controlling only the source of randomness.

**Cryptographic Undetectability** The breakthrough work of Christ et al. (2024) demonstrated that replacing true randomness with pseudorandom functions achieves perfect statistical indistinguishability. Their construction maintains $q_t \equiv p_t$ for all positions while making generation deterministic for key holders. Detection requires exact reproduction of PRF outputs, creating a cryptographic verification mechanism rather than statistical hypothesis testing.

Extensions by Zamir (2024) show that arbitrary payloads can be embedded within this framework by incorporating messages into PRF seeds, enabling covert communication channels with capacity $\Theta(L)$ bits for text length $L$.

***Shortcoming.*** *While perfectly undetectable in theory ($q_t \equiv p_t$), these schemes collapse under even light paraphrasing. Verification depends on intact PRF alignment, making edit resilience negligible. Attempts to strengthen robustness reintroduce detectable statistical drift, negating their undetectability advantage.*

### A.3 Detection Methods and Vulnerabilities

The arms race between watermarking and detection has produced increasingly sophisticated statistical tests that expose subtle artifacts across all scheme categories.

**Statistical Detection Methods** For probability-modifying watermarks, Sadasivan et al. (2023) demonstrates that simple frequency analysis suffices for detection. Their chi-squared test compares

observed versus expected token frequencies:

$$\chi^2 = \sum_{v \in \mathcal{V}} \frac{(f_v^{\text{obs}} - f_v^{\text{exp}})^2}{f_v^{\text{exp}}} \tag{12}$$

where $f_v$ denotes the frequency of token $v$. This test achieves high power against biased watermarks with modest sample sizes.

For expectation-preserving schemes, Gloaguen et al. (2025) develops second-moment tests that detect variance anomalies. Their test statistic aggregates squared deviations from expected variance:

$$T = \sum_{t=1}^{n} \left( \|\hat{p}_t\|_2^2 - \mathbb{E}[\|p_t\|_2^2] \right) \tag{13}$$

This approach succeeds because reweighting necessarily introduces variance distortions even when preserving expectations.

**Adaptive Attacks** Beyond passive detection, Liu et al. (2025) demonstrates active attacks using adversarial prompting. By crafting prompts that amplify watermark biases, they force watermarked models to produce highly distinguishable outputs. Their optimization finds prompts maximizing:

$$\Delta(x) = \mathbb{E}_{y \sim \hat{p}(\cdot|x)}[\text{score}(y)] - \mathbb{E}_{y \sim p(\cdot|x)}[\text{score}(y)] \tag{14}$$

where score measures watermark strength. Such targeted attacks reduce required sample sizes by orders of magnitude.

***Note.*** *These detection methods and attacks highlight a structural vulnerability: biased schemes are easily exposed via frequency analysis, bias-free schemes via variance anomalies, and both via adversarial prompting. Thus, neither family achieves low detectability in practice.*

### A.4 ALTERNATIVE APPROACHES TO WATERMARK ROBUSTNESS ANALYSIS

While our main analysis models text perturbations as a noisy channel, several alternative mathematical frameworks have been developed in the watermarking literature to analyze robustness. We review two prominent approaches here.

**Direct Statistical Analysis of Detection Scores**

Zhao et al. (2023) analyzed robustness by directly tracking how the watermark detection statistic degrades under edit operations. Their approach does not invoke channel capacity or information-theoretic arguments, but instead provides explicit bounds on the z-score used for detection.

For their UNIGRAM-WATERMARK scheme, they prove that if text $y$ is watermarked and an adversary produces modified text $u$ with edit distance $\eta = \text{ED}(y, u) < n$, then the detection z-scores satisfy:

$$z_u \geq z_y - \max \left\{ \frac{(1 + \gamma/2)\eta}{\sqrt{n}}, \frac{(1 - \gamma/2)\eta}{\sqrt{n - \eta}} \right\} \tag{15}$$

where $\gamma$ is the green list ratio parameter, the proof technique involves analyzing how each edit operation affects the count of green tokens, using a Taylor expansion argument to bound the worst-case degradation. This approach yields that the watermark can tolerate up to $O(n)$ arbitrary edits for text of length $n$ when the watermark strength parameter $\delta$ is constant.

The key advantage of this direct approach is its simplicity and explicitness; it provides concrete formulas for how detection degrades with edits. However, it is specific to their particular watermarking scheme and does not readily generalize to other watermarking methods.

***Shortcoming.*** *Although offering concrete edit tolerance formulas, this method is tied to UNIGRAM and does not generalize. Moreover, it provides no guarantees about detectability, limiting its applicability for designing low-detectability watermarks.*

### A.5 CODING-THEORETIC CONSTRUCTIONS WITH INDEXING

Moitra & Golowich (2024) adopts a fundamentally different approach by constructing watermarks using error-correcting codes. Their central idea is the use of *indexing pseudorandom codes*, which

provide edit-distance guarantees that tolerate a *constant fraction* of adversarial insertions, deletions, and substitutions in any sufficiently high-entropy substring. The construction starts with a binary pseudorandom code (PRC) of block length $n$, implying that an $n$-bit codeword is already resilient to substitutions. This codeword is then transformed into an "indexing PRC" by encoding each of its $n$ positions using symbols drawn from a much larger alphabet. The resulting alphabet must scale polynomially in $n$, with a polynomial degree determined by an entropy-rate parameter that governs its substring-robustness guarantee. Each symbol in this extended alphabet is associated with an index position via a (keyed) random hash function, with multiple symbols mapping to each index to supply the redundancy needed to handle insertions and deletions. The redundancy parameter is crucial for handling insertions and deletions: when an adversary introduces edits, the multiset of indices changes, but the redundancy ensures that, with high probability, the Hamming distance between the original and modified binary strings remains bounded.

Their analysis proves that this watermarking scheme achieves *substring robustness*: any sufficiently high-entropy substring of the watermarked text remains detectable even after a constant fraction of edits. More precisely, let $\Sigma$ denote the alphabet and let $\alpha$ be the model's *conditional entropy rate*, normalized by $\ln|\Sigma|$ so that $\alpha \in (0,1)$ measures how much randomness per token a substring retains relative to the maximum possible. Under this condition, their scheme is robust to a fraction $p = \Theta(\alpha^2)$ of arbitrary edit operations (insertions, deletions, and substitutions). This robustness requires two further conditions. First, the underlying binary pseudorandom code (PRC) of block length $n(\lambda)$ (where $n(\lambda)$ is the length of the PRC codeword and is polynomial in the security parameter $\lambda$) must itself be able to correct a

$$\big(1 - \Theta(\alpha) + O(p/\alpha)\big) \tag{16}$$

fraction of errors. Second, the alphabet used by the indexing construction must satisfy the lower bound

$$|\Sigma(\lambda)| \;\geq\; n(\lambda)^{C_2 \frac{1}{\alpha} \log \frac{1}{\alpha}}, \tag{17}$$

where $C_2 > 0$ is an absolute constant. Thus, although the alphabet size $|\Sigma(\lambda)|$ is polynomial in the security parameter $\lambda$ for any fixed $\alpha$, the *degree* of this polynomial grows like $\Theta\big(\frac{1}{\alpha} \log \frac{1}{\alpha}\big)$, which increases rapidly as $\alpha$ decreases.

***Shortcoming and relation to our impossibility results.***

While $|\Sigma(\lambda)|$ is polynomial in $\lambda$ for fixed $\alpha$, the polynomial degree grows as $\Theta\big(\frac{1}{\alpha} \log \frac{1}{\alpha}\big)$, meaning the required alphabet size increases exponentially in $1/\alpha$, the inverse of the entropy-rate parameter governing substring robustness. For realistic natural-language entropy levels (moderate $\alpha$) and constant-fraction edit tolerance $p = \Theta(\alpha^2)$, these bounds imply alphabet sizes far larger than the fixed vocabularies of approximately 30k to 100k tokens used by practical LLMs. Consequently, the construction operates in a large-alphabet setting rather than the fixed-vocabulary setting considered in our work.

Conceptually, Moitra & Golowich (2024) provides an *achievability* result. Given a tunable, sufficiently large alphabet and standard cryptographic assumptions, one can design watermarks that are both computationally undetectable and resilient to a constant fraction of edits on high-entropy substrings. By contrast, our Theorems 1 and 2 are *information-theoretic impossibility* statements for fixed-vocabulary LLMs whose watermarked outputs must remain close to a pre-trained distribution: in this regime, any scheme that carries enough information to withstand a constant fraction of edits necessarily induces nontrivial detectability. The two results therefore address different parameter regimes (large adjustable alphabets versus small fixed vocabularies) and are complementary rather than contradictory.

## B  NOTATION AND VARIABLES

### NOTATION CONVENTIONS

- **Vectors:** $\mathcal{V}^T$ denotes $T$-length token sequences from vocabulary $\mathcal{V}$.
- **Subscripts:** $t$ indexes token position (1 to $T$).
- **Superscripts:** On $Q$ indicate sampling method; asterisk ($*$) denotes optimal values.
- **Context:** Conditionals like $p_t(\cdot)$ depend on $\mathbf{y}_{<t}$ and prompt $x$.
- **Overloading note:** $M$ denotes outsider pooled tokens in §4.1; in Appendix H it denotes the number of covert messages.

## CORE VARIABLES AND DISTRIBUTIONS

| Symbol | Type/Dim | Description | Sections |
|---|---|---|---|
| $L, T$ | Scalar | Text length (number of tokens) | §3, §4 |
| $\mathcal{V}, \Sigma$ | Set | Token vocabulary | §3 |
| $x$ | Vector | Initial prompt | §3 |
| $\mathbf{y} = (y_1, \ldots, y_T)$ | $\mathcal{V}^T$ | Generated token sequence | §3 |
| $\mathbf{y}_{<t}$ | $\mathcal{V}^{t-1}$ | Tokens before position $t$ | §3 |
| $\tilde{\mathbf{y}}$ | $\mathcal{V}^T$ | Edited/noisy text | §3 |
| $\mathbf{y}^\star$ | $\mathcal{V}^T$ | Deterministic greedy path | §3 |
| $p_t(\cdot), p_\theta(\cdot|\cdot)$ | Function | Baseline LLM conditional probabilities | §3 |
| $q_t(\cdot)$ | Function | Watermarked conditional probabilities | §3 |
| $P^s$ | Distribution | Baseline sampling distribution over sequences | §3 |
| $Q^{\mathcal{W}}$ | Distribution | Sequence distribution for scheme $\mathcal{W}$ | §3 |
| $Q^{\text{greedy}}$ | Distribution | Greedy sampling distribution | §3 |
| $Q^{\text{bias}_\delta}$ | Distribution | Biased (tilted) sampling with parameter $\delta$ | §3 |
| $Q^{\text{bf}}_E$ | Distribution | Bias-free sampling with key/code $E$ | §3 |
| $Q^{\text{prf}}$ | Distribution | PRF-based distribution-preserving sampling | §3 |
| $U_t$ | $[0,1]$ | Uniform random variable used for sampling | §3 |
| $U$ | $\Delta(\Sigma)$ | Uniform distribution on $\Sigma$ | App. D |
| $T_\varepsilon(P)$ | Operator | Edit channel: $(1-\varepsilon)P + \varepsilon U$ | App. D |
| $p_{t,\varepsilon}, q_{t,\varepsilon}$ | Function | Edited conditionals: $T_\varepsilon(p_t), T_\varepsilon(q_t)$ | App. D |

## WATERMARKING PARAMETERS

| Symbol | Type/Dim | Description | Sections |
|---|---|---|---|
| $\delta, \delta^\star$ | Scalar | Bias strength (optimal value $\delta^\star$) | §3, §4 |
| $G_t \subset \mathcal{V}$ | Set | Keyed green token set at step $t$ | §3 |
| $g_t = p_t(G_t)$ | $[0,1]$ | Baseline green mass at step $t$ | §3, App. C |
| $\gamma, \gamma^\star$ | $[0,1]$ | Typical/target green mass (often $\gamma^\star = \frac{1}{2}$) | §3, §4 |
| $k$ | Key | Secret cryptographic key | §3 |
| $E, E_t$ | Code | Keyed code or permutation for bias-free schemes | §3 |
| $R_E$ | Function | Reweighting operator with $\mathbb{E}_E[R_E(p_t)] = p_t$ | §3 |
| $\sigma^2(v), \hat{\sigma}^2$ | Scalar | $\sigma^2(v) = \text{Var}_E[R_E(p_t)(v)], \quad \hat{\sigma}^2 = \sum_v p_t(v)\sigma^2(v)$ | §3, App. D |
| $Z_t$ | Scalar | Normalizer for tilted sampling | App. C |
| PRF | Function | Pseudorandom function for RNG replacement | §3 |
| $\mathcal{W}, \mathcal{W}^\star(\varepsilon)$ | Scheme | Watermarking scheme and the optimal hybrid | §4, App. E |
| $K, t$ | Scalars | DP verifier: marked positions $K$ and correction radius $t$ | §4, App. E |

## INFORMATION THEORY AND ROBUSTNESS

| Symbol | Type/Dim | Description | Sections |
|---|---|---|---|
| $D_0, D_\varepsilon$ | Bits/token | Per-token information at 0 edits and at rate $\varepsilon$ | §3, App. D |
| $C(\varepsilon)$ | Bits | Total usable information $\approx T(1-\varepsilon)^2 D_0$ | §3, App. D |
| $\varepsilon_\beta(T, D_0)$ | $[0,1]$ | "Knee": $1 - \sqrt{\log_2(1/\beta)/(TD_0)}$ | App. D |
| $\text{TV}(P, Q)$ | $[0,1]$ | Total variation distance | §3, App. C |
| $\text{KL}(Q\|P)$ | $[0,\infty)$ | Kullback–Leibler divergence (base 2 in proofs) | §3 |
| $H(\cdot), H_2(\cdot)$ | Function | Entropy, binary entropy | §3 |
| $I(\cdot; \cdot)$ | Bits | Mutual information | App. H |
| $\text{Detect}(s)$ | $[0,1]$ | Distinguishability for sampling rule $s$ | §3 |
| $\varepsilon, \hat{\varepsilon}$ | $[0,1]$ | Edit rate (true and estimated) | §3, §4 |
| $\alpha, \beta$ | $[0,1]$ | Detector level and miss probability (power $= 1 - \beta$) | §4, App. D |
| $D_{\text{req}}(\varepsilon, T, \beta)$ | Bits/token | $\log_2(1/\beta)/T(1-\varepsilon)^2$ | §4.1, App. D |
| $M, \tau$ | Scalar, $[0,1]$ | Outsider pooled tokens $M$ and TV budget $\tau$ | §4.1, App. D |
| $D_{\text{stealth}}(M, \tau)$ | Bits/token | Stealth cap $\frac{2\tau^2}{M \ln 2}$ | §4.1, App. D |
| $z, z_{\text{threshold}}$ | Scalar | Z-score statistic and threshold | App. D |
| $N_{\text{green}}$ | Scalar | Count of green tokens | App. D |
| $\Phi(\cdot), \Phi^{-1}(\cdot)$ | Function | Standard normal CDF and its inverse | §4 |

OPTIMIZATION AND OPERATORS

| Symbol | Type/Dim | Description | Sections |
|---|---|---|---|
| $\mathcal{L}(\theta; \hat{\varepsilon}, M, \tau)$ | Scalar | Composite loss | §4.1 |
| $\theta$ | Variable | Scheme parameters | §4 |
| $\lambda_r, \lambda_q, \lambda_a$ | Scalars | Weights for reliability, stealth penalty, amplitude | §4.1 |
| $D^\star$ | Bits/token | Target per-token information after constraints | §4.2, App. E |
| $D_{\mathrm{BF}}^{\max}, D_{\mathrm{B}}^{\max}$ | Bits/token | Available budgets for BF and B families | §4.2, App. E |
| $\mathrm{TV}_{\mathrm{pen}}(D_0; M)$ | Scalar | Monotone detectability penalty used in the loss | §4.1 |
| $\mathrm{Amp}(\theta)$ | Scalar | Amplitude regularizer (e.g., $\sqrt{\hat{\sigma}^2}$ or $|\delta|$) | §4.1 |
| $\mathbb{E}[\cdot], \mathrm{Var}[\cdot]$ | Operator | Expectation, variance | §3 |
| $\mathbf{1}[\cdot]$ | Function | Indicator | §3 |
| $\arg\max, \sup$ | Operator | Maximizer, supremum | §3 |
| $\ln, \log, \log_2$ | Function | Natural log, log, base-2 log | §3 |
| $O(\cdot), o(\cdot), \Theta(\cdot), \omega(\cdot), \Omega(\cdot)$ | Notation | Asymptotic notation | §3 |
| $\approx$ | Operator | Approximately equal | App. D |
| $\infty$ | Symbol | Infinity | App. E |

ADDITIONAL SYMBOLS USED IN APPENDIX H

| Symbol | Type/Dim | Description | Sections |
|---|---|---|---|
| $W$ | RV | Message index (uniform over $\{1, \ldots, M\}$) | App. H |
| $Q_w, Q$ | Distribution | Distribution for message $w$ and outsider mixture $\frac{1}{M}\sum_w Q_w$ | App. H |
| $C_\star$ | Scalar | Mixture divergence budget $D(Q\|P) \leq C_\star$ | App. H |
| $\theta$ (Appendix) | $[0, 1]$ | Activity probability $c/\sqrt{L}$ in square-root law construction | App. H |
| $c, \kappa$ | Scalar | Constants in square-root law achievability | App. H |

## C   PROOF OF THEOREM 1

This appendix proves the bounds in Theorem 1 and provides a detailed explanation of each step. The statement in the main paper is for computing the total variation distance from a *single-shot* or a single generated text from LLM. Multi-shot black-box detection over multiple LLM queries, key-averaged ($n$-shot) properties for bias-free watermarks, as well as the computational undetectability guarantees for PRF-seeded schemes, are also described later in this proof.

We measure distributional separation with the total variation distance

$$\mathrm{TV}(P, Q) = \frac{1}{2}\sum_{\mathbf{y}} |P(\mathbf{y}) - Q(\mathbf{y})|, \tag{18}$$

and we use the Kullback–Leibler (KL) divergence

$$\mathrm{KL}(Q\|P) = \mathbb{E}_{\mathbf{y} \sim Q}\left[\log \frac{Q(\mathbf{y})}{P(\mathbf{y})}\right]. \tag{19}$$

Pinsker's inequality connects these two quantities and will be invoked repeatedly:

$$\mathrm{TV}(P, Q) \leq \sqrt{\frac{1}{2}\mathrm{KL}(Q\|P)}. \tag{20}$$

For autoregressive distributions that factorize across positions, the KL chain rule expresses the sequence level divergence as a sum of conditional one-step divergences:

$$\mathrm{KL}(Q\|P) = \sum_{t=1}^{T} \mathbb{E}_{y_{<t} \sim Q}\Big[\mathrm{KL}\big(q_t(\cdot \mid y_{<t}) \,\|\, p_t(\cdot \mid y_{<t})\big)\Big]. \tag{21}$$

### C.1   GREEDY SAMPLING

Let $Q^{\mathrm{greedy}}$ be the degenerate distribution that puts unit mass on the unique greedy path $\mathbf{y}^\star = (y_1^\star, \ldots, y_T^\star)$, with $y_t^\star = \arg\max_v p_t(v \mid y_{<t}^\star)$. Since $Q^{\mathrm{greedy}}(\mathbf{y}^\star) = 1$ and $Q^{\mathrm{greedy}}(\mathbf{y}) = 0$ for all

$\mathbf{y} \neq \mathbf{y}^\star$, the total variation distance expands as

$$\mathrm{TV}(P^s, Q^{\mathrm{greedy}}) = \frac{1}{2} \sum_{\mathbf{y}} \left| P^s(\mathbf{y}) - Q^{\mathrm{greedy}}(\mathbf{y}) \right| \tag{22}$$

$$= \frac{1}{2} \left( \left| P^s(\mathbf{y}^\star) - 1 \right| + \sum_{\mathbf{y} \neq \mathbf{y}^\star} \left| P^s(\mathbf{y}) - 0 \right| \right). \tag{23}$$

The absolute value in the first term simplifies to $1 - P^s(\mathbf{y}^\star)$ because probabilities are at most one. The sum over the remaining sequences simplifies to $\sum_{\mathbf{y} \neq \mathbf{y}^\star} P^s(\mathbf{y}) = 1 - P^s(\mathbf{y}^\star)$ because the probabilities must sum to one. Therefore

$$\mathrm{TV}(P^s, Q^{\mathrm{greedy}}) = 1 - P^s(\mathbf{y}^\star). \tag{24}$$

## C.2 BIASED SAMPLING (EXPONENTIAL TILT, SOFT GREEN LIST)

At position $t$, let $G_t \subseteq \mathcal{V}$ denote the keyed green set and define its baseline probability mass $g_t := p_t(G_t) = \sum_{v \in G_t} p_t(v)$. The biased sampler applies an exponential tilt to tokens in $G_t$:

$$q_t(v) = \frac{p_t(v) \exp\{\delta \, \mathbf{1}[v \in G_t]\}}{Z_t}, \qquad Z_t = \sum_v p_t(v) \exp\{\delta \, \mathbf{1}[v \in G_t]\}. \tag{25}$$

The normalizer follows from splitting the sum into green and non-green tokens. The mass of the complement of $G_t$ is $1 - g_t$ and the mass of $G_t$ is $g_t$, hence

$$Z_t = (1 - g_t) + g_t e^\delta = 1 + g_t (e^\delta - 1). \tag{26}$$

The one-step KL divergence equals

$$\mathrm{KL}(q_t \| p_t) = \sum_v q_t(v) \log \frac{q_t(v)}{p_t(v)} \tag{27}$$

$$= \sum_v q_t(v) \left( \delta \, \mathbf{1}[v \in G_t] - \log Z_t \right) \tag{28}$$

$$= \delta \, q_t(G_t) - \log\big(1 + g_t(e^\delta - 1)\big). \tag{29}$$

The second line uses the explicit tilted form of $q_t$, which cancels the factor $p_t(v)$ and yields a term that depends only on $Z_t$. The last line replaces the indicator sum by the mass $q_t(G_t)$.

A small parameter expansion provides an explicit constant. Using $e^\delta = 1 + \delta + \frac{\delta^2}{2} + O(\delta^3)$ and $\log(1 + x) = x - \frac{x^2}{2} + O(x^3)$, the logarithm of the normalizer expands as

$$\log Z_t = \log\left(1 + g_t\left(\delta + \tfrac{\delta^2}{2} + O(\delta^3)\right)\right) \tag{30}$$

$$= g_t \delta + \frac{g_t(1 - g_t)}{2} \delta^2 + O(\delta^3). \tag{31}$$

In the previous step, since $g_t$ is a fixed probability mass, it is absorbed in the last term $O(\delta^3)$. The mass of the green set under $q_t$ is a ratio of two series. Using the series for $e^\delta$ and the identity $(1 + u)^{-1} = 1 - u + O(u^2)$ with $u = g_t(\delta + \frac{\delta^2}{2}) + O(\delta^3)$ gives

$$q_t(G_t) = \frac{g_t e^\delta}{1 + g_t(e^\delta - 1)} = g_t + g_t(1 - g_t)\delta + O(\delta^2). \tag{32}$$

Substituting both expansions into the one-step KL cancels the linear terms and leaves the quadratic coefficient

$$\mathrm{KL}(q_t \| p_t) = \frac{g_t(1 - g_t)}{2} \delta^2 + O(\delta^3). \tag{33}$$

At the sequence level, the chain rule equation 21 expresses the KL divergence as a sum of the conditional one-step terms under the biased process. Keeping the leading order in $\delta$ yields

$$\mathrm{KL}\big(Q^{\mathrm{bias}_\delta} \| P^s\big) = \frac{\delta^2}{2} \sum_{t=1}^T \mathbb{E}_{y_{<t} \sim Q^{\mathrm{bias}_\delta}}\big[g_t(1 - g_t)\big] + O(T|\delta|^3). \tag{34}$$

Finally, Pinsker's inequality converts this to a total variation bound,

$$\mathrm{TV}(P^s, Q^{\mathrm{bias}_\delta}) \le |\delta| \sqrt{\frac{1}{4} \sum_{t=1}^T \mathbb{E}[g_t(1 - g_t)]} + O(\sqrt{T}\,|\delta|^{3/2}), \tag{35}$$

which exhibits the $O(|\delta|\sqrt{T})$ scaling with an explicit leading constant.

## C.3 Bias-free sampling (unbiased reweighting)

In the bias free setting a keyed operator $R_E : \Delta(\mathcal{V}) \to \Delta(\mathcal{V})$ reweights the baseline, and unbiasedness requires $\mathbb{E}_E[R_E(p_t)] = p_t$ for every step. For a fixed key $E$ one can write

$$R_E(p_t)(v) = p_t(v) + \epsilon_t^{(E)}(v), \qquad \sum_v \epsilon_t^{(E)}(v) = 0, \tag{36}$$

where the sum constraint enforces normalization. The one-step KL divergence admits a Taylor expansion around $p_t$:

$$\mathrm{KL}(R_E(p_t) \,\|\, p_t) = \sum_v \left( p_t(v) + \epsilon_t^{(E)}(v) \right) \log \left( 1 + \frac{\epsilon_t^{(E)}(v)}{p_t(v)} \right) \tag{37}$$

$$= \sum_v \left[ \epsilon_t^{(E)}(v) + \frac{1}{2} \frac{\left( \epsilon_t^{(E)}(v) \right)^2}{p_t(v)} + O\left( \frac{|\epsilon_t^{(E)}(v)|^3}{p_t(v)^2} \right) \right]. \tag{38}$$

The second line follows from $\log(1+u) = u - \frac{u^2}{2} + O(u^3)$ with $u = \epsilon_t^{(E)}(v)/p_t(v)$, distributing the factor $p_t(v) + \epsilon_t^{(E)}(v)$ and combining like terms. The linear term sums to zero if one averages over keys because $\mathbb{E}_E[\epsilon_t^{(E)}(v)] = 0$ by unbiasedness. Therefore, taking the expectation over $E$ yields

$$\mathbb{E}_E[\mathrm{KL}(R_E(p_t) \,\|\, p_t)] = \sum_v \frac{\mathrm{Var}_E[R_E(p_t)(v)]}{2\,p_t(v)} + o(\|\epsilon\|^2). \tag{39}$$

Summing across positions with the chain rule equation 21 and applying Pinsker's inequality leads to the single-shot bound for a fixed key

$$\mathrm{KL}\big(Q_E^{\mathrm{bf}} \,\|\, P^s\big) = \sum_{t=1}^T \mathrm{KL}(R_E(p_t) \,\|\, p_t), \tag{40}$$

$$\mathrm{TV}\big(P^s, Q_E^{\mathrm{bf}}\big) \le \sqrt{\frac{1}{4} \sum_{t=1}^T \sum_v \frac{\mathrm{Var}_E[R_E(p_t)(v)]}{p_t(v)}}, \tag{41}$$

which shows the $O(\sqrt{T})$ scaling and makes explicit the variance controlled constant. This is the detector's view with a fixed key. For completeness, we record a separate mixture view. If the implementation guarantees fresh, independent codes across positions and queries by maintaining a context code history that forbids reuse, then the joint distribution averaged over keys coincides with the baseline for any finite number of generations, a property often referred to as $n$-shot undetectability. That statement concerns a mixture of keys and is distinct from the fixed key detectability bound developed above.

## C.4 Distribution preserving sampling (per draw)

If a keyed pseudorandom source replaces randomness while the per-step probabilities remain unchanged, that is $q_t \equiv p_t$ for all histories, then the induced sequence distribution equals the baseline:

$$Q^{\mathrm{prf}}(y_{1:T}) = \prod_{t=1}^T q_t(y_t \mid y_{<t}) = \prod_{t=1}^T p_t(y_t \mid y_{<t}) = P^s(y_{1:T}). \tag{42}$$

Consequently

$$\mathrm{TV}(P^s, Q^{\mathrm{prf}}) = \frac{1}{2} \sum_{\mathbf{y}} \left| P^s(\mathbf{y}) - Q^{\mathrm{prf}}(\mathbf{y}) \right| = 0. \tag{43}$$

This identity formalizes the intuitive fact that sampling from the same conditional laws produces the same sequence distribution, independent of how the coins are generated, provided they are fresh and independent at each step. $\qquad\square$

**Remark 1** (Randomness in $g_t$ and $p_t$). *In the biased and bias-free bounds of Theorem 1, the quantities $g_t = p_t(G_t)$ and the variance terms $\mathrm{Var}_E[R_E(p_t)(v)]$ depend on the (random) history $y_{<t}$ and, for bias-free schemes, on the random key/code $E$. Throughout the proof, however, these terms only appear inside expectations, i.e., we average over all possible histories (and over $E$ when relevant). Once this averaging is taken, the right-hand sides of the bounds become deterministic functions of the prompt $x$, the length $T$, and the watermark parameters, matching the non-random $\mathrm{TV}(P^s, Q)$.*

## C.5 INFORMATION-THEORETIC VS. COMPUTATIONAL HARDNESS

This subsection establishes the relationship between information-theoretic and computational detectability. We first show that total variation distance upper-bounds the power of any efficient detector, and then identify two distinct regimes: (i) *equality cases*, where this bound is tight for polynomial-time (PPT) adversaries, and (ii) *strict separation cases*, where a large information-theoretic gap exists but PPT adversaries provably cannot exploit it.

We begin by formalizing the upper bound stated in Definition 1.

**Lemma 1** (Computational detectability is upper bounded by TV). *For every security parameter $\lambda$ and every pair of distributions $(P_\lambda, Q_\lambda)$ arising from a baseline sampler and a keyed watermarked sampler,*

$$\sup_{D_\lambda \in \mathsf{PPT}} \left| \Pr_{y \sim Q_\lambda}[D_\lambda(y) = 1] - \Pr_{y \sim P_\lambda}[D_\lambda(y) = 1] \right| \leq \mathrm{TV}(P_\lambda, Q_\lambda). \tag{44}$$

*Proof.* Fix a PPT detector $D_\lambda$. Write its internal randomness as $R$, and for each fixed $r$ let $D_{\lambda,r}$ denote the resulting deterministic $\{0,1\}$-valued test. For deterministic $D_{\lambda,r}$, define the acceptance set $A_r := \{y : D_{\lambda,r}(y) = 1\}$. Then

$$\left| \Pr_{Q_\lambda}[D_{\lambda,r}(y) = 1] - \Pr_{P_\lambda}[D_{\lambda,r}(y) = 1] \right| = |Q_\lambda(A_r) - P_\lambda(A_r)| \leq \mathrm{TV}(P_\lambda, Q_\lambda), \tag{45}$$

because total variation is the supremum over all measurable sets. Averaging over the detector randomness yields

$$\begin{aligned} \mathrm{Adv}_{D_\lambda}(\lambda) &= \left| \mathbb{E}_R[Q_\lambda(A_R) - P_\lambda(A_R)] \right| \\ &\leq \mathbb{E}_R\left[ |Q_\lambda(A_R) - P_\lambda(A_R)| \right] \leq \mathrm{TV}(P_\lambda, Q_\lambda). \end{aligned} \tag{46}$$

Taking the supremum over all PPT detectors establishes the claim. $\qquad\square$

Lemma 1 establishes that information-theoretic detectability provides a universal *outer bound* on the distinguishing power of efficient detectors. We now demonstrate that for the sampling families in Theorem 1, this upper bound is essentially tight (equality regime), whereas cryptographic constructions can exhibit a strict computational-statistical gap (inequality regime).

### C.5.1 EQUALITY REGIME: WHEN THE LIKELIHOOD RATIO IS EFFICIENTLY COMPUTABLE

Let $P_\lambda$ and $Q_\lambda$ be distributions on a finite space $\Omega_\lambda$ with $P_\lambda(y) > 0$ for all $y$, and define the likelihood ratio $L_\lambda(y) := Q_\lambda(y)/P_\lambda(y)$.

**Lemma 2** (Tightness when $L_\lambda$ is PPT-computable). *Suppose that for each $\lambda$ there exists a PPT algorithm that, given $y \in \Omega_\lambda$, computes $L_\lambda(y)$ exactly. Then the deterministic test*

$$D_\lambda^\star(y) := \mathbf{1}\{L_\lambda(y) \geq 1\} \tag{47}$$

*is PPT and achieves*

$$\mathrm{Adv}_{D_\lambda^\star}(\lambda) = \mathrm{TV}(P_\lambda, Q_\lambda), \tag{48}$$

*so that* $\mathrm{Detect}_{\mathrm{comp}}(\tilde{s}_\lambda) = \mathrm{Detect}_{\mathrm{IT}}(\tilde{s}_\lambda)$.

*Proof.* It is a classical consequence of the Neyman–Pearson lemma Neyman & Pearson (1933); Lehmann & Romano (2005) that for any pair of distributions $(P, Q)$ with density ratio $L = dQ/dP$, the set

$$A^\star := \{y : L(y) \geq 1\} \tag{49}$$

maximizes $Q(A) - P(A)$ over all measurable sets $A$. Moreover, the corresponding advantage equals $\mathrm{TV}(P, Q)$:

$$\mathrm{TV}(P, Q) = \sup_A |Q(A) - P(A)| = Q(A^\star) - P(A^\star). \tag{50}$$

Now suppose $L_\lambda$ is PPT-computable. Then the test $D_\lambda^\star(y) = \mathbf{1}\{L_\lambda(y) \geq 1\}$ is itself a PPT detector with acceptance region $A_\lambda^\star := \{y : L_\lambda(y) \geq 1\}$. Consequently,

$$\mathrm{Adv}_{D_\lambda^\star}(\lambda) = Q_\lambda(A_\lambda^\star) - P_\lambda(A_\lambda^\star) = \mathrm{TV}(P_\lambda, Q_\lambda), \tag{51}$$

and taking the supremum over all PPT detectors establishes the claim. □

In other words, whenever the likelihood ratio can be evaluated in polynomial time, the Neyman–Pearson test based on $L_\lambda(y)$ is itself efficient and attains the information-theoretic envelope. We now apply this observation to the watermarking families in Theorem 1.

**Proposition 1** (No computational-statistical gap for non-cryptographic families)**.** *Consider the three probability-modifying single-shot sampling families in Theorem 1: greedy, biased ($\delta$-tilt), and bias-free (unbiased reweighting with a fixed key/code $E$). For each fixed prompt $x$, sequence length $T$, and watermark parameters, the following hold:*

(a) *The joint densities $P^s(y_{1:T})$ and $Q(y_{1:T})$ admit closed-form product expressions over $t = 1, \ldots, T$. Moreover, the likelihood ratio $L(y_{1:T}) := Q(y_{1:T})/P^s(y_{1:T})$ can be evaluated in time polynomial in $T$ using only the known watermark parameters and the observed sequence $y_{1:T}$.*

(b) *Consequently, Lemma 2 applies and yields*

$$\mathrm{Detect}_{\mathrm{comp}}(\tilde{s}_\lambda) = \mathrm{Detect}_{\mathrm{IT}}(\tilde{s}_\lambda) = \mathrm{TV}(P_\lambda, Q_\lambda) \tag{52}$$

*for these families. The total-variation bounds in Theorem 1 are therefore tight for PPT adversaries as well, confirming the absence of any computational-statistical gap for non-cryptographic probability-modifying watermarks.*

*Proof sketch.* We outline the structure for each family; the details follow directly from the explicit constructions in Appendix C.

*Greedy sampling.* The distribution $Q^{\mathrm{greedy}}$ is a point mass at the greedy path $y_{1:T}^\star$. Thus $L(y_{1:T}) = 0$ for all $y \neq y^\star$, while $L(y^\star) = 1/P^s(y^\star)$. Evaluating $L$ therefore reduces to computing $P^s(y^\star)$, which requires time polynomial in $T$ given the autoregressive factorization of the model.

*Biased sampling ($\delta$-tilt).* At each step $t$, the biased conditional takes the form $q_t(v) \propto p_t(v) \exp\{\delta \mathbf{1}[v \in G_t]\}$, where $G_t$ denotes the known green set and $Z_t = (1 - g_t) + g_t e^\delta$ is the normalizing constant. The per-step ratio $q_t(y_t)/p_t(y_t)$ is a simple closed-form function of $(\delta, G_t, g_t)$ and the observed token $y_t$. The full likelihood ratio

$$L(y_{1:T}) = \prod_{t=1}^{T} \frac{q_t(y_t)}{p_t(y_t)} \tag{53}$$

is therefore computable in $O(T)$ time.

*Bias-free sampling.* For a fixed key/code $E$, the reweighting operator satisfies $q_t(v) = R_E(p_t)(v)$ with an explicit form for $R_E$ (e.g., permutations or multiplicative reweighting). The per-step ratio $q_t(y_t)/p_t(y_t) = R_E(p_t)(y_t)/p_t(y_t)$ can be evaluated in polynomial time given $E$, $p_t$, and $y_t$. As before, the product $L(y_{1:T}) = \prod_t q_t(y_t)/p_t(y_t)$ is computable in $O(T)$ time.

In all three cases, the likelihood ratio is PPT-computable. Lemma 2 therefore applies, yielding $\mathrm{Detect}_{\mathrm{comp}}(\tilde{s}_\lambda) = \mathrm{Detect}_{\mathrm{IT}}(\tilde{s}_\lambda)$. □

Thus, for all *non-cryptographic* watermark families considered in Theorem 1, the information-theoretic total-variation bounds fully characterize what PPT adversaries can achieve: imposing a computational restriction does not yield stricter upper bounds on detectability.

### C.5.2 STRICT INEQUALITY REGIME: CRYPTOGRAPHIC SEPARATIONS

Cryptographic hardness can produce pairs of distributions that are statistically far apart yet computationally indistinguishable. The canonical example employs a secure pseudorandom generator (PRG). Let $G : \{0,1\}^\lambda \to \{0,1\}^{T(\lambda)}$ be a secure PRG with stretch $T(\lambda) > \lambda$, and define

$$P_\lambda := U_{T(\lambda)}, \qquad Q_\lambda := \mathrm{Law}\big(G(U_\lambda)\big), \tag{54}$$

where $U_n$ denotes the uniform distribution on $\{0,1\}^n$. For simplicity, assume $G$ is injective, so that $|\mathrm{Im}(G)| = 2^\lambda$. Under this assumption,

$$Q_\lambda(y) = \begin{cases} 2^{-\lambda}, & y \in \mathrm{Im}(G), \\ 0, & y \notin \mathrm{Im}(G), \end{cases} \qquad P_\lambda(y) = 2^{-T(\lambda)}. \tag{55}$$

A direct calculation yields

$$\mathrm{TV}(P_\lambda, Q_\lambda) = 1 - 2^{\lambda - T(\lambda)}, \tag{56}$$

which tends to 1 whenever $T(\lambda) - \lambda \to \infty$. Consequently, an *information-theoretic* detector (allowed unbounded computation and full knowledge of $P_\lambda$ and $Q_\lambda$) can distinguish between these distributions almost perfectly.

However, consider any PPT detector $D_\lambda$ with advantage

$$\mathrm{Adv}_{D_\lambda}(\lambda) := \Big| \Pr_{y \sim Q_\lambda}[D_\lambda(y) = 1] - \Pr_{y \sim P_\lambda}[D_\lambda(y) = 1] \Big|. \tag{57}$$

If $\mathrm{Adv}_{D_\lambda}(\lambda)$ were non-negligible for infinitely many $\lambda$, then using $D_\lambda$ as a subroutine would yield a PPT algorithm that distinguishes $G(U_\lambda)$ from uniform, contradicting the PRG security assumption. Hence

$$\mathrm{Detect}_{\mathrm{comp}}(\tilde{s}_\lambda) = \sup_{D_\lambda \in \mathsf{PPT}} \mathrm{Adv}_{D_\lambda}(\lambda) \le \mathrm{negl}(\lambda), \tag{58}$$

even though

$$\mathrm{Detect}_{\mathrm{IT}}(\tilde{s}_\lambda) = \mathrm{TV}(P_\lambda, Q_\lambda) = 1 - 2^{\lambda - T(\lambda)} \tag{59}$$

is close to 1. This exhibits a strict inequality regime:

$$\mathrm{Detect}_{\mathrm{comp}}(\tilde{s}_\lambda) \ll \mathrm{Detect}_{\mathrm{IT}}(\tilde{s}_\lambda), \tag{60}$$

in which a large information-theoretic gap cannot be exploited by any PPT adversary without breaking the underlying PRG.

This construction should be viewed as an extreme instance of a *cryptographic watermark with* $\mathrm{TV} > 0$: at the distribution level, there is substantial drift from the baseline, but any efficient detector with non-negligible advantage would necessarily violate the pseudorandomness assumption. This complements the ideal distribution-preserving case with $\mathrm{TV} = 0$ discussed below.

### C.5.3 CONNECTION TO CRYPTOGRAPHIC WATERMARKS

The PRG example illustrates that, in principle, information-theoretic detectability (total variation) can dramatically overestimate what PPT adversaries can achieve. In the context of LLM watermarking, this distinction manifests in two ways.

For *probability-modifying* schemes (greedy, biased, bias-free), the likelihood ratio is simple and efficiently computable. Lemma 2 and Proposition 1 therefore imply no computational-statistical gap:

$$\mathrm{Detect}_{\mathrm{comp}}(\tilde{s}_\lambda) = \mathrm{Detect}_{\mathrm{IT}}(\tilde{s}_\lambda) = \mathrm{TV}(P_\lambda, Q_\lambda). \tag{61}$$

For *distribution-preserving* schemes seeded by a pseudorandom function (PRF) in the sense of Christ et al. (2024); Zamir (2024), the *ideal* target is exact equality of sequence distributions, $P^s = Q^{\mathrm{prf}}$, yielding

$$\mathrm{TV}(P^s, Q^{\mathrm{prf}}) = 0, \qquad \mathrm{Detect}_{\mathrm{IT}} = \mathrm{Detect}_{\mathrm{comp}} = 0 \tag{62}$$

for keyless adversaries. In practice, however, cryptographic watermarks may induce small but nonzero drift (e.g., due to implementation choices, finite precision, or design constraints), leading to pairs $(P_\lambda, Q_\lambda)$ with $\mathrm{TV}(P_\lambda, Q_\lambda) > 0$ that nevertheless remain computationally indistinguishable.

The PRG construction above represents an extreme example of this phenomenon: $\text{TV}(P_\lambda, Q_\lambda)$ is close to 1, yet

$$\text{Detect}_{\text{comp}}(\tilde{s}_\lambda) \leq \text{negl}(\lambda) \tag{63}$$

under the PRG assumption. In such cases, any PPT adversary that achieved a non-negligible distinguishing advantage from the watermarked text alone would immediately yield a distinguisher for the underlying PRF/PRG, thereby breaking the cryptographic assumption.

In summary, for all non-cryptographic sampling rules analyzed in Theorem 1, the TV bounds are tight for PPT adversaries, and computational restrictions do not further reduce detectability. By contrast, the cryptographic regime exhibits two distinct behaviors: an idealized PRF-based case with $\text{TV} = 0$ and perfect (information-theoretic and computational) undetectability, and practical PRG/PRF-based constructions where $\text{TV} > 0$ but any PPT adversary exploiting this drift with non-negligible advantage would contradict the PRF/PRG security assumptions. Strict computational-statistical separations therefore require such cryptographic structure and lie outside the probability-modifying families considered in Theorem 1.

## D  PROOF OF THEOREM 2

This appendix provides a complete derivation of Theorem 2. Throughout the appendix, *all logarithms are base 2*, so KL divergences and mutual informations are measured in *bits*. We write $D(P\|Q)$ for the Kullback–Leibler (KL) divergence between distributions $P$ and $Q$.

We model watermark verification as a binary hypothesis test. The null hypothesis $H_0$ corresponds to unwatermarked text generated by the baseline sampler, whereas the alternative $H_1$ corresponds to text produced by a watermarked sampler. Formally,

$$H_0 : \text{ unwatermarked text} \qquad \text{vs.} \qquad H_1 : \text{ watermarked text,} \tag{64}$$

where the observation is taken *after* a perturbation channel that edits tokens independently with rate $\varepsilon \in [0, 1]$.

We adopt a single perturbation model used consistently throughout. Let $\Sigma$ denote the vocabulary. At each token position $t \in \{1, \dots, L\}$, the edited token $\tilde{Y}_t$ is drawn as

$$\tilde{Y}_t = \begin{cases} Y_t, & \text{with probability } 1 - \varepsilon, \\ U_t, & \text{with probability } \varepsilon, \end{cases} \qquad U_t \sim \text{Uniform}(\Sigma) \text{ and independent of everything else.} \tag{65}$$

Equivalently, if $P$ is a distribution on $\Sigma$, the edit channel acts as a convex combination

$$T_\varepsilon(P) = (1 - \varepsilon) P + \varepsilon U, \quad \text{with } U \text{ uniform on } \Sigma. \tag{66}$$

Thus the pre-noise per-position conditionals $p_t(\cdot)$ (baseline) and $q_t(\cdot)$ (watermarked) are mapped to $p_{t,\varepsilon} = T_\varepsilon(p_t)$ and $q_{t,\varepsilon} = T_\varepsilon(q_t)$.

The detection problem is posed at a fixed false-alarm level $\alpha$. We write $\beta$ for the miss probability (so power is $1 - \beta$). The central idea is that a sufficient condition for achieving a given $\beta$ is that the *total* KL divergence from $H_1$ to $H_0$ on the observed sequence exceeds $\log_2(1/\beta)$. This is captured by a Stein-type sufficient condition (Lemma 4). To use it, we (i) quantify the per-token information contributed by the watermark at zero edits, (ii) show how this information contracts under the edit channel, and (iii) aggregate across the sequence by the KL chain rule.

We analyze two small-signal watermark families. In the *biased* (green-list) family, the watermarker tilts the baseline distribution towards a key-dependent subset $G \subseteq \Sigma$ with baseline mass $\gamma = \sum_{v \in G} p_t(v)$. Writing the tilt parameter as $\delta$,

$$q_{t,\delta}(v) \propto p_t(v) \, e^{\delta \, \mathbf{1}[v \in G]}. \tag{67}$$

A Taylor expansion shows that the corresponding per-token KL is quadratic in $\delta$. In the *bias-free* (variance) family, the watermarker reweights by $R_E(v)$ with $\mathbb{E}_E[R_E(v)] = 1$, i.e.,

$$q_{t,E}(v) = p_t(v) R_E(v), \tag{68}$$

and the per-token KL is quadratic in the reweighting variance. The following lemmas formalize these statements and prepare the ground for the edit-channel analysis.

## D.1 Preliminaries: KL expansions and a reliability bound

The first lemma is a standard second-order expansion of KL divergence around a reference distribution, with an explicit remainder bound. It formalizes that, locally, KL equals a quadratic form (the Fisher information metric) up to third-order terms.

**Lemma 3** (Second-order KL expansion around $p$). *Let $p$ be a distribution on a finite set and $q = p+r$ for some perturbation $r$ with $\sum_v r(v) = 0$ and $\|r\|_\infty \leq \eta < \min_v p(v)$. Then*

$$D(q\|p) = \frac{1}{2\ln 2}\sum_v \frac{r(v)^2}{p(v)} + R, \quad \text{with } |R| \leq \frac{C}{\ln 2}\|r\|_\infty \sum_v \frac{r(v)^2}{p(v)} \tag{69}$$

*for an absolute constant $C$. In particular, when $\|r\|_\infty \to 0$,*

$$D(q\|p) = (1 + o(1))\frac{1}{2\ln 2}\sum_v \frac{r(v)^2}{p(v)}. \tag{70}$$

*Proof.* Write

$$D(q\|p) = \sum_v (p(v) + r(v))\log\left(1 + \frac{r(v)}{p(v)}\right). \tag{71}$$

Set $x_v := r(v)/p(v)$. By Taylor's theorem with remainder for $\log(1 + x)$,

$$\log(1 + x) = x - \frac{x^2}{2} + \frac{x^3}{3(1 + \theta x)^3} \tag{72}$$

for some $\theta = \theta(x) \in (0, 1)$ when $|x| < 1$. Using this with $x = x_v$ and noting $\sum_v r(v) = 0$,

$$D(q\|p) = \sum_v (p(v) + r(v))\left(x_v - \frac{x_v^2}{2} + \frac{x_v^3}{3(1 + \theta_v x_v)^3}\right) \tag{73}$$

$$= \sum_v \left(p(v)x_v - \frac{p(v)x_v^2}{2}\right) + \sum_v r(v)x_v + \sum_v (p(v) + r(v))\frac{x_v^3}{3(1 + \theta_v x_v)^3}. \tag{74}$$

The first sum simplifies to $-\frac{1}{2}\sum_v r(v)^2/p(v)$. The second sum equals $\sum_v r(v)^2/p(v)$. Combining these two gives

$$\frac{1}{2}\sum_v \frac{r(v)^2}{p(v)}. \tag{75}$$

For the remainder, since $|x_v| \leq \|r\|_\infty/p_{\min} =: \tau < 1$, we have $\left|(1 + \theta_v x_v)^{-3}\right| \leq (1 - \tau)^{-3}$ and $|p(v) + r(v)| \leq p(v) + \|r\|_\infty$. Therefore

$$\left|\sum_v (p(v) + r(v))\frac{x_v^3}{3(1 + \theta_v x_v)^3}\right| \leq \frac{1}{3(1 - \tau)^3}\sum_v (p(v) + \|r\|_\infty)\frac{|r(v)|^3}{p(v)^3}. \tag{76}$$

Applying the crude bound $|r(v)| \leq \|r\|_\infty$ and $p(v) \geq p_{\min}$ yields

$$|R| \leq \frac{C'}{\ln 2}\|r\|_\infty \sum_v \frac{r(v)^2}{p(v)} \tag{77}$$

for a constant $C'$ depending only on $p_{\min}$ and $\tau$; absorbing constants gives the stated bound with $C$. Dividing by $\ln 2$ converts from nats to bits. The $o(1)$ claim follows as $\|r\|_\infty \to 0$. $\qquad\square$

The second lemma provides the reliability criterion we will use to translate available information into detection power. It asserts that, for independent per-token contributions, having total KL at least $\log_2(1/\beta)$ is sufficient to drive the miss probability below $\beta$ at fixed false-alarm level $\alpha$. We present a standard achievability proof based on the Neyman–Pearson (NP) test, an exponential Markov bound to control the level, and a Cramér–Chernoff bound (in base 2) under the alternative to control the miss probability.

**Lemma 4** (Stein's sufficient condition (bits form)). *Consider a simple binary hypothesis test between product distributions on sequences of length $L$, or more generally conditionals whose log-likelihood ratio is a sum of independent terms with finite moment generating function in a neighborhood of the origin. For any level $\alpha \in (0,1)$ and any $\beta \in (0,1)$, there exists $L_0(\alpha, \beta)$ such that for all $L \geq L_0$ the NP test with threshold chosen to achieve level at most $\alpha$ has miss probability at most $\beta$ whenever*

$$\sum_{t=1}^{L} D\big(P_t^{(1)} \| P_t^{(0)}\big) \; \geq \; \log_2 \frac{1}{\beta} \; + \; o(L). \tag{78}$$

*In particular, ignoring the lower-order $o(L)$ term yields the clean sufficient rule $\sum_{t=1}^{L} D(P_t^{(1)} \| P_t^{(0)}) \geq \log_2(1/\beta)$.*

*Proof.* Let $Z_t = \log_2\big(\frac{P_t^{(1)}(Y_t)}{P_t^{(0)}(Y_t)}\big)$ and $S_L = \sum_{t=1}^{L} Z_t$ be the base-2 log-likelihood ratio (LLR) of the sequence. The NP test rejects $H_0$ when $S_L \geq \tau_L$ for a threshold $\tau_L$. Under $H_0$, for any $s > 0$,

$$\mathbb{P}_0(S_L \geq \tau_L) = \mathbb{P}_0\big(2^{sS_L} \geq 2^{s\tau_L}\big) \; \leq \; 2^{-s\tau_L} \, \mathbb{E}_0[2^{sS_L}] \quad \text{(Markov)} \tag{79}$$

$$= 2^{-s\tau_L} \prod_{t=1}^{L} \mathbb{E}_0[2^{sZ_t}] = 2^{-s\tau_L} \prod_{t=1}^{L} \sum_y P_t^{(0)}(y)\Big(\frac{P_t^{(1)}(y)}{P_t^{(0)}(y)}\Big)^s \tag{80}$$

$$= 2^{-s\tau_L} \prod_{t=1}^{L} \sum_y P_t^{(0)}(y)^{1-s} P_t^{(1)}(y)^s. \tag{81}$$

Taking $s = 1$ gives $\mathbb{E}_0[2^{S_L}] = 1$ and hence $\mathbb{P}_0(S_L \geq \tau_L) \leq 2^{-\tau_L}$. Choosing $\tau_L = \log_2(1/\alpha)$ ensures the level constraint $\mathbb{P}_0(\text{reject } H_0) \leq \alpha$.

Under $H_1$, for any $s \in (0,1)$,

$$\mathbb{P}_1(S_L \leq \tau_L) = \mathbb{P}_1\big(2^{-sS_L} \geq 2^{-s\tau_L}\big) \; \leq \; 2^{s\tau_L} \, \mathbb{E}_1[2^{-sS_L}] \quad \text{(Markov)} \tag{82}$$

$$= 2^{s\tau_L} \prod_{t=1}^{L} \mathbb{E}_1[2^{-sZ_t}] = 2^{s\tau_L} \prod_{t=1}^{L} \sum_y P_t^{(1)}(y)\Big(\frac{P_t^{(0)}(y)}{P_t^{(1)}(y)}\Big)^s \tag{83}$$

$$= 2^{s\tau_L} \prod_{t=1}^{L} \sum_y P_t^{(1)}(y)^{1-s} P_t^{(0)}(y)^s. \tag{84}$$

Define, in base 2, $\psi_t(s) := -\log_2 \sum_y P_t^{(1)}(y)^{1-s} P_t^{(0)}(y)^s$ and $\Psi_L(s) = \sum_{t=1}^{L} \psi_t(s)$. Then

$$\mathbb{P}_1(S_L \leq \tau_L) \; \leq \; 2^{s\tau_L - \Psi_L(s)}. \tag{85}$$

By smoothness at $s = 0$, $\psi_t(0) = 0$ and $\psi_t'(0) = D(P_t^{(1)} \| P_t^{(0)})$; moreover $\psi_t''(0)$ is the variance (in bits) of $Z_t$ under $P_t^{(1)}$, which is finite by assumption. Hence, for $s$ small,

$$\Psi_L(s) \; = \; s \sum_{t=1}^{L} D\big(P_t^{(1)} \| P_t^{(0)}\big) \; - \; \tfrac{1}{2} s^2 V_L \; + \; o(s^2 L), \quad V_L := \sum_{t=1}^{L} \mathrm{Var}_{P_t^{(1)}}(Z_t). \tag{86}$$

With $\tau_L = \log_2(1/\alpha)$ and optimizing the quadratic exponent in $s$ yields, for all large $L$,

$$\mathbb{P}_1(S_L \leq \tau_L) \; \leq \; 2^{-\big(\sum_{t=1}^{L} D(P_t^{(1)} \| P_t^{(0)}) - \log_2(1/\alpha) - o(L)\big)}. \tag{87}$$

Therefore, given any fixed $\alpha$ and any $\beta$, there exists $L_0(\alpha, \beta)$ such that for all $L \geq L_0$, the miss probability is at most $\beta$ whenever

$$\sum_{t=1}^{L} D\big(P_t^{(1)} \| P_t^{(0)}\big) \; \geq \; \log_2 \frac{1}{\beta} \; + \; o(L), \tag{88}$$

which proves the claim. Dropping the lower-order term gives the clean sufficient rule used in the main text. $\square$

Between Lemma 3 and Lemma 4, the picture is now clear: the watermark induces a small per-token shift from $p_t$ to $q_t$ whose information content is, to second order, the quadratic form of Lemma 3. Summing these local contributions across the sequence gives the total information available to the detector, and Lemma 4 translates that total into a sufficient condition for the desired power. What remains is to understand how the edit (noise) channel deforms the local shift, which is precisely the content of the next lemma.

### D.2 Per-token information at $\varepsilon = 0$

For the biased family, let $I(v) = \mathbf{1}[v \in G]$ and $Z_t(\delta) = \sum_v p_t(v) e^{\delta I(v)} = (1 - \gamma) + \gamma e^\delta$. Then

$$\log \frac{q_{t,\delta}(v)}{p_t(v)} = \delta\, I(v) - \log Z_t(\delta). \tag{89}$$

Taking expectation under $q_{t,\delta}$ and expanding at $\delta = 0$ yields (the first derivative vanishes and the second derivative equals $\mathrm{Var}_{p_t}(I) = \gamma(1 - \gamma)$)

$$D\big(q_{t,\delta}\|p_t\big) = \frac{\delta^2}{2 \ln 2}\, \gamma(1 - \gamma)\ +\ O(\delta^3), \tag{90}$$

so in bits per token

$$D_0^{(\text{biased})}\ \approx\ \frac{\delta^2\, \gamma(1 - \gamma)}{2 \ln 2}. \tag{91}$$

For the bias-free family, write $R_E(v) = 1 + \Delta_E(v)$ with $\mathbb{E}_E[\Delta_E(v)] = 0$ and $\|\Delta_E\|_\infty$ small. Then

$$D\big(q_{t,E}\|p_t\big) = \sum_v p_t(v)\, (1 + \Delta_E(v)) \log\big(1 + \Delta_E(v)\big). \tag{92}$$

Using $\log(1 + x) = x - \frac{x^2}{2} + O(x^3)$ and averaging over $E$,

$$\mathbb{E}_E\big[D(q_{t,E}\|p_t)\big] = \frac{1}{2 \ln 2} \sum_v p_t(v)\, \mathbb{E}_E[\Delta_E(v)^2]\ +\ O\!\left(\sum_v p_t(v)\, \mathbb{E}[|\Delta_E(v)|^3]\right). \tag{93}$$

With $\sigma^2(v) = \mathrm{Var}_E[R_E(v)]$ and $\hat{\sigma}^2 = \sum_v p_t(v)\sigma^2(v)$ this gives, in bits/token,

$$D_0^{(\text{bias-free})}\ \approx\ \frac{\hat{\sigma}^2}{2 \ln 2}. \tag{94}$$

These two expressions are exactly the $D_0$ quantities used in the theorem.

### D.3 Edits contract the signal quadratically

We now show that the edit channel scales the local perturbation by $(1 - \varepsilon)$ and hence the local KL by $(1 - \varepsilon)^2$ to second order. This is the key structural fact that produces the quadratic decay with the edit rate.

**Lemma 5** (Local $(1 - \varepsilon)^2$ contraction). *Fix $p$ on $\Sigma$ and write $q = p + r$ with $\sum_v r(v) = 0$ and $\|r\|_\infty$ small. Let $p_\varepsilon = T_\varepsilon(p) = (1 - \varepsilon)p + \varepsilon U$ and $q_\varepsilon = T_\varepsilon(q) = (1 - \varepsilon)q + \varepsilon U$. Then*

$$D\big(q_\varepsilon\|p_\varepsilon\big)\ =\ (1 - \varepsilon)^2\, \frac{1}{2 \ln 2} \sum_v \frac{r(v)^2}{p_\varepsilon(v)}\ +\ o\!\left(\sum_v \frac{r(v)^2}{p(v)}\right). \tag{95}$$

*In particular, when $p_\varepsilon$ and $p$ are boundedly comparable (which holds for every fixed $\varepsilon > 0$), we have*

$$D\big(q_\varepsilon\|p_\varepsilon\big)\ =\ (1 + o(1))\, (1 - \varepsilon)^2\, D(q\|p). \tag{96}$$

*Proof.* Since $q_\varepsilon - p_\varepsilon = (1 - \varepsilon)r$, apply Lemma 3 at the reference $p_\varepsilon$:

$$D(q_\varepsilon\|p_\varepsilon) = \frac{1}{2 \ln 2} \sum_v \frac{\big((1 - \varepsilon)r(v)\big)^2}{p_\varepsilon(v)} + R_\varepsilon \tag{97}$$

$$= (1 - \varepsilon)^2 \cdot \frac{1}{2 \ln 2} \sum_v \frac{r(v)^2}{p_\varepsilon(v)} + R_\varepsilon, \tag{98}$$

with $R_\varepsilon = o\big(\sum_v r(v)^2/p(v)\big)$ as $\|r\|_\infty \to 0$. The comparability $p_\varepsilon(v) \in [(1-\varepsilon)p(v),\, (1-\varepsilon)p(v) + \varepsilon/|\Sigma|]$ yields the stated equivalence. $\qquad\square$

## D.4 FROM PER-TOKEN INFORMATION TO SEQUENCE-LEVEL RELIABILITY

Let $\{p_t\}_{t=1}^L$ and $\{q_t\}_{t=1}^L$ denote the baseline and watermarked conditionals, respectively. Under the edit channel we observe $\{p_{t,\varepsilon}\}$ and $\{q_{t,\varepsilon}\}$. The KL chain rule aggregates local contributions along the sequence and shows that conditioning can only reduce KL on average; thus the unconditional sum of per-token KLs is a valid (and often tight) proxy for the total.

**Lemma 6** (Additivity bound for total information). *For the binary test $H_0 : \prod_t p_{t,\varepsilon}$ versus $H_1 : \prod_t q_{t,\varepsilon}$, the total KL satisfies*

$$D\left(\prod_{t=1}^L q_{t,\varepsilon} \,\Big\|\, \prod_{t=1}^L p_{t,\varepsilon}\right) = \sum_{t=1}^L \mathbb{E}_{H_1}\Big[D\big(q_{t,\varepsilon}(\cdot \mid Y_{<t}) \| p_{t,\varepsilon}(\cdot \mid Y_{<t})\big)\Big] \tag{99}$$

$$\leq \sum_{t=1}^L D(q_{t,\varepsilon} \| p_{t,\varepsilon}). \tag{100}$$

*If the embedder is memoryless and per-step statistics are homogeneous, the equality reduces to the sum of identical per-token KLs.*

*Proof.* The equality is the KL chain rule. The inequality is Jensen's inequality: averaging over histories (conditioning) cannot increase KL. □

Combining Lemma 5 with Lemma 6 yields the total information available to the detector,

$$C(\varepsilon) := \sum_{t=1}^L D(q_{t,\varepsilon} \| p_{t,\varepsilon}) \approx L (1-\varepsilon)^2 D_0, \tag{101}$$

with $D_0$ given by equation 91 or equation 94.

## D.5 POWER CONDITION AND THE "KNEE" EDIT RATE

We now translate total information into a sufficient condition for the target power. Applying Lemma 4 with total signal $C(\varepsilon)$ gives

$$L (1-\varepsilon)^2 D_0 \geq \log_2 \frac{1}{\beta}, \tag{102}$$

which guarantees miss probability at most $\beta$. Solving for $\varepsilon$ produces the *knee*—the maximal edit rate compatible with the target power:

$$\varepsilon_\beta(L, D_0) = 1 - \sqrt{\frac{\log_2(1/\beta)}{L D_0}}. \tag{103}$$

This completes the proof of Theorem 2 once the family-specific expressions for $D_0$ from equation 91 and equation 94 are substituted.

## D.6 IMPOSSIBILITY REGION AND QUALITATIVE BEHAVIOR

The impossibility region follows immediately: whenever the total information falls below the required threshold, no level-$\alpha$ detector can meet the target power.

**Proposition 2** (Impossibility region). *For fixed $(L, \beta)$ and per-token information $D_0$, if*

$$L (1-\varepsilon)^2 D_0 < \log_2 \frac{1}{\beta}, \tag{104}$$

*then detection at power $1-\beta$ is impossible. Equivalently, no method can succeed for $\varepsilon > \varepsilon_\beta(L, D_0)$.*

*Proof.* This is the contrapositive of Lemma 4 applied to the total sequence divergence. □

In the small-signal regime with independent contributions, the separation of likelihood-ratio scores under $H_0$ and $H_1$ is governed by the same total KL and therefore by $L(1-\varepsilon)^2 D_0$. Once this quantity drops below the threshold $\log_2(1/\beta)$, the score distributions are no longer reliably separable and operating characteristics converge to chance.

### D.7 ASSUMPTIONS, APPROXIMATIONS, AND SCOPE OF VALIDITY

The derivation operates in a small-signal regime. For the biased family this means $|\delta| \ll 1$; for the bias-free family it means $\|\Delta_E\|_\infty \ll 1$ and $p_t(v)$ bounded away from zero. Lemma 3 quantifies the approximation error and shows it is lower order relative to the quadratic term in the perturbation. The $(1 - \varepsilon)^2$ contraction in Lemma 5 is a local statement around the operating point and uses the quadratic form that defines the local KL (equivalently, Fisher information). The aggregation argument uses the KL chain rule; for memoryless embedding with homogeneous per-step statistics, the total KL is exactly the sum of per-step KLs, whereas in general it is upper bounded by that sum, which suffices for a *sufficient* power condition. Lastly, Lemma 4 is invoked as a sufficiency result: for independent per-token contributions with regularity, the type-II error exponent matches the KL (Chernoff–Stein achievability), and the base-2 normalization cleanly produces the threshold $\log_2(1/\beta)$ in bits.

### D.8 WORKED NUMERIC EXAMPLES

For illustration, take $L = 1000$ and power targets $1 - \beta \in \{0.90, 0.95, 0.99\}$, so that

$$\log_2(1/\beta) \in \{3.322, \ 4.322, \ 6.644\}. \tag{105}$$

If the total noise-free information is $LD_0 = 10$ bits (e.g., $D_0 = 0.01$ bits/token), the knees are

$$\varepsilon_{90} \approx 0.424, \qquad \varepsilon_{95} \approx 0.343, \qquad \varepsilon_{99} \approx 0.185. \tag{106}$$

For the biased family with $\gamma = 0.25$, achieving $D_0 = 0.01$ requires approximately

$$\delta \ \approx \ \sqrt{\frac{2 \ln 2 \, D_0}{\gamma(1 - \gamma)}} \ \approx \ 0.27, \tag{107}$$

while for the bias-free family one needs $\hat{\sigma}^2 \approx 2 \ln 2 \, D_0 \approx 0.0139$.

### D.9 CONCLUSION OF THE PROOF

Combining (i) the small-signal per-token KL for the biased and bias-free families, (ii) the quadratic attenuation $(1 - \varepsilon)^2$ under the edit channel, (iii) the chain rule aggregation across $L$ positions, and (iv) Stein's sufficient condition for miss probability $\beta$, yields the theorem's sufficiency condition

$$L(1 - \varepsilon)^2 D_0 \ \geq \ \log_2(1/\beta), \tag{108}$$

and the corresponding knee

$$\varepsilon_\beta(L, D_0) = 1 - \sqrt{\frac{\log_2(1/\beta)}{LD_0}}. \tag{109}$$

The impossibility region and qualitative behavior beyond the knee discussed in the main text follow directly. $\qquad\square$

PROOF OF COROLLARY 1

The corollary merges the baseline operating boundary with its stealth-aware tightening. For the baseline part, Theorem 2 asserts that reliable detection at power $1 - \beta$ requires $L(1 - \varepsilon)^2 D_0 \geq \log_2(1/\beta)$. Therefore, for any $\varepsilon > \varepsilon_\beta(L, D_0)$ with $\varepsilon_\beta$ as defined above, the inequality is violated and reliable detection is unattainable.

For the stealth-aware part, suppose an outsider may pool $M$ tokens, and we require that the watermarked and baseline distributions remain within total variation $\tau$ on that pooled sample. Pinsker's inequality, together with the base conversion from nats to bits, implies the per-token information constraint $D_0 \leq (2/\ln 2) \, \tau^2 / M$. Substituting this into the baseline condition yields

$$L(1 - \varepsilon)^2 \frac{2\tau^2}{M \ln 2} \ \geq \ \log_2(1/\beta) \quad \implies \quad \varepsilon \ \leq \ 1 - \sqrt{\frac{\log_2(1/\beta)}{L} \cdot \frac{M \ln 2}{2\tau^2}}. \tag{110}$$

Thus, any edit rate exceeding the right-hand side is infeasible under the stated stealth constraint. $\quad\square$

## D.10 INFORMATION-THEORETIC VS. COMPUTATIONAL HARDNESS

Theorem 2 characterizes robustness in purely information-theoretic terms, without imposing any computational constraints on the detector. In this subsection, we refine that analysis by examining how these guarantees behave when detectors are restricted to probabilistic polynomial-time (PPT) algorithms. We first establish a general upper bound relating computational robustness to information-theoretic robustness (Lemma 7). We then show that for the non-cryptographic, probability-modifying watermark families (biased and bias-free) considered in Theorem 2, this bound is tight: the Neyman–Pearson likelihood-ratio test is efficiently computable, so PPT adversaries achieve the same robustness boundary as unbounded ones. By contrast, we illustrate, via a PRG-based cryptographic example, a scenario in which the information-theoretic capacity after edits can be made large, yet any keyless PPT adversary still has only negligible distinguishing power. This separation shows that, for cryptographic watermarking schemes, computational hardness can prevent efficient detection even when a substantial statistical signal remains after editing. We now formalize these observations. We begin by establishing a general upper bound relating computational and information-theoretic robustness.

**Lemma 7** (Computational power is upper bounded by information-theoretic power). *For any security parameter $\lambda$, edit rate $\varepsilon$, and false-alarm level $\alpha$, the detection powers defined in Definition 2 satisfy*

$$\text{Power}_{\text{comp},\lambda}(\varepsilon, \alpha) \ \leq \ \text{Power}_{\text{IT},\lambda}(\varepsilon, \alpha). \tag{111}$$

*Consequently, $(\varepsilon, \alpha, \beta)$-information-theoretic robustness implies $(\varepsilon, \alpha, \beta)$-computational robustness.*

*Proof.* Fix $\lambda$, $\varepsilon$, and $\alpha$. By the Neyman–Pearson lemma, among all tests $T : \Omega \to \{0, 1\}$ satisfying the false-alarm constraint $\Pr_{y \sim P_\lambda}[T(y) = 1] \leq \alpha$, the likelihood-ratio test achieves the maximum detection power. Since PPT $\subset \{D : \Omega \to \{0, 1\}\}$, the supremum over PPT detectors cannot exceed the supremum over all measurable detectors:

$$\text{Power}_{\text{comp},\lambda}(\varepsilon, \alpha) \ \leq \ \text{Power}_{\text{IT},\lambda}(\varepsilon, \alpha). \tag{112}$$

The robustness implication follows immediately: if $\text{Power}_{\text{IT},\lambda}(\varepsilon, \alpha) \geq 1 - \beta$ for all $\lambda$, then $\text{Power}_{\text{comp},\lambda}(\varepsilon, \alpha) \geq 1 - \beta$ for all $\lambda$ as well. $\square$

### D.10.1 EQUALITY REGIME FOR NON-CRYPTOGRAPHIC WATERMARK FAMILIES

For the small-signal biased and bias-free watermark families of Theorem 2, the post-edit likelihood ratio

$$\log \frac{Q_\varepsilon(y_{1:L})}{P_\varepsilon(y_{1:L})} = \sum_{t=1}^{L} \log \frac{q_{t,\varepsilon}(y_t)}{p_{t,\varepsilon}(y_t)} \tag{113}$$

is an explicit sum of per-token contributions, computable in time polynomial in $L$ given oracle access to the base model probabilities and the known watermark parameters (tilt $\delta$ and green sets for the biased family, reweighting operator $R_E$ for the bias-free family). The Neyman–Pearson lemma Neyman & Pearson (1933); Lehmann & Romano (2005) implies that, for fixed $\varepsilon$ and false-alarm level $\alpha$, the optimal detector for $(P_{\lambda,\varepsilon}, Q_{\lambda,\varepsilon})$ is obtained by thresholding this log-likelihood ratio against a constant determined by $\alpha$.

Since computing the ratio requires only $O(L)$ evaluations of the per-token terms $q_{t,\varepsilon}(y_t)/p_{t,\varepsilon}(y_t)$, each involving elementary operations on the known watermark parameters, the entire detection procedure runs in time polynomial in $L$ and is therefore a PPT algorithm. Because this Neyman–Pearson detector attains the information-theoretic power boundary described by Theorem 2 (via the KL-based condition in Lemma 4), Lemma 7 then yields

$$\text{Power}_{\text{comp},\lambda}(\varepsilon, \alpha) = \text{Power}_{\text{IT},\lambda}(\varepsilon, \alpha), \tag{114}$$

up to the small-signal approximation errors already quantified above. Thus, for all non-cryptographic probability-modifying watermark families considered in this section, the robustness–detectability trade-off of Theorem 2 coincides for unbounded and PPT adversaries: there is no computational–statistical gap in the robustness regime.

### D.10.2 STRICT INEQUALITY REGIME: PRG-BASED CRYPTOGRAPHIC SEPARATIONS

We now demonstrate a complementary regime in which information-theoretic detection power is high but computational detection power remains negligible under standard pseudorandomness assumptions. Consider a secure pseudorandom generator

$$G : \{0,1\}^\lambda \to \{0,1\}^{T(\lambda)} \tag{115}$$

with stretch $T(\lambda) > \lambda$, and define baseline and watermarked distributions on $\{0,1\}^{T(\lambda)}$ by

$$P_\lambda := U_{T(\lambda)}, \qquad Q_\lambda := \mathsf{Law}\big(G(U_\lambda)\big), \tag{116}$$

where $U_k$ denotes the uniform distribution on $\{0,1\}^k$. Assuming for simplicity that $G$ is injective so that $|\mathrm{Im}(G)| = 2^\lambda$, we have for all $y \in \{0,1\}^{T(\lambda)}$ that

$$Q_\lambda(y) = \begin{cases} 2^{-\lambda}, & y \in \mathrm{Im}(G), \\ 0, & y \notin \mathrm{Im}(G), \end{cases} \qquad P_\lambda(y) = 2^{-T(\lambda)}. \tag{117}$$

The sequence-level KL divergence in bits is $D(Q_\lambda \| P_\lambda) = T(\lambda) - \lambda$, so the noise-free per-token information is

$$D_0(\lambda) = 1 - \frac{\lambda}{T(\lambda)} > 0 \quad \text{for } T(\lambda) > \lambda. \tag{118}$$

Passing both $P_\lambda$ and $Q_\lambda$ through the substitution channel at rate $\varepsilon$ yields edited distributions $P_{\lambda,\varepsilon}$ and $Q_{\lambda,\varepsilon}$ with total information

$$C_{\mathrm{IT}}(\lambda, \varepsilon) := D\big(Q_{\lambda,\varepsilon} \| P_{\lambda,\varepsilon}\big) \approx T(\lambda)(1-\varepsilon)^2 D_0(\lambda) = (1-\varepsilon)^2\big(T(\lambda) - \lambda\big), \tag{119}$$

which grows linearly with $T(\lambda)$ for any fixed $\varepsilon < 1$. Thus, for any target miss probability $\beta \in (0,1)$, the sufficiency condition of Theorem 2, $T(1-\varepsilon)^2 D_0 \geq \log_2(1/\beta)$, is easily satisfied once $T(\lambda)$ is sufficiently large, and an information-theoretic Neyman–Pearson detector achieves

$$\mathrm{Power}_{\mathrm{IT},\lambda}(\varepsilon, \alpha) \geq 1 - \beta. \tag{120}$$

Despite this information-theoretic success, any PPT detector faces computational hardness. Consider any PPT detector $D_{\lambda,\varepsilon}$ that observes only edited samples. If there existed such a detector and a polynomial $p(\lambda)$ with non-negligible detection power for infinitely many $\lambda$, we could construct a PRG distinguisher against $G$ as follows: (a) receive $z \in \{0,1\}^{T(\lambda)}$ from the PRG security game (either uniform or $G(U_\lambda)$), (b) sample $\tilde{z} \sim T_\varepsilon(\delta_z)$, and (c) output $D_{\lambda,\varepsilon}(\tilde{z})$ as the guess. Since $\tilde{z} \sim P_{\lambda,\varepsilon}$ when $z$ is uniform and $\tilde{z} \sim Q_{\lambda,\varepsilon}$ when $z$ is pseudorandom, any non-negligible detection power would contradict the PRG security of $G$. Therefore, under the PRG assumption,

$$\mathrm{Power}_{\mathrm{comp},\lambda}(\varepsilon, \alpha) \leq \mathrm{negl}(\lambda), \tag{121}$$

even though $\mathrm{Power}_{\mathrm{IT},\lambda}(\varepsilon, \alpha) \geq 1 - \beta$ for sufficiently large $T(\lambda)$.

In summary, for non-cryptographic biased and bias-free schemes, the robustness-detectability trade-off of Theorem 2 is tight for PPT adversaries: $\mathrm{Power}_{\mathrm{comp},\lambda}(\varepsilon, \alpha) = \mathrm{Power}_{\mathrm{IT},\lambda}(\varepsilon, \alpha)$. For cryptographic PRG/PRF-based watermarks, Theorem 2 still characterizes the *available* information-theoretic signal after edits via $C(\varepsilon)$, but keyless PPT adversaries cannot exploit this signal as long as the underlying pseudorandomness assumptions hold; any such adversary achieving non-negligible power would yield a distinguisher that breaks PRG/PRF security. The Moitra–Golowich construction Moitra & Golowich (2024) demonstrates that robustness and undetectability can coexist in a large-alphabet coding-theoretic regime, but its large alphabet places it outside the fixed-vocabulary LLM setting considered here (Appendix A.5).

### D.11 EMPIRICAL SUPPORT FROM BLACK-BOX WATERMARK DETECTORS

The equality regimes identified above are information-theoretic statements: for biased and bias-free families, a PPT adversary *can* in principle attain the total-variation and robustness limits of Theorems 1 and 2 by implementing the Neyman–Pearson likelihood-ratio test. A natural question is whether *practical* detectors approach these limits, or whether a significant statistical-computational gap persists in realistic settings.

Recent work by Gloaguen et al. (2025) provides strong empirical evidence that, for non-cryptographic watermarks, simple polynomial-time tests already operate near our theoretical boundary. Their "black-box" detectors are PPT algorithms that query the model only as an oracle, without access to internal logits or watermark parameters. Nevertheless, these detectors successfully identify several non-cryptographic watermark families within our probability-modifying framework.

For *biased* red-green schemes, they construct prompts whose continuations concentrate mass on a small controlled vocabulary and apply a permutation test to the empirical token frequencies across many queries. This test exploits exactly the per-token mean shift appearing in our KL expansion for biased sampling, and their reported sample complexities are consistent with the $O(|\delta|\sqrt{T})$ total-variation scaling in Theorem 1 together with the $TD_0 \gtrsim \log(1/\beta)$ condition of Theorem 2 at $\varepsilon = 0$.

For *bias-free* schemes such as fixed-sampling and cache-augmented watermarks, they employ rank-based and contingency-table tests to detect saturation effects and distributional shifts in repeated queries to the same prompt. These statistics capture higher-order diversity and variance anomalies, precisely the phenomena reflected in the variance term of our bias-free TV bound and in the per-token information $D_0$ of Theorem 2.

In all cases, the detectors of Gloaguen et al. (2025) are realizable by PPT adversaries and achieve near-perfect AUROC once the total number of queried tokens $T_{\text{tot}}$ is large enough that $T_{\text{tot}} D_0$ exceeds a modest constant (on the order of $\log(1/\beta)$ for the reported false-negative levels). This behavior aligns with the qualitative predictions of our information-theoretic analysis: the available statistical signal grows linearly with the number of biased or variance-perturbed tokens, and a simple polynomial-time test suffices to exploit it. Conversely, their detectors do not succeed against schemes that are distribution-preserving at the text level, consistent with our $\text{TV} = 0$ corner.

We emphasize that Gloaguen et al. (2025) do not claim minimax optimality, and our theorems do not specify the exact constants in the KL-based error exponents. Nonetheless, their empirical results strongly support the conclusion that, for the non-cryptographic biased and bias-free watermark families considered in Theorems 1 and 2, there is *no meaningful statistical-computational gap*: practical PPT detectors already operate close to the detectability and (zero-edit) robustness limits prescribed by our information-theoretic framework. Robustness under paraphrasing and structured edit channels is not addressed in Gloaguen et al. (2025); in our terminology, their experiments probe the $\varepsilon = 0$ slice of Theorem 2, while our analysis additionally characterizes how the information budget degrades as the edit rate $\varepsilon$ increases.

### D.12 RELATION TO CLASSICAL CODING-THEORETIC BOUNDS

The scaling $C(\varepsilon) \approx T(1-\varepsilon)^2 D_0$ in Theorem 2 should not be interpreted as a new capacity formula for generic substitution or insertion-deletion channels. Classical coding results Haeupler & Shahrasbi (2018); Yasunaga (2024) for the edit channels study the maximal message rate achievable when the encoder is free to choose arbitrary codewords, with no requirement that codewords resemble natural language or draw from a fixed source distribution, and without any constraint on how close the code-induced distribution must remain to a given baseline. In that regime, one can construct insertion-deletion codes with rate $1 - O(\varepsilon)$, and derive upper bounds of the form $(1 - H(\varepsilon))/(1-\varepsilon)$ for binary edit channels by optimizing directly over unconstrained codebooks.

Our setting is fundamentally different and tailored to LLM watermarking. First, all "codewords" are constrained to lie in the typical set of a fixed base model $P$: watermarked outputs are obtained by perturbing the *sampling rule* over $P$, not by freely selecting arbitrary sequences in $\Sigma^T$. Second, the perturbation is required to satisfy a stealth constraint, expressed as a small per-token KL drift $D_0$ between the watermarked sampler and the baseline sampler (equivalently, small total variation between the induced sequence distributions for untrusted observers). Third, the verifier's task is not to recover an arbitrary message, but to solve a binary hypothesis test $H_0 : P$ versus $H_1 : Q_k$ for a fixed provider after edits. In this constrained regime, the relevant quantity is the *sequence-level* KL divergence between the edited watermarked and unwatermarked distributions, which in the small-signal limit factorizes as

$$C(\varepsilon) \approx T(1-\varepsilon)^2 D_0. \tag{122}$$

Theorem 2 is therefore novel in that it translates the classical Chernoff–Stein KL criterion for hypothesis testing into an explicit *stealth-constrained* robustness bound for sampling-based LLM wa-

termarks. It pinpoints how a designer-chosen stealth budget (fixing $D_0$ via how close $Q_k$ must remain to $P$ at each token) jointly limits the tolerable edit rate $\varepsilon$ and the achievable power of *any* detector, independent of its computational resources. Classical insertion-deletion capacity results describe what is possible with unconstrained codebooks; Theorem 2 instead characterizes the detectability–robustness frontier in the much more restrictive and practically relevant setting where codewords must look like typical LLM text and watermarking is implemented only via small, distribution-constrained changes to the sampler.

# E    PROOF OF THEOREM 3

All logarithms are base 2, so every divergence and information quantity is measured in bits. The proof is organized into several stages, each of which builds toward the statement of the theorem. We begin with the local information contributed per token by biased and bias-free watermarking families. We then quantify the attenuation introduced by the substitution edit channel and extend this to sequences using the KL chain rule. We next invoke the Chernoff–Stein lemma to obtain a sufficiency condition for reliable detection. After this, we translate stealth requirements into information caps using Pinsker's inequality. Finally, we combine these pieces into the composite loss, which determines the optimal operating point, and analyze how the allocation between families should be made. The proof concludes by identifying conditions under which distribution-preserving watermarking strictly dominates.

## E.1    PER-TOKEN INFORMATION IN THE SMALL-SIGNAL REGIME

We begin with the biased (tilt) family. At a given position with baseline conditional distribution $p_t$ over the vocabulary $\Sigma$, a key-selected subset $G \subseteq \Sigma$ with baseline mass $\gamma = \sum_{v \in G} p_t(v)$ is exponentially tilted with parameter $\delta \in \mathbb{R}$. This produces the conditional

$$q_{t,\delta}(v) \;=\; \frac{p_t(v)\, e^{\delta\, \mathbf{1}[v \in G]}}{Z_t(\delta)}, \qquad Z_t(\delta) = (1 - \gamma) + \gamma e^{\delta}. \tag{123}$$

Expanding $\log_2(q_{t,\delta}(v)/p_t(v))$ around $\delta = 0$ and retaining the leading nonzero term gives

$$D\big(q_{t,\delta} \,\|\, p_t\big) = \frac{\gamma(1 - \gamma)}{2 \ln 2}\, \delta^2 + O(\delta^3). \tag{124}$$

Thus, the small-signal per-token information is

$$D_0^{\mathrm{B}} \;\approx\; \frac{\gamma(1 - \gamma)}{2 \ln 2}\, \delta^2, \tag{125}$$

which is maximized at $\gamma^\star = \frac{1}{2}$ for fixed $D_0$.

For the bias-free family, the watermarked conditional is a mean-one reweighting $q_{t,E}(v) = p_t(v) R_E(v)$ with $\mathbb{E}[R_E(v)] = 1$. Writing $R_E(v) = 1 + \Delta_E(v)$ and expanding $\log(1 + \Delta_E(v))$ shows that the quadratic variance term dominates, yielding

$$D_0^{\mathrm{BF}} \;\approx\; \frac{\hat{\sigma}^2}{2 \ln 2}, \qquad \hat{\sigma}^2 = \sum_v p_t(v)\, \mathrm{Var}[R_E(v)]. \tag{126}$$

## E.2    ATTENUATION UNDER EDITS

Each token passes through the substitution channel

$$T_\varepsilon(P) \;:=\; (1 - \varepsilon)P + \varepsilon U, \tag{127}$$

where $U$ is uniform on $\Sigma$. If $q = p + r$ with $\sum_v r(v) = 0$, then $T_\varepsilon(q) - T_\varepsilon(p) = (1 - \varepsilon)r$. Since KL divergence is locally quadratic in $r$, the attenuation factor is squared, giving

$$D\big(T_\varepsilon(q) \,\|\, T_\varepsilon(p)\big) = (1 - \varepsilon)^2\, D(q\|p)\, (1 + o(1)). \tag{128}$$

Consequently, for either family the per-token information after edits is

$$D_\varepsilon \;\approx\; (1 - \varepsilon)^2 D_0. \tag{129}$$

### E.3 SEQUENCE-LEVEL INFORMATION ACCUMULATION

The KL chain rule extends the per-token information to sequences. Writing $p_{t,\varepsilon} = T_\varepsilon(p_t)$ and $q_{t,\varepsilon} = T_\varepsilon(q_t)$, one obtains

$$D\left(\prod_{t=1}^{T} q_{t,\varepsilon} \,\Big\|\, \prod_{t=1}^{T} p_{t,\varepsilon}\right) \leq \sum_{t=1}^{T} D(q_{t,\varepsilon}\|p_{t,\varepsilon}). \tag{130}$$

In the homogeneous small-signal regime each summand is approximately $D_\varepsilon$, so the total usable signal is

$$C(\varepsilon) \approx T(1-\varepsilon)^2 D_0. \tag{131}$$

### E.4 RELIABILITY REQUIREMENT VIA CHERNOFF–STEIN

A level-$\alpha$ Neyman–Pearson test achieves miss probability at most $\beta$ if the sequence-level KL under the alternative exceeds $\log_2(1/\beta)$. Combining this condition with equation 131 gives

$$D_0 \geq D_{\text{req}}(\varepsilon, T, \beta) := \frac{\log_2(1/\beta)}{T(1-\varepsilon)^2}. \tag{132}$$

This inequality captures the robustness requirement: a minimum information budget per token is needed to guarantee detection.

### E.5 STEALTH CONSTRAINTS VIA PINSKER

Pinsker's inequality in nats yields $\text{TV} \leq \sqrt{D_{\text{nat}}/2}$, and converting bits to nats gives $D_{\text{nat}} = M(\ln 2)D_0$ for $M$ pooled tokens. Thus,

$$\text{TV} \leq \sqrt{\tfrac{\ln 2}{2} M D_0}. \tag{133}$$

Imposing a budget $\text{TV} \leq \tau$ leads to the stealth cap

$$D_0 \leq D_{\text{stealth}}(M, \tau) := \frac{2\tau^2}{M \ln 2}. \tag{134}$$

### E.6 MINIMIZATION OF THE COMPOSITE LOSS

The composite loss is

$$\mathcal{L}(\theta; \varepsilon, M, \tau) = \lambda_r [\log_2(1/\beta) - T(1-\varepsilon)^2 D_0(\theta)]_+ + \lambda_q \, \text{TV}_{\text{pen}}(D_0(\theta); M) + \lambda_a \, \text{Amp}(\theta). \tag{135}$$

Because the hinge vanishes once $D_0$ reaches the required threshold, while both detectability and amplitude penalties increase with $D_0$, the optimizer must select the smallest feasible $D_0$. This gives

$$D^\star = \min\{D_{\text{stealth}}(M, \tau), D_{\text{BF}}^{\text{max}} + D_{\text{B}}^{\text{max}}\}, \qquad D^\star \geq D_{\text{req}}(\varepsilon, T, \beta). \tag{136}$$

If this inequality cannot be satisfied, reliable detection is impossible at the given edit rate.

### E.7 OPTIMAL ALLOCATION BETWEEN FAMILIES

With $D^\star$ fixed, the TV penalty depends only on its value, not on the split between families. Hence the allocation minimizes the amplitude term. Since

$$\hat{\sigma}^2 = 2\ln 2 \, D_0^{\text{BF}}, \qquad \delta^2 = 8\ln 2 \, D_0^{\text{B}} \quad (\gamma = \tfrac{1}{2}), \tag{137}$$

the amplitude penalty is

$$\lambda_a \left(\sqrt{2\ln 2}\sqrt{D_0^{\text{BF}}} + \sqrt{8\ln 2}\sqrt{D_0^{\text{B}}}\right). \tag{138}$$

This is minimized by maximizing the allocation to BF, subject to its budget. Therefore,

$$D_0^{\text{BF}\star} = \min\{D^\star, D_{\text{BF}}^{\text{max}}\}, \qquad D_0^{\text{B}\star} = D^\star - D_0^{\text{BF}\star}. \tag{139}$$

The corresponding parameter values are

$$\hat{\sigma}^{2\star} = 2\ln 2 \, D_0^{\text{BF}\star}, \qquad \delta^\star = \sqrt{8\ln 2 \, D_0^{\text{B}\star}}, \qquad \gamma^\star = \tfrac{1}{2}. \tag{140}$$

If $D_{\text{req}}(\varepsilon, T, \beta) \leq D_{\text{BF}}^{\text{max}}$, the optimizer chooses pure BF; otherwise BF is saturated and the remainder is realized with B.

### E.8 Dominance of distribution-preserving watermarking

Finally, we examine when distribution-preserving watermarking is preferable. Suppose $K$ positions are marked and the verifier corrects up to $t$ errors. If $X \sim \text{Binomial}(K, 1 - \varepsilon)$ counts surviving marks, then

$$\Pr[X < K - t] \leq \exp\left(-2K\left((1-\varepsilon) - t/K\right)^2\right). \tag{141}$$

Thus DP achieves miss probability at most $\beta$ whenever

$$(1 - \varepsilon) \geq \frac{t}{K} + \sqrt{\frac{\ln(1/\beta)}{2K}}. \tag{142}$$

Because DP leaves the token distribution unchanged, it yields zero detectability and, therefore, strictly dominates any statistical scheme meeting the same robustness target. In this region, DP is optimal; outside of it, the statistical allocation of equation 139 applies.

### E.9 Conclusion

Combining the small-signal identities equation 125–equation 129, the sequence accumulation equation 131, the reliability requirement equation 132, the stealth cap equation 134, the composite loss equation 135, the allocation rule equation 139, and the DP dominance condition equation 142 establishes the full structure of the hybrid watermarking strategy and completes the proof of Theorem 3.

### E.10 Practical edit-rate estimation for the hybrid watermark

Theorem 3 is formulated as a design-time result: given an anticipated edit regime, it prescribes the smallest per-token information budget $D_0$ and the corresponding allocation between distribution-preserving (DP), bias-free (BF), and biased (B) components that achieve a target power $1 - \beta$ while respecting a stealth constraint. The parameter $\hat{\varepsilon}$ that appears in the loss $L(\theta; \hat{\varepsilon}, M, \tau)$ in Section 4 should therefore be understood as a design-time estimate of the edit rate in the deployment environment, rather than a per-sample quantity computed at inference time. In typical applications, the edit process is part of the system configuration (e.g., human-in-the-loop editing, a fixed paraphraser, or a known downstream model) and is stable over time, making $\hat{\varepsilon}$ a property of the channel rather than of individual texts.

#### E.10.1 Offline calibration procedure

We recommend estimating $\hat{\varepsilon}$ via a short calibration phase and then fixing the hybrid watermark parameters accordingly. A simple procedure is as follows:

1. **Collect calibration pairs.** For each application, sample prompts from the target distribution and generate baseline watermarked outputs $y_i$ of length $T_i$ using a provisional configuration.
2. **Apply the editing process.** Pass $y_i$ through the expected editing pipeline (e.g., paraphrasing model, summarizer, or representative human post-editing workflow) to obtain edited outputs $\tilde{y}_i$.
3. **Estimate the edit-rate distribution.** Compute empirical token-level edit rates

$$\varepsilon_i := \frac{\text{ED}(y_i, \tilde{y}_i)}{T_i}, \tag{143}$$

   where ED denotes the token edit distance. Aggregate $\{\varepsilon_i\}$ to obtain a distribution over edit rates (e.g., by computing mean, median, and upper quantiles such as the 90th or 95th percentile).
4. **Choose a design edit range.** Select a conservative design interval $[\varepsilon_{\min}, \varepsilon_{\max}]$; for example, $\varepsilon_{\max}$ can be the 95th percentile of the observed $\varepsilon_i$ values, and $\varepsilon_{\min}$ the median or a lower quantile.
5. **Compute the required information budget.** For the chosen power target $1 - \beta$ and typical length $T$, use Theorem 2 to compute the required per-token information at the worst-case edit rate in the range:

$$D_{\text{req}}(\varepsilon_{\max}, T, \beta) = \frac{\log_2(1/\beta)}{T(1 - \varepsilon_{\max})^2}. \tag{144}$$

6. **Instantiate the hybrid watermark.** Treat $D_{\mathrm{req}}$ as the target $D^\star$ and apply Theorem 3 to obtain the optimal allocation between BF and B (and the decision whether DP suffices). The resulting parameters $(\sigma^{2\star}, \delta^\star, \gamma^\star)$ are then *fixed* for deployment in that application.

In this workflow, $\hat{\varepsilon}$ is a design-time summary of the empirical edit-rate distribution (e.g., $\hat{\varepsilon} = \varepsilon_{\max}$), and the watermark sampler at inference time does not require access to any per-sample edit-rate estimate.

### E.10.2 Static versus coarse-grained adaptive deployment

The above procedure yields a static configuration: for a given application and editing pipeline, a single hybrid watermark is selected and reused across all generations. For deployments that operate across heterogeneous editing environments, the same calibration can be performed per environment to create a small menu of configurations (e.g., "low-noise" DP-heavy, "moderate-noise" BF-only, and "high-noise" BF+biased). At runtime, the system selects a configuration based on coarse metadata (such as which rewriting tool is invoked), rather than attempting to infer $\varepsilon$ from the final text.

This clarifies that Theorem 3 is agnostic to the specifics of the estimation procedure: any method that provides a reasonable bound on the edit regime can be plugged into the design equations. Our guarantees are monotone in $\varepsilon$: designing for a conservative $\hat{\varepsilon}$ ensures that the resulting hybrid watermark remains valid for any smaller actual edit rate, at the cost of potentially using a slightly stronger signal than strictly necessary.

## F  Watermarks beyond sampling

Our formal theorems are proved for watermarking schemes that inject their signal through the *sampling rule*, but the underlying notion of detectability does not depend on how the watermark is implemented. As defined in Section 3, detectability is the total variation distance between the distribution $P$ of texts produced by an unwatermarked model and the distribution $Q$ produced by a watermarked one, under a fixed prompting distribution. This quantity captures the optimal distinguishing advantage of any black-box test on generated text. Accordingly, weight-embedded or structural watermarks such as GaussMark Block et al. (2025) fit naturally within the same two-hypothesis framework whenever they induce an output distribution $Q_{\theta+\xi}$ that differs from the original $P_\theta$. In such cases, black-box detectability depends only on the divergence between $P_\theta$ and $Q_{\theta+\xi}$, regardless of whether this divergence arises from sampling-time biasing or parameter perturbations.

Within this unified perspective, our information-theoretic quantities have a model-agnostic interpretation. The relevant detectability parameter is the per-token KL signal

$$D_0 \equiv \mathbb{E}[D(q_t \| p_t)], \tag{145}$$

where $p_t$ and $q_t$ denote the unwatermarked and watermarked next-token distributions. For structural or weight-based schemes, $D_0$ represents the average effect of the parameter perturbation on the conditional distribution $p_{\theta+\xi}(y_t \mid y_{<t}, x)$. Thus, as long as a watermarking mechanism produces a nonzero $D_0$ in the output distribution, our detectability results (Theorem 1) apply at the level of text distributions.

For robustness, Theorem 2 and the attenuation law $D_\varepsilon \approx (1-\varepsilon)^2 D_0$ are derived under a *text-level* corruption model acting on tokens (edits, paraphrases, etc.). Any watermark, whether sampling-based or structural, falls under this analysis when the adversary acts only on outputs and the induced corruption can be modeled by our edit channel. However, our current theory does not address robustness of structural watermarks to *parameter-space* transformations such as fine-tuning, additional training, adapter layers, distillation, pruning, or interpolation, where the transformation is applied to $\theta$ before generation rather than to the generated text. Capturing such attacks would require an explicit model of noise or transformations in parameter space, which lies beyond our present scope.

In summary, our results characterize (i) black-box detectability as measured from text distributions and (ii) robustness to text-level corruptions modeled by an edit channel. Extending the same information-theoretic approach to parameter-space robustness for weight-based watermarks remains a natural direction for future work.

### F.1 EXTENSION TO TRAINING-TIME WATERMARKS

Recent work Gu et al. (2024); Block et al. (2025) has explored embedding watermarks directly into model parameters during training, enabling the resulting model to generate watermarked text under standard decoding without explicit modifications to the sampling rule. From our perspective, any such scheme fits into the same two-hypothesis framework via the induced text distributions. If the trained model with parameters $\theta + \xi$ produces next-token conditionals $q_t(\cdot) = p_{\theta+\xi}(\cdot \mid x, y_{<t})$ that differ from the baseline $p_t(\cdot) = p_\theta(\cdot \mid x, y_{<t})$, then detectability and tolerance to text-level edits are governed by the per-token signal, and the capacity evaluation does not change. Conversely, if a particular parameter change leaves $q_t \approx p_t$ for typical prompts, then $D_0 \approx 0$ and there is essentially no text-level watermark signal for any black-box detector on outputs to exploit, regardless of how the watermark is implemented internally.

However, our current framework does not explicitly model how subsequent parameter-space operations, such as continued training, fine-tuning, distillation, pruning, or interpolation, affect the pair $(p_t, q_t)$ and, consequently, the induced $D_0$ over time. For weight-based schemes, questions such as how long the watermark survives further training or how to trade the watermark signal against task performance depend on these parameter dynamics rather than on the edit channel acting on outputs. This provides a complementary setting in which combining training-time embeddings with inference-time mechanisms may yield improved trade-offs, and it motivates future extensions of our information-theoretic analysis to explicit models for structural watermarks.

### F.2 MULTI-KEY SCENARIOS

Throughout Theorem 1, we analyze detectability for a fixed watermark key, so that the watermarked sampler induces a single distribution $Q$ over sequences. If a scheme employs multiple keys $k \in \mathcal{K}$, and an adversary without the key observes outputs generated under different keys, then their effective observation model is the key-averaged distribution

$$\overline{Q} = \mathbb{E}_k[Q_k], \tag{146}$$

where $Q_k$ is the sequence distribution induced by key $k$. Detectability is always defined with respect to the distribution actually accessible to the adversary: for a fixed but unknown key, this is $Q_k$, whereas under re-keying across queries, it is $\overline{Q}$. In particular, Theorem 1 continues to apply with $Q$ replaced by $\overline{Q}$. For distribution-preserving schemes, we have $Q_k = P^s$ for every key, and hence $\overline{Q} = P^s$, yielding zero total variation even in the multi-key setting. For probability-modifying schemes (biased or bias-free), the mixture $\overline{Q}$ generally remains separated from $P^s$, so the same TV and KL-based detectability bounds apply after this substitution.

## G ADDITIONAL EXPERIMENTAL RESULTS AND DISCUSSION

For each base model in Table 1 (Llama 2 7B and Mistral 7B), we evaluate three editing conditions and then measure detection strength and third-party detectability for each watermarking scheme. The two paraphrasing conditions apply DIPPER Krishna et al. (2023) with a token editing rate of $\epsilon = 0.25$ and OPT 2.7B, prompted with "Rewrite the following paragraph:" with an average $\epsilon = 0.15$, which induces higher and lower token changes, respectively. For every condition, we report detection metrics with access to the key (area under the ROC curve, TPR at 1% FPR, and F1 at 1% FPR) and detectability metrics without the key using p-score and z-score from black box statistical tests Gloaguen et al. (2025); Liu et al. (2025).

We evaluate the following families and instances: Biased (KGW Kirchenbauer et al. (2023), Unigram Zhao et al. (2023)), Bias free (DiPMark Wu et al. (2024), HCW Hu et al. (2024)), and distribution preserving CGW Christ et al. (2024), along with our Optimal Hybrid (Theorem 3). This setup places each watermarking scheme at a point on the plane that balances detection strength against detectability, revealing how that point moves as edit intensity changes under using DIPPER and OPT 2.7B paraphrasing attacks.

Across both models, the detection–detectability tradeoff primarily depends on the watermarking family, rather than the underlying LLM. In the no-paraphrasing condition (reference), all methods achieve near-perfect detection strength; however, detectability differs markedly: CGW sits near the low detectability corner, KGW and Unigram are easily flagged statistically, and DiPMark and HCW

Table 1: Performance evaluation of Biased (KGW Kirchenbauer et al. (2023), Unigram Zhao et al. (2023)), Bias-free (DiPMark Wu et al. (2024), HCW Hu et al. (2024)), and undetectable CGW Christ et al. (2024) watermarking schemes on Llama-2-7B and Mistral-7B. For all cases, we evaluate robustness metrics (Reliable detection with key in presence of noise): AUROC, TPR at 1% FPR, and F1 at 1% FPR. We also evaluate detectability metrics (detection without key using statistical tests) via p-score and z-score.

| Model | Attack | Method | Robustness (with key) | | | Detectability (no key) | |
|---|---|---|---|---|---|---|---|
| | | | AUROC | TPR@1% | F1@1% | p-score[a] | z-score[b] |
| Llama-2-7B | Reference (no paraphrasing) | KGW (Biased) | 0.99 | 1.000 | 0.995 | 0.72 | 30.1 |
| | | Unigram (Biased) | 0.99 | 1.000 | 0.995 | 0.68 | 11.2 |
| | | DiPMark (Bias-free) | 0.99 | 1.000 | 0.995 | 0.31 | 43.2 |
| | | HCW (Bias-free) | 0.99 | 1.000 | 0.995 | 0.28 | 105.1 |
| | | CGW (Dist-pres.) | 0.99 | 1.000 | 0.995 | — | -5.8 |
| | | **Optimal Hybrid**[c] | 0.99 | 1.000 | 0.995 | — | -7.8 |
| | DIPPER (avg $\epsilon = 0.25$) | KGW (Biased) | 0.860 | 0.640 | 0.780 | 0.72 | 9.6 |
| | | Unigram (Biased) | 0.875 | 0.665 | 0.795 | 0.68 | 8.8 |
| | | DiPMark (Bias-free) | 0.895 | 0.800 | 0.865 | 0.31 | 3.9 |
| | | HCW (Bias-free) | 0.905 | 0.820 | 0.875 | 0.28 | 3.4 |
| | | CGW (Dist-pres.) | 0.500 | 0.150 | 0.230 | — | -10.2 |
| | | **Optimal Hybrid**[c] | 0.910 | 0.835 | 0.885 | — | 5.7 |
| | OPT-2.7B (avg $\epsilon = 0.15$) | KGW (Biased) | 0.780 | 0.590 | 0.720 | 0.72 | 8.4 |
| | | Unigram (Biased) | 0.790 | 0.615 | 0.740 | 0.68 | 7.9 |
| | | DiPMark (Bias-free) | 0.905 | 0.855 | 0.900 | 0.31 | 3.6 |
| | | HCW (Bias-free) | 0.920 | 0.880 | 0.915 | 0.28 | 3.1 |
| | | CGW (Dist-pres.) | 0.502 | 0.310 | 0.420 | — | -5.4 |
| | | **Optimal Hybrid**[c] | 0.930 | 0.895 | 0.922 | — | 4.5 |
| Mistral-7B | Reference (no paraphrasing) | KGW (Biased) | 0.99 | 1.000 | 0.995 | 0.69 | 27.8 |
| | | Unigram (Biased) | 0.99 | 1.000 | 0.995 | 0.66 | 10.5 |
| | | DiPMark (Bias-free) | 0.99 | 1.000 | 0.995 | 0.34 | 39.5 |
| | | HCW (Bias-free) | 0.99 | 1.000 | 0.995 | 0.26 | 98.7 |
| | | CGW (Dist-pres.) | 0.99 | 1.000 | 0.995 | — | -12.5 |
| | | **Optimal Hybrid**[c] | 0.99 | 1.000 | 0.995 | — | -11.0 |
| | DIPPER (avg $\epsilon = 0.25$) | KGW (Biased) | 0.845 | 0.615 | 0.765 | 0.71 | 9.0 |
| | | Unigram (Biased) | 0.860 | 0.640 | 0.780 | 0.67 | 8.2 |
| | | DiPMark (Bias-free) | 0.885 | 0.785 | 0.860 | 0.33 | 4.1 |
| | | HCW (Bias-free) | 0.895 | 0.805 | 0.872 | 0.29 | 3.5 |
| | | CGW (Dist-pres.) | 0.500 | 0.135 | 0.210 | — | -8.9 |
| | | **Optimal Hybrid**[c] | 0.902 | 0.820 | 0.880 | — | 7.6 |
| | OPT-2.7B (avg $\epsilon = 0.15$) | KGW (Biased) | 0.760 | 0.565 | 0.705 | 0.71 | 8.2 |
| | | Unigram (Biased) | 0.770 | 0.585 | 0.720 | 0.67 | 7.7 |
| | | DiPMark (Bias-free) | 0.890 | 0.840 | 0.890 | 0.32 | 3.8 |
| | | HCW (Bias-free) | 0.910 | 0.865 | 0.902 | 0.29 | 3.2 |
| | | CGW (Dist-pres.) | 0.501 | 0.285 | 0.400 | — | -9.7 |
| | | **Optimal Hybrid**[c] | 0.922 | 0.875 | 0.910 | — | 8.8 |

[a] p-score detectability metric reported by Gloaguen et al. (2025), which is watermark specific, hence left blank for CGW Christ et al. (2024) and proposed optimal hybrid watermarking scheme.

[b] z-score detectability metric reported by Liu et al. (2025), with negative score meaning less detectability.

[c] Proposed Pareto-optimal hybrid watermarking scheme by Theorem 3.

occupy the middle. Under DIPPER with average $\epsilon = 0.25$, CGW loses most of its detection strength, DiPMark and HCW maintain midrange values, and KGW and Unigram lie between these extremes; OPT 2.7B paraphrasing with average $\epsilon = 0.15$ causes a milder shift but preserves the same ordering. A single fixed family does not satisfy both needs over the full range of edits. In contrast, the Optimal Hybrid uses a simple estimate of edit intensity to select the active family, moving toward CGW when edits are light to keep detectability low and shifting toward HCW or KGW/Unigram as edits increase to keep high TPR at a fixed false positive rate. The empirical results align with our theory, and the closely matched trends for Llama and Mistral indicate that placement on the accuracy–detectability plane is driven by the watermarking type rather than the model type.

We additionally evaluate three GPT-3.5-based paraphrasing attacks Liu et al. (2025), including synonym substitution, adversarial watermark-removal, and back-translation from English to French and back, each configured to produce an average edit rate of approximately 15% in Table 2. Despite their different mechanisms and linguistic behaviors, synonym substitution and adversarial watermark-

Table 2: Performance of Biased (KGW, Unigram), Bias-free (DiPMark, HCW), and distribution-preserving (CGW), Optimal Hybrid watermarking schemes on Llama-2-7B under three GPT-3.5 paraphrasing attacks at 15% edit rate. Robustness: AUROC, TPR at 1% FPR, F1 at 1% FPR. Detectability: z-score (black-box, no key).

| Model | Attack | Method | Robustness (with key) | | | Detectability (no key) |
|---|---|---|---|---|---|---|
| | | | **AUROC** | **TPR@1%** | **F1@1%** | **z-score** |
| **Llama-2-7B** | GPT-3.5 Synonym-based (avg $\epsilon \approx 0.15$) | KGW (Biased) | 0.780 | 0.590 | 0.720 | 8.3 |
| | | Unigram (Biased) | 0.790 | 0.615 | 0.740 | 7.8 |
| | | DiPMark (Bias-free) | 0.905 | 0.855 | 0.900 | 3.5 |
| | | HCW (Bias-free) | 0.920 | 0.880 | 0.915 | 3.0 |
| | | CGW (Dist-pres.) | 0.502 | 0.310 | 0.420 | -5.5 |
| | | **Optimal Hybrid** | 0.930 | 0.895 | 0.922 | 4.4 |
| | GPT-3.5 mathrmAdversarial prompting ("remove watermark", avg $\epsilon \approx 0.15$) | KGW (Biased) | 0.775 | 0.585 | 0.715 | 8.1 |
| | | Unigram (Biased) | 0.785 | 0.610 | 0.735 | 7.6 |
| | | DiPMark (Bias-free) | 0.900 | 0.850 | 0.895 | 3.4 |
| | | HCW (Bias-free) | 0.918 | 0.878 | 0.912 | 3.0 |
| | | CGW (Dist-pres.) | 0.501 | 0.305 | 0.415 | -5.6 |
| | | **Optimal Hybrid** | 0.928 | 0.892 | 0.919 | 4.3 |
| | GPT-3.5 Back-translation (en→fr→en) | KGW (Biased) | 0.580 | 0.520 | 0.560 | -1.2 |
| | | Unigram (Biased) | 0.570 | 0.505 | 0.552 | -1.0 |
| | | DiPMark (Bias-free) | 0.600 | 0.540 | 0.575 | -0.8 |
| | | HCW (Bias-free) | 0.590 | 0.530 | 0.568 | -0.6 |
| | | CGW (Dist-pres.) | 0.505 | 0.210 | 0.350 | -6.5 |
| | | **Optimal Hybrid** | 0.605 | 0.548 | 0.582 | -0.5 |

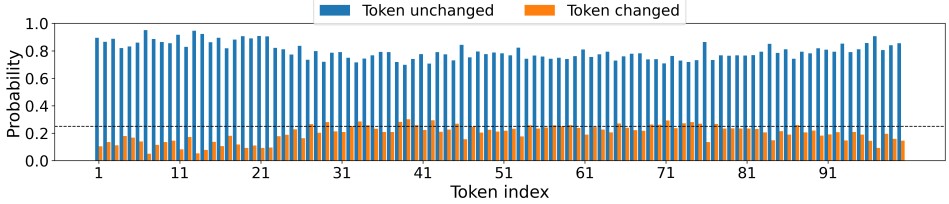

(a) DIPPER paraphrasing attack (average edit rate $\epsilon \approx 0.25$).

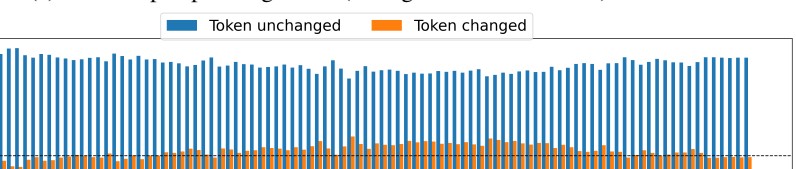

(b) Synonym–substitution attack (lexical substitution calibrated to $\epsilon \approx 0.15$).

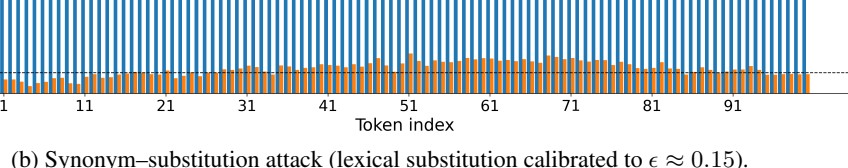

(c) Back–translation attack (en→fr→en).

Figure 3: Per–token change probabilities under different edit channels. For each attack type, we fix ten single 100-token watermarked Llama-2 outputs, apply the attack 10 times, and for every token index $t$, estimate the empirical probability that the token at position $t$ is left unchanged (blue) or modified (orange) across 10 iterations of the same prompt. The horizontal dashed line marks the global average edit rate $\epsilon$ for that attack for (a,b).

removal attacks yield a detectability-to-robustness profile that is close to that observed under OPT-2.7B paraphrasing with 15% edit rate. Once conditioned on the effective edit fraction $\varepsilon$, the relative placement of the watermarking schemes on the accuracy versus detectability plane remains unchanged. This supports our theoretical claim that the dominant factor governing robustness is the aggregate edit rate, rather than the specific structure of the edits.

However, the back-translation attack (English → French → English) operates without a preset edit rate, producing substantial semantic rewrites and correspondingly elevated token-level edit rates. This places it in the high-noise regime, where Theorem 2 predicts detection failure; indeed, all watermarking schemes exhibit poor detectability and robustness under this attack. These observations are consistent with the robustness-detectability trade-off landscape illustrated in Fig. 2(b), where high edit rates render reliable detection infeasible regardless of the watermarking scheme employed.

### G.1 EMPIRICAL ANALYSIS OF EDIT RATE

To assess the validity of the i.i.d. edit channel assumption, we conducted a token-level analysis of several representative paraphrasing attacks. We fixed ten 100-token watermarked LLaMA 2 generated texts and applied each attack type ten times, maintaining the global edit rate near a target value $\epsilon$ of approximately 0.25 for DIPPER and 0.15 for synonym substitution. A back-translation attack involves English-to-French and back-to-English translation without edit rate restrictions. For each token position $t$, we computed the empirical probability that the token at that position was modified.

Figure 3 shows, for each attack, we evaluate the per-position probability that a token remains unchanged (blue) or is modified (orange), with a horizontal line indicating the overall edit rate $\epsilon$ (if applicable). Fig. 3(a) demonstrates that DIPPER paraphrasing produces a nearly uniform profile: each position is edited with probability close to $\epsilon = 0.25$, exhibiting minor fluctuations, thus empirically capturing the channel edit model in Section 3. The synonym-substitution attack (Fig. 3(b)) shows somewhat greater variability and a slight increase in modification probability in the second half of the sequence, while back-translation (Fig. 3(c)) produces localized spikes but no systematic positional bias.

These measurements demonstrate that practical paraphrasers are not strictly i.i.d., as they introduce short correlated spans of edits. However, for paraphrasers that maintain an approximately constant edit rate across tokens (as observed for DIPPER and synonym substitution), our first-order i.i.d. substitution channel model in Theorem 2 provides a sound approximation, with higher-order semantic dependencies manifesting only as small deviations around the global edit rate. Back-translation presents a distinct case: its unconstrained semantic rewriting produces substantially higher edit rates, placing it in the high-noise regime where Theorem 2 predicts detection failure, consistent with the experimental observation that all watermarking schemes fail under this attack (Table 2).

## H WATERMARKING AS COVERT CHANNELS

Modern watermark detectors aggregate a small per-token statistical signal and then apply a Neyman-Pearson test to distinguish watermarked text from baseline text. The same statistical signal can be intentionally controlled to convey side information, thereby turning watermarking mechanisms into covert channels. Let $D_0$ denote the noise-free per-token information in bits per token contributed by a given watermark family. When the text passes through a substitution edit channel with edit rate $\varepsilon$, this quantity contracts quadratically:

$$D_\varepsilon \approx (1 - \varepsilon)^2 D_0. \tag{147}$$

For a sequence of length $L$, the total detector signal available after edits is

$$C(\varepsilon) \approx L(1 - \varepsilon)^2 D_0. \tag{148}$$

In the biased green list family, where a subset of the vocabulary of baseline mass $\gamma$ is exponentially tilted by a factor $\delta$, a second-order expansion gives

$$D_0 \approx \frac{\delta^2 \gamma(1 - \gamma)}{2 \ln 2}. \tag{149}$$

In the bias-free family, where unbiased reweighting with variance $\hat{\sigma}^2$ is applied, the corresponding expression is

$$D_0 \approx \frac{\hat{\sigma}^2}{2 \ln 2}. \tag{150}$$

A level $\alpha$ Neyman Pearson test achieves miss probability at most $\beta$ whenever

$$L(1 - \varepsilon)^2 D_0 \geq \log_2\left(\frac{1}{\beta}\right). \tag{151}$$

Table 3: Covert use of common watermark families (width safe TabularX)

| Family | How Alice encodes | How Bob decodes |
|---|---|---|
| Greedy tie breaking | Use a pre-shared set of tie-breaking keys. Select $k_m$ to represent message $m$ and add a small term $\epsilon \, \mathrm{PRF}_{k_m}(t, v)$ to logits when two top scores fall within a narrow band. | Given the prompt and model, simulate greedy decisions under each $k \in \mathcal{K}$ and choose the key that reproduces the observed tie outcomes. |
| Biased green list (tilt) | Partition $m$ into $w$ bit chunks and use each chunk to seed the green list in its window. Apply the exponential tilt $q_t \propto p_t \exp\{\delta \, \mathbf{1}[v \in G_t]\}$ with small $\delta$. | Regenerate the candidate green lists and evaluate the matched filter score of green token counts per window to select the chunk. |
| Bias-free (variance) | Use keyed permutations or codes $R_{E_t}$ with $\mathbb{E}[R_{E_t}(v)] = 1$ so that $\mathbb{E}[q_t] = p_t$ while the variance carries the information. | Apply the family-specific key verified statistic and perform maximum likelihood over the $2^w$ codes in each window. |
| Distribution preserving (PRF-driven RNG) | Replace $U_t \sim \mathrm{Uniform}[0, 1]$ by $U_t = \mathrm{PRF}_{k,m}(\mathrm{context}_t, t)$ and draw $v_t = F_{p_t}^{-1}(U_t)$, leaving one step marginals unchanged. | Resimulate the RNG stream with $k$ and candidate $\hat{m}$ and select the $\hat{m}$ that reproduces the observed sequence. |

Table 4: Message size scaling on length $L$ and edits at rate $\varepsilon$ (width safe TabularX)

| Family | Message size on length $L$ | Outsider evidence driver |
|---|---|---|
| Greedy tie breaking | $\log_2 |\mathcal{K}|$ once per document | Large sequence drift relative to the stochastic baseline. |
| Biased green list (tilt) | $\Theta(\sqrt{L})$ under a fixed outsider mixture budget (Lemma 8) | $D_0 \approx \delta^2 \gamma (1 - \gamma)/(2 \ln 2)$ and edited signal $L(1 - \varepsilon)^2 D_0$. |
| Bias free (variance) | $\Theta(\sqrt{L})$ under a fixed outsider mixture budget (Lemma 8) | $D_0 \approx \hat{\sigma}^2/(2 \ln 2)$ and edited signal $L(1 - \varepsilon)^2 D_0$. |
| Distribution preserving (PRF RNG) | $\Theta(L)$ in a single pass; repeated queries reveal determinism unless the seed is ephemeral | One step marginals match the baseline; a single pass outsider sees no local drift, but identical replays can expose determinism. |

Solving for the maximum admissible edit rate that still guarantees power $1 - \beta$ yields

$$\varepsilon_\beta(L, D_0) \;=\; 1 - \sqrt{\frac{\log_2(1/\beta)}{L D_0}}. \tag{152}$$

This expression shows that there is no universal critical edit rate; instead, performance depends jointly on $L$, $D_0$, and $\beta$.

### H.1 TURNING WATERMARK RULES INTO CHANNELS

Alice and Bob share a secret key $k$. During generation, Alice steers a standard probability-modifying watermark family to encode a message, and Bob decodes it using the matched key and verified statistics. An outsider observes only the text and is unaware of $k$. The constructions below are representative and capture the essential scaling laws. The receiver always applies the detector that is matched to the family and keyed to $k$.

## H.2 CAPACITY VERSUS DETECTABILITY: A SQUARE ROOT LAW

The following lemma (based on Theorem 2) formalizes the relationship between achievable message size and outsider evidence. It captures the square root scaling for biased and bias-free families under a realistic stealth requirement on the outsider mixture, and it clarifies the stronger constraint that arises if one demands small drift for every message separately.

**Lemma 8** (Capacity detectability law for watermark driven channels). *Let a watermark family contribute $D_0$ bits of information per token. A covert transmitter chooses a message $W \in \{1, \ldots, M\}$ uniformly and uses a secret key so that the outsider observes the mixture $Q = \frac{1}{M} \sum_{w=1}^{M} Q_w$. Then:*

(a) Mixture budget. *If the outsider mixture satisfies $D(Q\|P) \leq C_\star$ for a constant $C_\star$ independent of $L$, then for biased and bias free families*

$$\log M = \Theta(\sqrt{L}) \tag{153}$$

*in the noise-free case, and*

$$\log M = \Theta\big((1-\varepsilon)^2\sqrt{L}\big) \tag{154}$$

*under the substitution edit channel at rate $\varepsilon$.*

(b) Per message pooling. *If one imposes the stronger constraint $\mathrm{TV}(Q_w, P) \leq \tau$ for every message $w$, then Pinsker's inequality gives $D_0 \leq 2\tau^2/(L \ln 2)$ and hence*

$$\log M = O(1). \tag{155}$$

(c) Linear growth requires vanishing per token drift. *Any scheme that achieves $\log M = \omega(\sqrt{L})$ while keeping $D(Q\|P) \leq C_\star$ must satisfy $D_0 \to 0$ at the one-step margin, that is, it must be distribution preserving.*

## H.3 PROOF OF LEMMA 8

We first recall the small signal identities that underlie all bounds. For the biased family,

$$D_0 \approx \frac{\delta^2 \gamma(1-\gamma)}{2 \ln 2}, \tag{156}$$

and for the bias-free family,

$$D_0 \approx \frac{\hat{\sigma}^2}{2 \ln 2}. \tag{157}$$

Under the substitution channel, the per-token information contracts as

$$D_\varepsilon \approx (1-\varepsilon)^2 D_0, \tag{158}$$

so the total sequence level signal equals

$$C(\varepsilon) \approx L(1-\varepsilon)^2 D_0. \tag{159}$$

A level $\alpha$ Neyman Pearson test reaches miss probability at most $\beta$ once

$$L(1-\varepsilon)^2 D_0 \geq \log_2\left(\frac{1}{\beta}\right). \tag{160}$$

*Achievability under the mixture constraint.* Consider a sparse activity design. Fix $\theta_L = c/\sqrt{L}$ with $c > 0$. Using the secret key, mark each position active independently with probability $\theta_L$; inactive positions are sampled from the baseline. On active positions, apply a constant tilt $\delta = \delta_0$ and select the green list using successive message chunks. The outsider mixture at a given token is

$$(1 - \theta_L)p_t + \theta_L q_{t,\delta_0}, \tag{161}$$

and a second-order expansion gives its KL to $p_t$ as

$$D((1-\theta_L)p_t + \theta_L q_{t,\delta_0} \| p_t) = \frac{\theta_L^2}{2 \ln 2} \sum_v \frac{\big(q_{t,\delta_0}(v) - p_t(v)\big)^2}{p_t(v)} + o(\theta_L^2) \tag{162}$$

$$\approx \theta_L^2 \cdot \frac{\delta_0^2 \gamma_t(1-\gamma_t)}{2 \ln 2}. \tag{163}$$

Summing over $L$ tokens yields

$$D(Q\|P) \approx L\,\theta_L^2 \cdot \frac{\delta_0^2\,\overline{\gamma(1-\gamma)}}{2\ln 2} = O(1)\,, \tag{164}$$

since $L\theta_L^2 = c^2$ and $\delta_0$ is constant. Thus the mixture divergence remains bounded uniformly in $L$. Conditioned on the key there are $T = \theta_L L = c\sqrt{L}$ active positions. On each active position, the matched statistic provides a constant positive information increment $\kappa > 0$. Standard concentration for log likelihood ratios then gives reliable decoding, provided

$$\log M \le \frac{1}{2}\,T\,\kappa - \omega(1) = \Theta(\sqrt{L})\,. \tag{165}$$

Under edits at rate $\varepsilon$, each active increment contracts by $(1-\varepsilon)^2$, so the same argument yields

$$\log M = \Theta\big((1-\varepsilon)^2\sqrt{L}\big). \tag{166}$$

*Converse under the mixture constraint.* Let $Q_w$ denote the distribution induced by message $w$ and $Q = \frac{1}{M}\sum_w Q_w$ the outsider mixture. The mutual information satisfies

$$I(W; Y_{1:L}) = \frac{1}{M}\sum_{w=1}^{M} D(Q_w\|Q)\,. \tag{167}$$

The log sum inequality together with the small signal expansion that controls $D(Q\|P)$ implies that the average squared perturbation around $P$ is of order $1/L$, which limits the aggregate distinguishability across messages to order $\sqrt{L}$. A sphere packing bound for multi-hypothesis testing with total information budget of order $\sqrt{L}$, therefore, yields

$$\log M \le c_1\sqrt{L} + O(1)\,, \tag{168}$$

for a constant $c_1$ determined by the family and the map from watermark strength to $D_0$. The same $(1-\varepsilon)^2$ contraction applies under edits.

*Per message pooling constraint.* If for every $w$ one requires $\mathrm{TV}(Q_w, P) \le \tau$, then Pinsker and unit conversion imply

$$LD_0 \le \frac{2\tau^2}{\ln 2}\,, \tag{169}$$

so $D_0 = O(1/L)$ and any two messages have only a constant order separation across the entire text. Reliable decoding is then possible for at most a constant number of hypotheses, which proves the stated order.

*Distribution preserving case.* If $D_0 = 0$ at the one-step margin, for example, by replacing the RNG with a pseudorandom stream, then one can place one bit of seed-controlled entropy per token without changing one-step marginals. In a single-pass setting, this allows

$$\log M = \Theta(L)\,, \tag{170}$$

although repeated queries with the same seed reveal determinism unless the seed is refreshed, making this a pure covert channel rather than a forensic watermark.

## H.4 Reliability under edits: knee and AUROC

With total usable information

$$C(\varepsilon) \approx L(1-\varepsilon)^2 D_0\,, \tag{171}$$

the sufficiency condition for miss probability $\beta$ is

$$L(1-\varepsilon)^2 D_0 \ge \log_2\Big(\frac{1}{\beta}\Big), \tag{172}$$

and the corresponding knee is

$$\varepsilon_\beta(L, D_0) = 1 - \sqrt{\frac{\log_2(1/\beta)}{LD_0}}\,. \tag{173}$$

Beyond this point, the score distributions of the likelihood ratio test largely overlap, and the area under the ROC curve approaches $0.5$ with only finite sample fluctuations.

