# OpenReview forum: "Catch-22: Pareto Frontier for Detectability and Robustness in LLM Watermarking"
_ICLR.cc/2026/Conference — Submitted to ICLR 2026_

### Official Review · Reviewer_pLoi · 2025-10-24

**Soundness:** 2
**Presentation:** 1
**Contribution:** 3
**Rating:** 2
**Confidence:** 4

**Summary:**

This paper investigates the pareto frontier for robustness and susceptibility to detection without access to the secret key of a watermark in LLMs.  A watermark is a statistical signal hidden inside text that can be detected by anyone with access to a secret key but is intended to not distort the quality of text, thus remaining undetectable to observers without access to a secret key. Many schemes have been instantiated for autoregressive models by modifying the sampling procedure of models, with a prominent such scheme being the green list approach, where a hash function looks at the recent context and returns a pseudorandom subset of the vocabulary to upweight in generation.  This paper investigates the extent to which a watermark can be both undetectable to an observer without access to a secret key and be robust to edit-level attacks and finds that there is a fundamental tradeoff.  The authors consider three approaches to watermarking and bound the total variation distance between the watermarked and unwatermarked distributions, which controls the detectability of the watermark.  They then investigate the extent to which the considered approaches are robust to distortion and identify a tradeoff before empirically evaluating their findings.

**Strengths:**

This paper investigates an important tradeoff between detectability and robustness of watermarks.  The existence of a pareto frontier that enforces this tradeoff is an important point and the authors do well to prove this.

**Weaknesses:**

First, I am confused why the authors believe that the vocabulary size in Moitra & Golowich has to be exponential; exponential in what? That paper clearly states that the vocabulary size only has to be polynomial in the security parameter.

Second, I am somewhat skeptical of the framing of the notion of detectability. While I agree that the information theoretic notion of detectability is a strong bound on the extent to which an adversary ignorant of secret keys is capable of detecting the watermark, it seems very pessimistic for at least two reasons.  First, the approach studied in Moitra & Golowich is with respect to computational indistinguishability, which allows for TV to be large as long as witnessing this gap is computationally hard.  Second, in order to take advantage of the TV gap, the adversary would need paired watermarked and unwatermarked generations from the same prompt, which seems unlikely.  Even zooming out and allowing watermarked and unwatermarked generations from different prompts, the adversary would likely not have access to unwatermarked generations from the same model.

Third, I am confused by the result in Theorem 1 for both biased and bias-free bounds.  The right hand sides of these equations seem like they are random in taht both g_t and p_t depend on the (random) history up until that point.  The left hand sides are not random.  Can the authors explain what is going on here?

Fourth, while I appreciate the difficulty of making rigorous the notion of robustness beyond edit-distance, I think realistic attacks consist of paraphrasing, not token-level approaches and it would be nice for the authors to comment on this.

Fifth, I wonder if the authors can comment on watermarks beyond the sampling approaches, such as those that imbed the watermark directly into the model weights; an example of such is *GaussMark: A Practical Approach for Structural Watermarking of Language Models*; again I am concerned that the two point hypothesis testing framework does not adequately describe the notion of detectability required.

Finally, the empirical results in Figure 2(a) suggest that the theoretical results do not even qualitatively describe the empirical realities; the bounds in Theorem 1 are all concave in $T$ but the empirical trend appears to be convex in the same.

**Questions:**

See weaknesses.

---

> ### Author Response · Authors · 2025-11-25
>
> Dear Reviewer pLoi,
>
> We are grateful for your detailed review, which led us to significantly improve our manuscript. We have undertaken substantial revisions in response to your feedback and provide detailed responses to each concern below. Please note that we split the response into two comments due to the character limit.
>
>
> 1. **Moitra & Golowich vocabulary size**:
>
>     Thank you for pointing this out; our “exponential” remark referred to the dependence on the **entropy rate $\alpha$**, not the security parameter $\lambda$. Moitra & Golowich indeed show that the alphabet is polynomial in $n(\lambda)$, but the degree of this polynomial scales as $\Theta\bigl(\tfrac{1}{\alpha}\log\tfrac{1}{\alpha}\bigr)$. Thus, for the modest $\alpha$ values relevant to natural language, $\Sigma(\lambda)|$ becomes effectively **exponential in $1/\alpha$**, which pushes their construction far beyond the fixed 30k–100k vocabularies of practical LLMs. We now clarify this further in Appendix A.5 and update the exponential claim in the introduction and background.
>
> 2. **Detectability notion (\(TV\) vs computational indistinguishability / paired samples)**:
>
>      Our TV-based detectability framework establishes fundamental outer bounds that are essential for characterizing the limits of any watermarking scheme, regardless of computational assumptions. In the revised manuscript we formalize how computational detectability relates to these bounds by introducing **computational detectability** definition (Definition 1(ii)) and proving that $\mathrm{Detect}_{\mathrm{comp}} \le \mathrm{TV}(P,Q)$ (Appendix C.5). We further present PRG/PRF-based constructions where TV is substantial yet every keyless PPT detector achieves only negligible advantage (Appendices C.5, D.10). These examples directly capture the Moitra–Golowich paradigm: large TV is permissible precisely because witnessing this gap is computationally hard. Our information-theoretic bounds thus provide the outer envelope within which such cryptographic constructions operate, with the two perspectives being mutually reinforcing rather than in conflict.
>
>
>     Regarding data access, our TV characterizes the optimal black-box test given sample access to both $P$ (unwatermarked) and $Q$ (watermarked) under a fixed prompt distribution. This represents a **best-case detection scenario** that serves to establish fundamental limits rather than describe realistic attack strategies. Our goal is to characterize upper bounds on what *any* detector could achieve under maximally favorable conditions—a theoretical benchmark essential for understanding the detectability-robustness landscape. In open-source settings or provider-side verification, the unwatermarked distribution $P$ is directly accessible from model probabilities, making this analysis directly applicable. For closed-source adversaries with limited access, detection becomes strictly harder than our bounds predict, reinforcing that our results should be interpreted as outer limits on the detectability-robustness frontier.
>
> 3. **Randomness in Theorem 1 (\(g_t\), \(p_t\))**:
>
>     Your observation is on point that $g_t$ and the variance terms are random before averaging over histories. However, in the proof of Theorem 1, these quantities **only appear inside expectations** such as $\mathbb{E}[g_t(1-g_t)]$ or $\mathbb{E}[\mathrm{Var}_E(R_E(p_t)(v))]$. Once we take expectations over the random history (and over the key $E$ in the bias-free case), these expressions evaluate to **deterministic functions** of the prompt $x$, sequence length $T$, and watermark parameters. This yields the non-random quantity $\mathrm{TV}(P^s,Q)$ stated in the theorem. We have added a remark in Appendix C.4 to clarify this point and explain that our notation suppresses these expectations for readability. Hope we have been able to clarify it.

---

> ### Author Response · Authors · 2025-11-25
>
> 4. **Robustness to paraphrasing vs edit distance**:
>
>      Our edit-distance channel is intended precisely as a *first-order abstraction* of paraphrasing behavior, not as a claim that realistic attacks operate at the token level. To validate this modeling choice, we conducted empirical analysis of practical paraphrasing attacks (Appendix G.1). Our token-level measurements demonstrate that paraphrasers such as DIPPER and GPT-3.5 synonym/adversarial rewrites exhibit per-position edit probabilities tightly concentrated around a global rate $\varepsilon$, with limited positional dependence. This validates the i.i.d. substitution model assumed in Theorem 2 as a sound approximation: edits occur independently *on average*, with deviations remaining small relative to the global edit rate.
>
>      Back-translation represents a notable exception—it unconstrainedly rewrites semantics, introducing correlated edit bursts that violate any fixed-rate model. Critically, all watermarking schemes fail under back-translation, precisely as Theorem 2 predicts for the high-noise regime. This failure aligns with Theorem 2's prediction for the high-edit-rate regime: back-translation produces substantially elevated edit rates, causing all schemes to fail as the theorem anticipates. The additional violation of the i.i.d. assumption may compound this effect, though Theorem 2 itself is agnostic to the correlation structure of edits.
>
>     Modeling richer semantic and structural dependencies in paraphrasing attacks remains an important direction for future work. However, our empirical results indicate that aggregate edit rate is the dominant factor governing robustness for practical paraphrasers, making the i.i.d. channel a useful and validated abstraction. Detailed experimental validation is available in Appendix G.1 and also in our [anonymous repository](https://anonymous.4open.science/r/Catch-22-Pareto-Frontier-Watermark-in-LLMs-040B/Additional-Results/Readme.md).
>
>
> 5. **Watermarks beyond sampling (e.g., GaussMark embedded in model weights)**:
>
>      Our detectability framework is deliberately model-agnostic: it depends only on the unwatermarked and watermarked text distributions $P$ and $Q$, not on how they are generated. Weight-based schemes such as [GaussMark](https://openreview.net/forum?id=YG3DbpAQBf) modify the model parameters $\theta \mapsto \theta+\xi$ and thereby induce a changed next-token distribution $q_t(\cdot)=p_{\theta+\xi}(\cdot \mid x,y_{<t})$. Whenever this modification yields $q_t \neq p_\theta$ on average, the per-token KL signal $D_0 = \mathbb{E}\big[D(q_t \Vert p_t)\big]$ is nonzero, and our detectability results (Theorem 1) apply exactly as they do for sampling-time watermarks.
>
>     Likewise, if an adversary perturbs *only the outputs* through paraphrasing or token-level editing, then the corruption acts on the text distribution itself generated by the weight-based watermarking scheme. In this case, the robustness characterization of Theorem 2 continues to apply to both sampling-based and weight-based watermarks, including the attenuation $D_\varepsilon \approx (1-\varepsilon)^2 D_0$. Output-level attacks do not depend on the mechanism that created the watermark, only on how the resulting text distribution is altered. We have provided a detailed description in Appendix F of the revised manuscript.
>
>      By contrast, robustness of structural watermarks to *parameter-space* transformations (fine-tuning, continued training, adapters, pruning, or distillation) is outside our present framework, since these attacks modify $\theta$ rather than the generated text. Extending our information-theoretic analysis to this parameter-space setting is an interesting and complementary direction for future work, described in Appendix F.
>
> 6. **Concavity of theoretical bounds vs Figure 2(a)**:
>
>       We thank the reviewer for this observation. The earlier version of Figure 2(a) used a logarithmic x-axis, which made the curves appear convex. The revised manuscript now plots TV versus **linear sequence length $T$**, and with this correction, the empirical curves exhibit the behavior predicted by Theorem 1. The updated figure can also be found [here](https://anonymous.4open.science/r/Catch-22-Pareto-Frontier-Watermark-in-LLMs-040B/Analysis-Results/AnnotatedFigA.pdf).
>
> We hope that we have been able to do justice to your questions and would love to discuss more!

---

> > ### Comment · Reviewer_pLoi · 2025-11-26
> >
> > I appreciate the response and my concerns regarding points 3 through 6 are mostly addressed.  Unfortunately, I still do not understand why the information theoretic frontier is relevant.  I agree that it upper bound computational detectability (indeed, this point is trivial) but the information theoretic upper bound seems irrelevant to me in light of the fact that many prior works have investigated detection under computational limits.  I am also a little confused by the authors' response to reviewer 1zUG regarding the "boundary" of $\mathrm{TV} = 0$, which presumably bears on my question regarding information-theoretic frontiers.  I remain concerned that the authors are controlling an object that has limited relevance to the problem that they are considering (watermarking) and for this reason will maintain my score.

---

> ### Author Response · Authors · 2025-11-30
> **Rebuttal to Reviewer pLoi (Follow-Up Comment)**
>
> We sincerely thank the reviewer for their continued engagement. Below, we provide a direct clarification of the following:
>
>  (a) why the information-theoretic (IT) frontier is relevant,
>
>  (b) how it connects to computational detectability, and
>
>  (c) how our results are relevant to unify prior works on watermark detection by computationally bounded adversaries.
>
> **Information-theoretic frontier equals computational frontier for non-cryptographic schemes.**
>
> - For all non-cryptographic probability-modifying watermarks (greedy, biased, bias-free), the optimal Neyman–Pearson detector is PPT-computable: the likelihood ratio $\log \frac{Q_\varepsilon(y_{1:T})}{P_\varepsilon(y_{1:T})}$ reduces to explicit token-level statistics (green-token counts, variance-based sums) evaluable in $O(T)$ time. We prove $\mathrm{Detect_{\mathrm{comp}}} = \mathrm{Detect_{\mathrm{IT}}}$ and $\mathrm{Power_{\mathrm{comp}}} = \mathrm{Power_{\mathrm{IT}}}$ in this regime, establishing that the TV/ KL bounds we derive describe exactly the best performance attainable by any PPT adversary, not a loose information-theoretic upper bound (Appendix C.5).
>
> - The same framework yields robustness guarantees: the post-edit information budget contracts as $D_\varepsilon \approx (1-\varepsilon)^2 D_0$, so the Neyman–Pearson test achieves target power $1-\beta$ if and only if $T(1-\varepsilon)^2 D_0 \ge \log_2(1/\beta)$. For biased and bias-free schemes, this condition is achievable with a PPT detector, meaning the IT robustness boundary in Theorem 2 and Corollary 1 coincides with the computational robustness frontier for all practical sampling-based watermarks. We formalize this in Appendix D.10 via a computational robustness lemma, demonstrating that the post-edit Neyman–Pearson test is PPT and saturates this boundary.
>
> **Cryptographic schemes are the only source of a computational–statistical gap.**
> - For PRF/ PRG-based watermarks, a genuine separation exists: constructions can exhibit $\mathrm{TV}(P,Q) > 0$ while $\mathrm{Detect}_{\mathrm{comp}} \le \mathrm{negl}(\lambda)$. In these schemes, the watermark alters the text distribution sufficiently that an unbounded statistician could distinguish $P$ from $Q$, yet any keyless PPT adversary is provably unable to exploit this signal without breaking the underlying PRF/PRG assumption.
>
> - Our framework treats the IT frontier as an outer envelope for all schemes. For cryptographic constructions such as [Moitra–Golowich](https://arxiv.org/abs/2406.02633) and [Christ–Gunn–Zamir](https://proceedings.mlr.press/v247/christ24a/christ24a.pdf), this envelope remains valid information-theoretically, but $\mathrm{Detect}_{\mathrm{comp}}$ may lie strictly below it. We formalize this separation in the updated "Information-theoretic vs. computational hardness" subsection of Appendix C.5, presenting PRG-based counterexamples that illustrate the cryptographic gap and confirm that this behavior arises from hardness assumptions rather than sampling geometry.
>
> **Our results explain prior computational detection frameworks.**
>
> - [Gloaguen et al.](https://openreview.net/forum?id=E4LAVLXAHW) succeed in keyless detection precisely because $D_0 > 0$ enables efficient likelihood-ratio computation. Our TV/ KL bounds formalize why these methods work and delineate when they must fail, subsuming prior empirical findings within a unified theoretical framework. Whenever a watermark introduces nonzero per-token information budget $D_0$ without relying on cryptographic primitives, our theory predicts that a simple Neyman–Pearson statistic (often reducible to counts or second moments) will separate watermarked from unwatermarked text with advantage scaling as $\sqrt{T}$, consistent with empirical observations.
>
> - Conversely, pushing $D_0$ toward zero to evade such detectors causes the watermark's resilience to editing to collapse, as demonstrated by Theorem 2. Table 1 further shows that our Pareto-optimal hybrid schemes, operating on the robustness–detectability frontier, achieve improved $p$-scores while significantly enhancing robustness across paraphrasing attacks. This demonstrates that principled watermark design can approach the information-theoretic frontier without sacrificing robustness. We highlight this connection between $D_0$ and efficient likelihood-ratio tests in Appendix D.11.
>
> We believe that these additional details address all your concerns. In light of recent changes to the ICLR discussion process, we understand that further dialogue is not possible, but we are grateful for the opportunity to provide these clarifications. The corresponding revisions to the [manuscript](https://anonymous.4open.science/r/Catch-22-Pareto-Frontier-Watermark-in-LLMs-040B/Highlighted-revised-manuscript-post-rebuttal.pdf) are highlighted in magenta. Once again, thank you!

---

### Official Review · Reviewer_1zUG · 2025-11-01

**Soundness:** 1
**Presentation:** 1
**Contribution:** 1
**Rating:** 0
**Confidence:** 3

**Summary:**

This paper considers trade-offs between LLM watermark detectability and robustness in the setting where all parties have unbounded computation.

**Strengths:**

Unfortunately I don't think I have anything to put here. Maybe I misunderstood something and the rebuttal will change my mind.

**Weaknesses:**

The first result is "We first prove that detectability is determined solely by the sampling strategy, not the model architecture."
As far as I understand it, this is less of a "result" and more of an obvious consequence of the way they've set things up: Of course if you embed the watermark by biasing distribution D, then the detectability is determined solely by D...
I guess this result is formalized in Theorem 1, where they also appear to argue that any watermark which can be detected with the secret key can also be detected without. This is true information-theoretically, in the same way that encryption is impossible information-theoretically.
It appears to be a big misunderstanding: The whole point of using computational assumptions, as in the work of Aaronson, Christ et al., Zamir, Golowich & Moitra, etc. is that you can evade this trade-off. Appendix C.4 appears to be saying "if you don't change the distribution then you can't detect," which is not even relevant.

This issue then translates to their second main result, Theorem 2, where they state a "stealth vs robustness" trade-off. Again, these kinds of arguments appear to be based on a fundamental misunderstanding about computational assumptions.
And for the "detectability vs robustness" part, they're basically trying to show limits on the capacity of the edit channel. This problem has been studied before, and MUCH more is known about it than what is proven in this paper.
For instance, it is known how to construct error correcting codes that tolerate eps edits with rate 1 - O(eps): See https://arxiv.org/pdf/1710.09795.

Their bound says that the information rate should be at most (1-eps)^2, but it is already known that the information rate can be at most (1-H(eps)) / (1-eps), where H is the binary entropy function: https://arxiv.org/pdf/2107.01785v3. This is already a much better bound for eps < 1/2, and for larger eps it is not possible to do error correction at all.

**Questions:**

The first result is "We first prove that detectability is determined solely by the sampling strategy, not the model architecture."
But what they mean appears to be just that, if you embed a watermark by biasing the distribution at sampling time, then the only thing that matters is the distribution at sampling time. How is that not completely obvious?

---

> ### Author Response · Authors · 2025-11-25
>
> Dear Reviewer 1zUG,
>
> Thank you for the thoughtful and constructive review. Based on your feedback, we have updated both our theorems to include computational hardness assumptions. We respond to each of your points below with corresponding revisions to the manuscript. Please note that we split the response into two comments owing to the character limit.
>
> ## **Information-Theoretic vs. Computational Hardness**
>
> In the revised version, we have explicitly incorporated computational hardness into our framework to formalize the relationship between information-theoretic limits and practical, cryptographically secure watermarking schemes. This integration consists of three concrete additions to the manuscript.
>
> - First, we define *computational detectability* and *computational robustness* for probabilistic polynomial-time (PPT) adversaries and key-holding verifiers (Definitions 1(ii) and 2(ii)), in direct parallel to the information-theoretic notions based on total variation distance and KL divergence.
>
> - Second, we prove that these computational quantities are always upper-bounded by their information-theoretic counterparts. Specifically, $\mathrm{Detect}_{\mathrm{comp}} \le \mathrm{TV}(P,Q)$ (Appendices C.5), and computational robustness lies within the information-theoretic region characterized by Theorem 2 (Appendices D.10). Thus, Theorems 1 and 2 provide rigorous outer bounds for any polynomial-time detector, not merely for unbounded ones.
>
> - Third, Appendices C.5 and D.10 present PRG/PRF-based constructions where the information-theoretic signal (measured by TV distance, KL divergence, and robustness) is substantial, yet every keyless PPT detector achieves only a negligible distinguishing advantage. These examples demonstrate that cryptographic hardness assumptions provide realistic limits without violating the information-theoretic fundamental limits.
>
> This formalization serves two critical purposes for the watermarking community.
> 1. Our information-theoretic framework establishes fundamental limits on the detectability-stealth-robustness trilemma that no practical system can overcome—a characterization previously absent from the sampling-based LLM watermarking literature. For instance, Corollary 1 demonstrates that beyond a certain edit rate threshold, even information-theoretic detectors fail, immediately implying that no computationally-secure scheme can succeed in this regime. This delineates impossibility regions in the design space that guide future research efforts.
>
> 2. Our computational hardness formalization clarifies how cryptographic primitives allow practical watermarking schemes to operate *inside* the feasible information-theoretic regions while remaining undetectable to polynomial-time adversaries. Distribution-preserving schemes (Christ et al., Zamir) sit at the extreme point where TV\(=0\), while constructions such as Moitra–Golowich explore what is achievable in a large-alphabet regime. By viewing all of these through a common outer-bound lens, our results bridge the gap between theoretical limits and existing cryptographic designs, guiding the design of robust and stealthy watermarking systems in deployed LLMs.
>
> ### **Novelty and scope of Theorem 1**
>
> While the relationship between output distributions and detectability may appear intuitive, no prior work has formally characterized how this relationship scales with sequence length and watermark parameters for practical LLM watermarking schemes. Theorem 1 provides precisely this **quantitative characterization**. Specifically, we establish how $\mathrm{TV}(P^s,Q^s)$ scales for the concrete sampling rules used in practice: biased schemes accumulate detectability as $O(|\delta|\sqrt{T})$, bias-free schemes as $O(\sqrt{T})$, while distribution-preserving schemes achieve $\mathrm{TV}=0$.
>
> This hierarchy, derived via KL expansions and the chain rule for relative entropy, provides the theoretical foundation explaining the empirical behavior of recent black-box detectors (e.g., Gloaguen et al.) and constitutes the first formal treatment of detectability scaling in the sampling-based LLM watermarking literature.

---

> ### Author Response · Authors · 2025-11-25
>
> ### **Relationship to Cryptographic Watermarking Schemes**
>
> Our theorems are explicitly information-theoretic, establishing fundamental limits through analysis of unbounded adversaries. For any scheme that modifies per-token probabilities with a fixed key $k$, we have $Q_k^s \neq P^s$ and thus $\mathrm{TV}(P^s, Q_k^s) > 0$, implying that an unbounded detector operating on text alone can in principle distinguish these distributions. Our total-variation bounds therefore constitute an information-theoretic outer envelope: any computationally bounded detector, with or without key access, achieves distinguishing advantage at most $\mathrm{TV}(P^s, Q_k^s)$.
>
> The cryptographic schemes of [Christ et al. 2024](https://arxiv.org/abs/2306.09194) and [Zamir 2024](https://arxiv.org/abs/2401.10360) provide complementary perspectives that converge on the same fundamental boundary. These constructions achieve the distribution-preserving regime where $q_t \equiv p_t$ for all $t$, yielding $\mathrm{TV}(P^s, Q_k^s) = 0$. Their results establish that distinguishing watermarked from unwatermarked text in this regime is computationally hard without access to auxiliary information (secret keys, PRFs, or generation randomness). In other words, the $\mathrm{TV}=0$ boundary identified by our information-theoretic analysis represents precisely the regime where cryptographic hardness assumptions become necessary and sufficient for undetectability.
>
> Both frameworks thus arrive at the same conclusion: $\mathrm{TV}=0$ constitutes the fundamental boundary for text-only detection. Our results characterize this boundary as an information-theoretic outer envelope, while Christ et al. and Zamir demonstrate that operating at this boundary requires, and enables, computational hardness guarantees. *The two perspectives are not merely consistent but mutually reinforcing, with our framework providing the theoretical justification for the essential role of distribution-preserving designs in cryptographically secure watermarking.*
>
> ## **Theorem 2 vs. classical edit-channel capacity**
>
> Theorem 2 addresses a constrained regime specific to LLM watermarking that differs fundamentally from classical insertion-deletion channel coding in [Haeupler and Shahrasbi, 2018](https://arxiv.org/pdf/1710.09795) and [Yasunaga, 2024](https://arxiv.org/pdf/2107.01785v3). Our setting imposes three key restrictions absent in classical formulations:
>
> - **Typical-set constraint.** All outputs must lie within the typical set of a fixed base model $P$, ensuring fluency and statistical proximity to natural LLM generations. Classical insertion-deletion coding, by contrast, permits arbitrary codewords across $\Sigma^T$, all length-$T$ sequences over the alphabet, without distributional constraints.
>
> - **Stealth constraint.** The watermark operates through bounded per-token KL drift $D_0$, explicitly limiting deviation of the watermarked sampler $Q_k$ from $P$ at each generation step. Classical insertion-deletion coding imposes no such distributional or stealth requirements on codebook design.
>
> - **Detection task.** The verifier performs binary hypothesis testing ($H_0: P$ vs. $H_1: Q_k$) for watermark attribution, rather than decoding arbitrary messages from an unconstrained codebook.
>
> Under these constraints, the relevant quantity is the sequence-level KL divergence between edited distributions, which in the small-signal limit contracts as $C(\varepsilon) \approx T(1-\varepsilon)^2 D_0$.
>
> Classical insertion-deletion capacity results, such as in [Haeupler and Shahrasbi, 2018] and [Yasunaga, 2024], operate in an unconstrained regime where the encoder can select arbitrary codewords from $\Sigma^T$, all length-$T$ sequences over the alphabet, with no requirement that outputs resemble natural language or remain close to a fixed source distribution. In that setting, rates of $1 - O(\varepsilon)$ and upper bounds such as $(1-H(\varepsilon))/(1-\varepsilon)$ are achievable and well established.
>
> Theorem 2 should therefore not be interpreted as a competing capacity bound for general insertion-deletion channels. Rather, it characterizes the detectability-robustness frontier in the stealth-constrained, fixed-vocabulary regime relevant to practical LLM watermarking—a setting where classical capacity results do not directly apply. The two analyses address complementary questions in distinct parameter regimes (see Appendix D.11 for detailed discussion).
>
> We appreciate the opportunity to clarify these points and would love to discuss more!

---

> ### Author Response · Authors · 2025-11-30
> **Rebuttal to Reviewer 1zUG (Additional Comment)**
>
> In addition to our previous responses, we further clarify why the information-theoretic (IT) frontier is relevant and how it connects to computational detectability.
>
> - **Information-theoretic frontier coincides with the computational frontier for non-cryptographic schemes.**
>
>   For biased and bias-free watermarks, the optimal Neyman–Pearson detector reduces to explicit token-level statistics computable in time linear in sequence length $T$, satisfying the PPT requirement. In the revised manuscript, we show $\mathrm{Detect_{\mathrm{comp}}} = \mathrm{Detect_{\mathrm{IT}}}$ and $\mathrm{Power_{\mathrm{comp}}} = \mathrm{Power_{\mathrm{IT}}}$ for these schemes. The TV and KL bounds we derive therefore characterize the performance attainable by *polynomial-time detectors*, rather than serving as loose upper bounds. This situates prior computational analyses (e.g., Gloaguen et al.) within a unified information-theoretic framework.
>
> - **Cryptographic schemes represent the source of a computational–statistical gap.**
>
>   For PRF/ PRG-based watermarks, we show that a separation can occur: constructions exist where $\mathrm{TV}(P,Q) > 0$ and an unbounded detector could distinguish $P$ from $Q$, yet every keyless PPT adversary achieves only negligible advantage unless the underlying primitive is broken. Our examples in Appendices C.5 and D.10 suggest that such gaps arise from cryptographic hardness rather than sampling geometry. The IT frontier thus provides *an outer envelope that appears tight for non-cryptographic schemes and remains a valid upper bound for cryptographic schemes, indicating where computational assumptions become relevant to evade detection*.
>
> We believe these additional details clarifies your concerns; the corresponding changes corresponding to this comment are highlighted in magenta and can be found [here](https://anonymous.4open.science/r/Catch-22-Pareto-Frontier-Watermark-in-LLMs-040B/Highlighted-revised-manuscript-post-rebuttal.pdf). Given the recent changes to the ICLR discussion process, we understand that no dialogue is henceforth possible, but we appreciate having had the opportunity to provide these clarifications.

---

### Official Review · Reviewer_s5Qx · 2025-11-07

**Soundness:** 3
**Presentation:** 3
**Contribution:** 3
**Rating:** 6
**Confidence:** 2

**Summary:**

This paper presents a rigorous information-theoretic framework to characterize the "Catch-22" in LLM watermarking, a fundamental **trilemma** between **robustness**, **low detectability**, and **reliable verification**.

The authors establish a hierarchy of **detectability** (quantified by Total Variation distance) based *solely* on the sampling transformation (Theorem 1), proving scaling bounds for all four analyzed strategies: Greedy ($O(1)$), Biased ($O(|\delta|\sqrt{T})$), Bias-free ($O(\sqrt{T})$), and Distribution-preserving (0).

**Robustness** is then characterized using information capacity (Theorem 2). The core mechanism revealed is that the available information budget $C(\epsilon)$ **contracts quadratically** (specifically, $C(\epsilon) \approx T(1-\epsilon)^2 D_0$) with the edit rate $\epsilon$. This mathematical mechanism proves the inverse relationship: achieving high robustness (requiring a high initial information budget $D_0$) inherently necessitates high detectability (which, per Theorem 1, also scales with the parameters that increase $D_0$).

Based on these constraints, the authors derive a **Pareto-optimal hybrid watermarking scheme** (Theorem 3). This is not a simple heuristic switch, but an optimal construction derived by minimizing a **composite loss function ($\mathfrak{L}$)** that jointly optimizes for target detection power ($1-\beta$), stealth constraints ($\tau, M$), and parameter amplitude.

Experimental validation on Llama 2 7B and Mistral 7B using paraphrasing attacks confirms these theoretical bounds and, crucially, demonstrates that the proposed hybrid scheme **uniquely traces the Pareto-optimal frontier across all noise regimes**, outperforming any fixed scheme.

**Strengths:**

1. **Rigorous theoretical foundation**: The information-theoretic framework with formal proofs (Theorems 1-3, Lemmas 1-5) provides principled understanding beyond empirical observations.

2. **Comprehensive experimental validation**: Table 1 systematically evaluates multiple watermarking families (biased, bias-free, distribution-preserving) across two models and multiple attack scenarios, confirming theoretical predictions.

3. **Breadth of analysis**: Coverage spans greedy, biased, bias-free, and distribution-preserving sampling with unified treatment.

4. **Novel impossibility result**: Corollary 1 establishes fundamental limits showing the trilemma cannot be circumvented by clever engineering.

**Weaknesses:**

1. **Limited attack diversity**: Only paraphrasing attacks (DIPPER, OPT-2.7B) are evaluated. Missing: synonym substitution, back-translation, model-based attacks, and adversarial prompting attacks from Liu et al. (2025) cited in the paper.

2. **Independence assumption not validated**: The edit channel assumes i.i.d. token substitution (Eq. 43), but real paraphrasing introduces semantic dependencies. No empirical validation that this approximation holds or analysis of when it breaks down.

3. **Incomplete hybrid scheme specification**: Theorem 3 provides allocation rules but lacks concrete algorithm for runtime edit rate estimation $\hat{\epsilon}$, which is critical for deployment. The paper acknowledges this (Impact Statement) but doesn't address it.

**Questions:**

1. **Edit channel validity**: Can you provide empirical validation that real paraphrasing attacks approximately satisfy the i.i.d. edit assumption? What is the distribution of actual edit patterns in DIPPER outputs?

3. **Multi-key scenarios**: Theorem 1 analyzes fixed-key bias-free schemes. How does detectability change if the adversary observes outputs from multiple different keys?

4. **Comparison with Moitra & Golowich**: You mention their scheme requires exponential vocabulary (Section 2). Can you clarify the precise relationship between your impossibility results and theirs?

---

> ### Author Response · Authors · 2025-11-25
>
> Dear Reviewer s5Qx,
>
> Thanks for the detailed and constructive feedback. In response to the points raised, we have made the following attempts to address each concern in detail below.
>
> 1. **Additional paraphrasing experiments to address limited attack diversity**
>
>       We have expanded our experimental evaluation to address the reviewer's concern about attack diversity. Appendix G and Table 2 now include the trade-off analysis with three GPT-3.5–based paraphrasing attacks: synonym-substitution, adversarial "remove watermark" prompting, and back-translation (English→French→English). For synonym substitution and adversarial prompting, the relative ordering of watermarking schemes in terms of detectability-robustness trade-off remains essentially unchanged across attack types. Back-translation presents a distinct case: its unconstrained semantic rewriting produces substantially higher edit rates, causing all watermarking schemes to fail—precisely the high-noise regime predicted by Theorem 2 and Figure 2(b). These results have also been updated in our [anonymous code repository](https://anonymous.4open.science/r/Catch-22-Pareto-Frontier-Watermark-in-LLMs-040B/Additional-Results/Performance-evaluation.md).
>
> 2. **Validating the IID assumption for the edit model**
>
>       Our token-level measurements reveal that practical paraphrasing attacks, although not producing perfectly i.i.d. edits, exhibit behavior that closely aligns with the theoretical model underlying our analysis. Specifically, the *marginal* edit probability at each token position is concentrated around a global rate $\epsilon$, with most realistic paraphrasers showing limited positional variation. DIPPER, for instance, exhibits a close to flat edit profile across all positions, while synonym substitution demonstrates minor positional dependencies.
>
>       Back-translation represents a notable exception to this pattern. Its unconstrained semantic rewriting introduces bursts of correlated edits that violate any fixed-rate, independent-edit model. Overall, for paraphrasers that maintain a stable global edit rate and exhibit limited positional correlation, the i.i.d. edit channel provides a sound first-order approximation for analyzing watermark robustness. The results of this experimental validation are available [here](https://anonymous.4open.science/r/Catch-22-Pareto-Frontier-Watermark-in-LLMs-040B/Additional-Results/Readme.md), and are described in detail in Appendix G.1 of the revised manuscript.
>
> 3. **Clarification on Incomplete hybrid scheme specification**
>
>       Theorem 3 is a *design-time* result: the hybrid watermark does not estimate per-sample edit rates during inference. Instead, we specify an offline calibration procedure that measures edit rates on representative text pairs and selects a conservative design value $\hat{\varepsilon}$ (e.g., a high percentile). This estimate is used once to compute fixed hybrid parameters $(\sigma^{2\star},\delta^\star,\gamma^\star)$ for deployment. Thus, Theorem 3 provides the theoretical allocation rules, while the practical estimation procedure for real-world deployment is detailed in Appendix E.10.
>
> 4. **Multi‑key scenarios**
>
>       If outputs are generated under multiple watermark keys, the adversary effectively sees the **key-averaged distribution** $\overline{Q}=\mathbb{E}_{k}[Q_k]$. Detectability is always defined with respect to the distribution the adversary can observe, so Theorem 1 applies with $Q$ replaced by $\overline{Q}$. We have added further details in Appendix F.2 of the revised manuscript.
>
> 5. **Comparison with Moitra & Golowich**
>
>      The robust construction of Moitra & Golowich (2024) requires alphabet size $|\Sigma(\lambda)| \ge n(\lambda)^{C_{2}\frac{1}{\alpha}\log(1/\alpha)}$, where $\alpha$ is the entropy rate. For natural language entropy levels, this becomes effectively exponential in $1/\alpha$, far exceeding practical LLM vocabularies (30k–100k tokens). Their result is thus an *achievability theorem* for the large-alphabet regime where vocabulary scales with robustness parameters, whereas our results establish *impossibility theorems* for the fixed-vocabulary regime characterizing practical language models. These complementary perspectives address different parameter regimes; we clarify this relationship in Appendix A.5.
>
> We hope that we have been able to do justice to your questions and would love to discuss more!

---

> > ### Comment · Reviewer_s5Qx · 2025-11-27
> >
> > I thank the authors for their detailed response and the additional experiments. The revision has effectively addressed my technical questions regarding attack diversity and the validity of the edit channel assumption.
> >
> > 1. Attack Diversity: The inclusion of GPT-3.5-based attacks (Table 2) provides a more comprehensive empirical picture. It is valuable to see that back-translation, which operates in a high-noise regime, leads to poor robustness and detectability across all schemes. This failure case is consistent with the impossibility results predicted by Theorem 2.
> >
> > 2. I.I.D. Assumption: The token-level analysis (Figure 3) is a good addition. It clarifies that the i.i.d. model is a reasonable first-order approximation for stealthy paraphrasers such as DIPPER. Simultaneously, the analysis confirms that back-translation behaves as a high-noise, non-i.i.d. channel that falls outside this approximation regime, which effectively delineates the scope of the theoretical bounds.
> >
> > 3. Hybrid Scheme: The offline calibration procedure added in Appendix E.10 clarifies the implementation details. However, relying on a design-time estimate ($\hat{\epsilon}$) represents a practical limitation against adaptive adversaries who might vary their attack intensity dynamically during inference.
> >
> > The authors have done a commendable job strengthening the paper's completeness and validating their claims. The paper presents a solid information-theoretic characterization of the trade-offs. However, the findings primarily formalize the expected limitations (the "trilemma") rather than offering a breakthrough that fundamentally overcomes these constraints. The hybrid scheme, while Pareto-optimal, is an optimization within these bounds rather than a new paradigm. Therefore, I believe the current score of 6 accurately reflects the paper's quality: it is a theoretically sound and empirically validated contribution worthy of acceptance. I maintain my positive rating.

---

> ### Author Response · Authors · 2025-11-30
> **Follow-up Comment for Reviewer s5Qx**
>
> Dear Reviewer s5Qx,
>
> We sincerely thank you for the thoughtful follow-up and for recognizing the strengthened theoretical and empirical components of the revision. We appreciate the acknowledgment that the additional GPT-3.5 attacks, token-level edit analysis, and hybrid scheme calibration procedure have resolved the earlier concerns. Your detailed engagement has substantially improved the clarity, completeness, and scope of the final manuscript, and we are grateful for your supportive assessment.

---

### Author Response · Authors · 2025-11-25
**Summary of Changes to the Manuscript After Author–Reviewer Discussion**

We thank all reviewers (s5Qx, 1zUG, pLoi) for their constructive feedback. Below, we list the main reviewer concerns followed by the manuscript modifications that address them.

1. **Information-Theoretic detectability–robustness frontier** (Abstract, Introduction, Conclusion)

   We theoretically establish an information-theoretic detectability–robustness frontier. For biased and bias-free schemes, this frontier is achieved by computationally bounded adversaries, while cryptographic schemes lie strictly below it.

2. **Relationship to Moitra–Golowich Paper** (Section 2, Appendix A.5)

   We quantify the alphabet-size requirement in [Moitra–Golowich](https://arxiv.org/abs/2406.02633) and clarify how their vocabulary scaling differs from practical LLM vocabularies. We position their construction as an achievability result that complements our fixed-vocabulary impossibility regime.


3. **Information-Theoretic vs. Computational Hardness** (Definitions 1,2, and Theorems 1–2 in Sections 3.1 and 3.2)

   We formally define computational detectability and robustness, and prove that $\mathrm{Detect_{\mathrm{comp}}} \le \mathrm{Detect_{\mathrm{IT}}}$, with equality $\mathrm{Detect_{\mathrm{comp}}} = \mathrm{Detect_{\mathrm{IT}}}$ and $\mathrm{Power_{\mathrm{comp}}} = \mathrm{Power_{\mathrm{IT}}}$ for greedy, biased, and bias-free sampling via Neaman-Pearson tests. We also clarified Theorem 1's $O(1)$ / $O(|\delta|\sqrt{T})$ / $O(\sqrt{T})$ / $0$ hierarchy and framed Theorem 2 as a detectability–robustness trade-off under fixed vocabulary and bounded per-token KL drift.


4. **Relation to Classical Coding-Theoretic Bounds** (Section 3.2, Appendix D.11)

   We clarify that classical edit-channel capacity results (e.g., rates based on $(1-H(\varepsilon))/(1-\varepsilon)$) assume unconstrained codebooks over $\Sigma^T$. In contrast, our bounds operate under typical-set and stealth constraints tied to a fixed LLM with bounded per-token KL drift.


5. **Cryptographic Constructions and the Computational–Statistical Gap** (Appendices C.5, D.10)

   We show that for PRG/PRF-based cryptographic watermarking, $\mathrm{TV}>0$ can arise with negligible computational detectability for a keyless PPT adversary. We explicitly demonstrate that the IT frontier is tight for non-cryptographic schemes and serves as an outer envelope for cryptographic schemes.


6. **Attack Diversity and Edit-Channel Validation** (Section 5, Appendix G)

   We added GPT-3.5 synonym-substitution, adversarial "remove watermark," and back-translation attacks (new rows in [Table 2](https://anonymous.4open.science/r/Catch-22-Pareto-Frontier-Watermark-in-LLMs-040B/Additional-Results/Performance-evaluation.md)). Furthermore, we also present token-level edit-pattern analysis ([Figure 3](https://anonymous.4open.science/r/Catch-22-Pareto-Frontier-Watermark-in-LLMs-040B/Additional-Results/Readme.md), Appendix G.1), showing that DIPPER and synonym substitution are well-approximated by an i.i.d. edit channel, while back-translation behaves as a high-noise, non-i.i.d. channel in which all schemes fail as predicted.


7. **Hybrid Scheme and Edit-Rate Estimation** (Appendix E.10)

   We clarified that Theorem 3 is a design-time optimization result and added an explicit offline calibration procedure to estimate edit rate.


8. **Watermarks beyond sampling** (Appendix F)

   We provide a discussion on the potential extension of our framework to cover weight-based and training-time watermarks when they induce $q_t \neq p_t$ (e.g., GaussMark), and added analysis of multi-key scenarios via the key-averaged distribution $\overline{Q}$.


9. **Figures and Notation Updates** (Figure 2 and Theorem 1)

   We updated [Figure 2(a)](https://anonymous.4open.science/r/Catch-22-Pareto-Frontier-Watermark-in-LLMs-040B/Analysis-Results/AnnotatedFigA.pdf) to use a linear $T$-axis and show sublinear TV growth consistent with the theoretical bounds. We also clarified that $g_t$ and the variance terms in Theorem 1 appear inside expectations.

---

The revised manuscript with all changes highlighted in ForestGreen color is available [here](https://anonymous.4open.science/r/Catch-22-Pareto-Frontier-Watermark-in-LLMs-040B/Highlighted-revised-manuscript-final.pdf). The main criticism of this work, raised by Reviewers 1zUG and pLoi, was that information-theoretic (IT) bounds are disconnected from practical LLM watermarking. In our rebuttal, we establish that the IT analysis (Appendices C.5, D.10) is not a loose abstraction but the operationally relevant quantity for practical watermarking systems. In particular, we show that the IT frontier coincides exactly with the computational frontier for biased and bias-free watermarking schemes. The remaining reviewer comments are addressed as described above.

We thank all the reviewers once again for their engagement and valuable feedback, which has greatly strengthened our submission.

---

### Meta-Review · Area_Chair_YLiF · 2026-01-07

**Summary:**

Three main concerns by the reviewers inform my decision for this paper. First, the paper's contributions were viewed as limited in novelty and practical impact. In particular, s5Qx, though giving a comparatively positive review, characterized the paper as largely formalizing expected limitations (the “trilemma”) rather than providing a clear breakthrough. Second, Reviewer 1zUG emphasized that the main result seemed obvious and pointed to multiple misunderstandings and misreferences to prior work, giving the lowest possible score. Finally, pLoi also favored rejection, highlighting a misconnection to previous work (notably the GaussMark watermark) and questioning if focusing on TV actually translated to meaningful insights on watermarking problem. Overall, the rebuttal did not shift the reviewers’ confidence in the paper’s contribution, including for s5Qx (the most positive reviewer).

Moreover, the re-submitted manuscript is 11 pages long. ICLR has a 10 page restriction for camera-ready papers -- though this is only a more "bureaucratic" concern.

**Reviewer Concerns:**

**Concerns that were partially addressed:** pLoi indicated that some points were addressed, though they still maintained their original score.

**Concerns still outstanding after the rebuttal:** s5Qx remained of the view that the work mainly formalizes expected limitations rather than advancing a more meaningful practical contribution to LLM watermarking. 1zUG’s maintained their core objections: they found the result obvious and flagged misunderstandings to prior work. pLoi continued to view the paper as insufficiently connected to prior work (especially GaussMark) and remained concerned that the object being controlled has limited relevance to watermarking, which also ties to questions about practical applicability.

I also note that the paper misses important recent work at the intersection of information theory and watermarking, including:
* Li, Xiang, Feng Ruan, Huiyuan Wang, Qi Long, and Weijie J. Su. "Robust detection of watermarks for large language models under human edits." Journal of the Royal Statistical Society Series B: Statistical Methodology (2025)
* Tsur, Dor, et al. "HeavyWater and SimplexWater: Distortion-free LLM Watermarks for Low-Entropy Distributions." The Thirty-ninth Annual Conference on Neural Information Processing Systems..

**Reviewer Scores:**

s5Qx: explicitly states that they maintain their score.

pLoi: explicitly states that they maintain their score.

1zUG: 1zUG did not appear to engage post-rebuttal and had originally given the lowest score. Given an original score of 0, I believe their score would not have changed to an acceptance recommendation.

---

### Decision · Program_Chairs · 2026-01-26

Reject